# Hijacking a bacterial ABC transporter for genetic code expansion

Tarun Iype[1,3], Maximilian Fottner[1,3], Paul Böhm[1], Carlos Piedrafita[2], Yannis Möller[1], Michael Groll[2] & Kathrin Lang[1,2 ✉]

The site-specific encoding of non-canonical amino acids (ncAAs) provides a powerful tool for expanding the functional repertoire of proteins[1–4]. Its widespread use for basic research and biotechnological applications is, however, hampered by the low efficiencies of current ncAA incorporation strategies. Here we reveal poor cellular ncAA uptake as a main obstacle to efficient genetic code expansion and overcome this bottleneck by hijacking a bacterial ATP-binding cassette (ABC) transporter[5] to actively import easily synthesizable isopeptide-linked tripeptides that are processed into ncAAs within the cell. Using this approach, we enable efficient encoding of a variety of previously inaccessible ncAAs, decorating proteins with bioorthogonal[6] and crosslinker[7] moieties, post-translational modifications[8,9] and functionalities for chemoenzymatic conjugation. We then devise a high-throughput directed evolution platform to engineer tailored transporter systems for the import of ncAAs that were historically refractory to efficient uptake. Customized *Escherichia coli* strains expressing these evolved transporters facilitate single and multi-site ncAA incorporation with wild-type efficiencies. Additionally, we adapt the tripeptide scaffolds for the co-transport of two different ncAAs, enabling their efficient dual incorporation. Collectively, our study demonstrates that engineering of uptake systems is a powerful strategy for programmable import of chemically diverse building blocks.

Co-translational incorporation of ncAAs via genetic code expansion (GCE) enables precise reprogramming of the proteome's chemical diversity. By leveraging orthogonal aminoacyl-tRNA synthetases (aaRSs), a wide range of functionalities, including post-translational modifications (PTMs), bioorthogonal handles, crosslinking moieties, spectroscopic probes and photocaged amino acids, have been site-specifically introduced into proteins of interest (POIs) across all domains of life, typically via amber suppression[1–9]. These strategies offer powerful tools for studying and manipulating protein functions and for generating proteins with therapeutic and biotechnological importance.

Despite substantial progress, broad implementation of GCE remains limited by low protein production yields. Inefficiencies arise from insufficient substrate activation by orthogonal aaRSs, as well as unfavourable competition of aminoacylated tRNAs with release factors at introduced nonsense codons[10]. In addition, many ncAAs require advanced expertise in chemical synthesis and are used at high concentrations in typical experiments, making them prohibitively expensive, a factor that is exacerbated by the low incorporation yields.

Multiple efforts have addressed these limitations through optimized aaRS/tRNA expression systems combined with novel selection and evolution strategies that enhance suppression efficiencies[11–13]. Additional advances include orthogonal ribosomes[14], release-factor knockouts[15,16] and recoded genomes that permit sense codon reassignment[17–20].

A less explored, but critical factor lies in the intracellular bioavailability of ncAAs. In most GCE applications, ncAAs are added exogenously to cells and taken up by passive diffusion or via native amino acid transporters. This often results in low intracellular concentrations, which is especially detrimental for aaRS/ncAA pairs with low catalytic efficiency, for which aminoacylation operates below optimal conditions[10,21]. Furthermore, reliance on passive diffusion and endogenous importers restricts the available design space for novel building blocks. For a small number of ncAAs, efforts in engineering biosynthetic pathways to produce them directly within cells have overcome some of these limitations, but such approaches require substantial strain development and are currently applicable to only a narrow range of functionalities[22–26].

Engineering membrane transport systems presents a promising, yet underexplored, strategy to enhance ncAA uptake and has potential to be widely applicable. Prior work has investigated the substrate scope of a periplasmic leucine-binding protein towards known ncAAs[27] and 'Trojan horse' strategies, in which ncAAs are conjugated to carrier groups that facilitate recognition and uptake by transporters[28–32].

Here we leverage a modular propeptide-based strategy coupled with engineering of a bacterial ABC transporter for programmable import of ncAAs, enabling their efficient encoding in *E. coli*. We demonstrate that isopeptide-linked tripeptides (Z-XisoK, where Z and X are natural or non-canonical residues) are actively imported into

[1]Department of Chemistry and Applied Biosciences (D-CHAB), ETH Zurich, Zurich, Switzerland. [2]Center for Protein Assemblies, TUM School of Natural Sciences, Department of Bioscience, Technical University of Munich, Garching, Germany. [3]These authors contributed equally: Tarun Iype, Maximilian Fottner. ✉e-mail: kathrin.lang@org.chem.ethz.ch

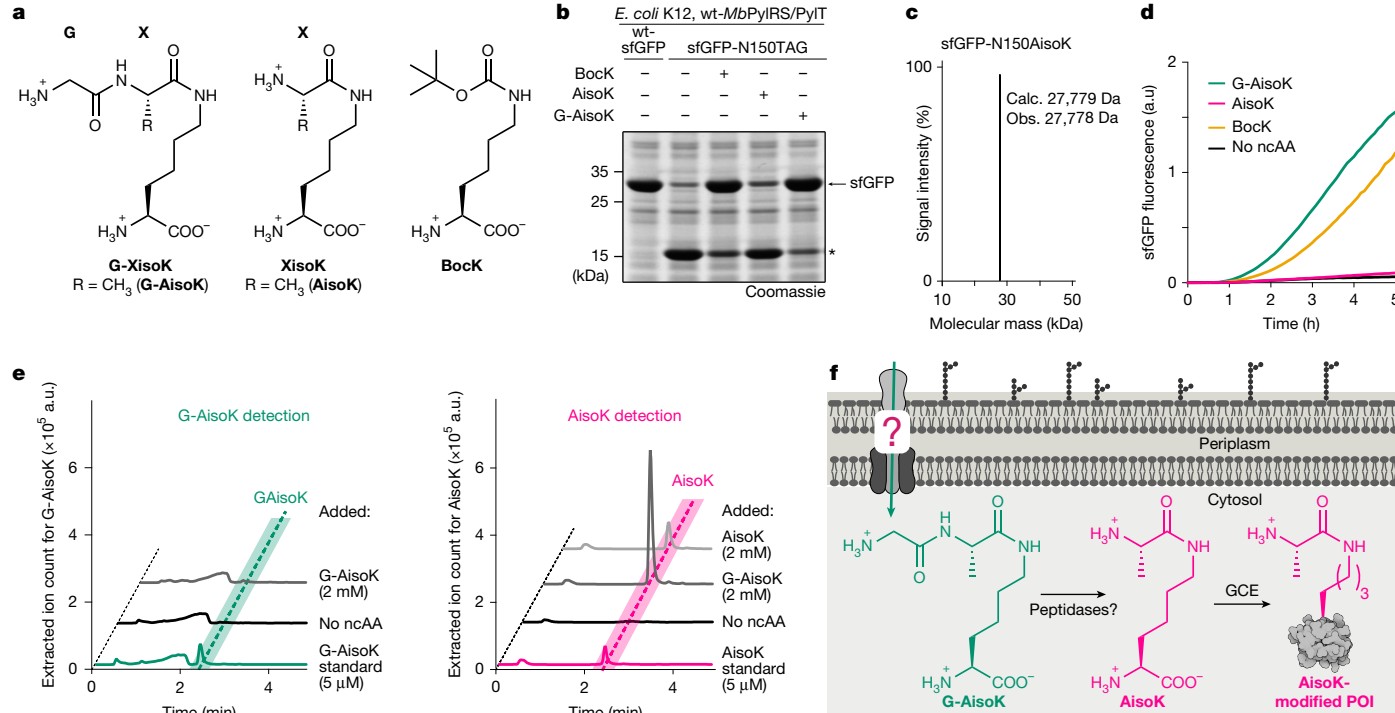

**Fig. 1 | Isopeptide-linked tripeptides are privileged scaffolds for efficient *E. coli* uptake. a**, Chemical structures of G-XisoK, XisoK and BocK. X is alanine in G-AisoK and AisoK. **b**, SDS–PAGE analysis of wild-type sfGFP (wt-sfGFP) and sfGFP-N150TAG expression in K12 bearing wt-*Mb*PylRS/PylT in the presence of 2 mM BocK, AisoK or G-AisoK. Asterisk indicates truncated protein. Consistent results were obtained over three independent replicate experiments. **c**, LC–MS analysis of sfGFP-N150AisoK. Calc., calculated molecular mass; obs., observed molecular mass. **d**, Time-course measurements of sfGFP fluorescence from K12 cultures expressing sfGFP-N150TAG and wt-*Mb*PylRS/PylT in the presence of G-AisoK, AisoK or BocK, or grown in the absence of ncAAs. Consistent results were obtained over three independent replicate experiments. **e**, Extracted ion chromatograms for determining intracellular concentrations of G-AisoK and AisoK by an LC–MS assay, performed on K12 cell extracts. Intracellular G-AisoK concentrations in K12 grown with 2 mM G-AisoK are negligible. Intracellular AisoK concentrations in K12 grown with 2 mM G-AisoK are 5- to 10-fold higher than when grown with 2 mM AisoK. Consistent results were obtained over three independent replicate experiments. **f**, Proposed model for increased AisoK incorporation. The tripeptide G-AisoK is actively taken up via an *E. coli* transporter. Within the cytosol, G-AisoK is processed to AisoK, which is a substrate for wt-*Mb*PylRS/PylT and is incorporated site-specifically into a POI.

*E. coli* via the oligopeptide permease (Opp) and processed intracellularly, resulting in high accumulation of Z and XisoK. Using G-XisoK scaffolds, we efficiently incorporate 11 previously inaccessible XisoK ncAAs bearing functionalities such as bioorthogonal handles, crosslinkers and PTMs. We further devise a directed evolution platform to reprogramme the periplasmic binding protein of the transporter (OppA) for preferential uptake of G-XisoK tripeptides over competing linear peptides that are present in commonly used growth media. Expanding this approach, we adapt our platform for importing Z-XisoK tripeptides with diverse Z groups, including bulky or negatively charged ncAAs that are cell-impermeable on their own. Genomic integration of evolved OppA variants creates *E. coli* strains that are tailored for efficient single and multi-site ncAA incorporation. Finally, we adapt our scaffolds for the incorporation of two distinct ncAAs, mediated by their concomitant transport via a single tripeptide. Together, our results establish transporter engineering as a powerful strategy to unlock and customize ncAA import for the efficient production of proteins with an expanded alphabet.

## G-AisoK is transported into *E. coli*

Previous work in our group combined transpeptidases with GCE to generate defined protein–protein conjugates. By site-specifically encoding an azide-caged diglycine acceptor motif (AzGGisoK) (Supplementary Fig. 1a) followed by on-protein Staudinger reduction, GGisoK-bearing proteins can undergo transpeptidation with donor proteins bearing a C-terminal recognition sequence. We applied this strategy to generate ubiquitin (Ub)- and Ub-like modifier (Ubl)–POI conjugates using sortase or an asparaginyl endopeptidase as transpeptidases[33–35].

To diversify the linker sequence in the generated protein conjugates, we explored site-specific incorporation of ncAAs resembling a general G-XisoK scaffold (Fig. 1a). Supplementing *E. coli* K12 with the alanine-bearing G-XisoK tripeptide (G-AisoK; Fig. 1a) enabled efficient amber suppression of superfolder GFP (sfGFP-N150TAG) using the wild-type *Methanosarcina barkeri* pyrrolysine-tRNA synthetase/tRNA pair (wt-*Mb*PylRS/PylT), with yields comparable to that of wild-type sfGFP production and similar to the gold-standard ncAA BocK (Fig. 1a,b). Mass spectrometric analysis revealed site-specific incorporation of AisoK (Fig. 1c), suggesting intracellular cleavage of the N-terminal glycine, either on the free ncAA, co-translationally or post-translationally. By contrast, direct supplementation of K12 with AisoK resulted in negligible sfGFP production (Fig. 1b). This was corroborated by live-cell sfGFP fluorescence measurements, which showed minimal signal with AisoK, whereas G-AisoK induced earlier and stronger fluorescence than BocK (Fig. 1d).

Similarly, G-AisoK-mediated AisoK incorporation was observed for other amber-containing target proteins (Supplementary Fig. 1b). To investigate the underlying mechanism, we performed liquid chromatography–mass spectrometry (LC–MS)-based uptake assays[21,36]. We did not detect any intracellular G-AisoK after G-AisoK supplementation, but AisoK accumulated at fivefold to tenfold higher concentrations compared with supplementing K12 with AisoK directly (Fig. 1e and Supplementary Fig. 1c).

These findings led us to hypothesize that a specific transport mechanism actively imports G-AisoK into cells. Within the cytosol, G-AisoK is enzymatically processed to AisoK, which accumulates in

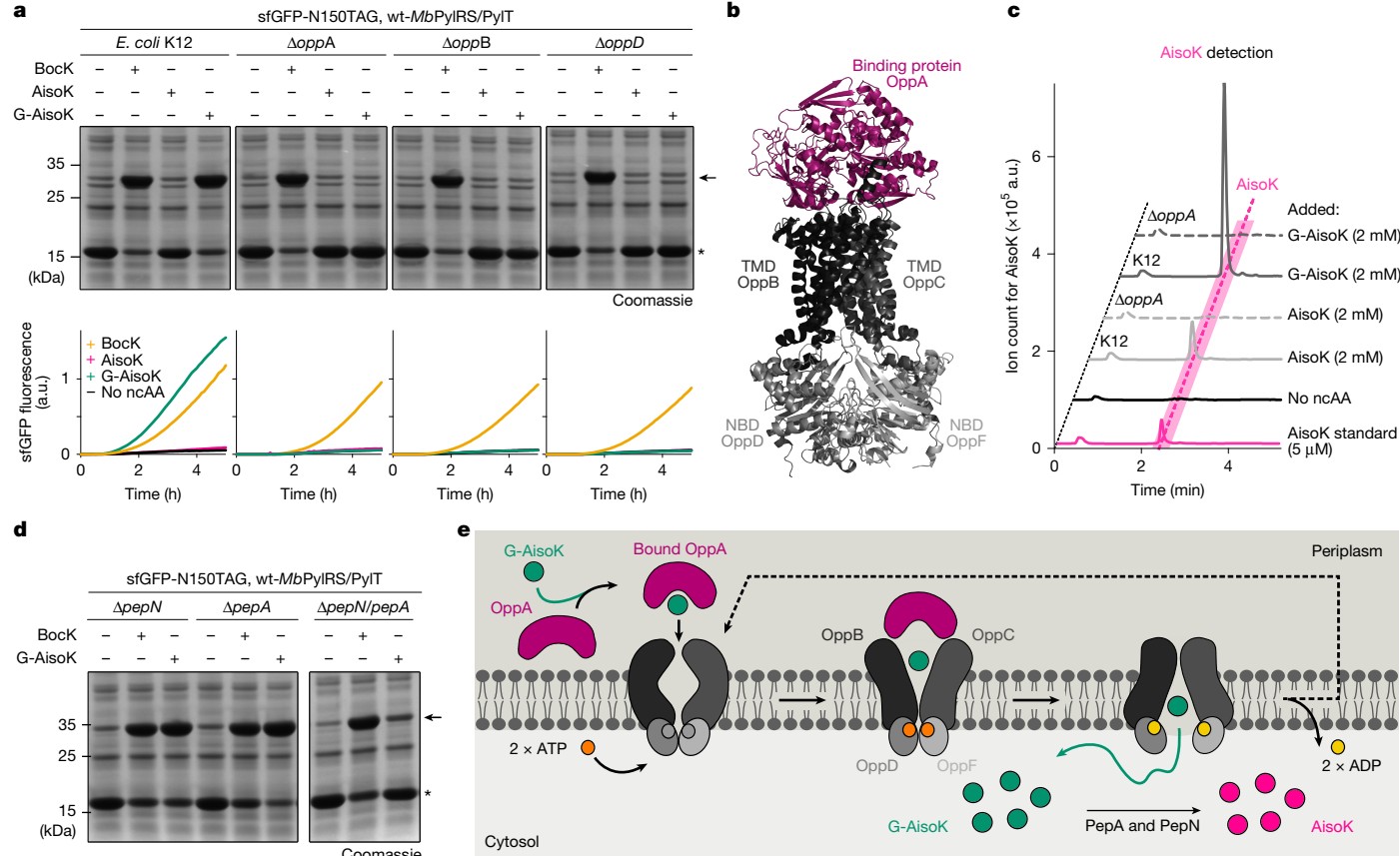

**Fig. 2 | The Opp transporter is responsible for efficient G-AisoK uptake.**
**a**, SDS–PAGE analysis (top) and time-course fluorescence measurements (bottom) of sfGFP-N150TAG expression in the presence of BocK, AisoK or G-AisoK in wild-type K12 and in Δ*opp*A, Δ*opp*B or Δ*opp*D knockouts, indicating that the Opp transporter is responsible for G-AisoK uptake. Results for other knockouts can be found in Extended Data Fig. 1. Consistent results were obtained over three independent replicate experiments. Arrow indicates full-length sfGFP, asterisk indicates truncated sfGFP. **b**, AlphaFold2 predicted structure of the Opp transporter, consisting of the periplasmic binding protein OppA, two TMDs (OppB and OppC) and two NBDs (OppD and OppF). **c**, Extracted ion chromatograms of *E. coli* K12 lysates for determination of intracellular AisoK concentrations in wild-type K12 versus Δ*opp*A. Genomic deletion of *oppA* results in undetectable AisoK concentrations when growing cells with 2 mM G-AisoK. Consistent results were obtained over three independent replicate

experiments. **d**, SDS–PAGE analysis of sfGFP-N150TAG expression with BocK or G-AisoK in single peptidase knockouts Δ*pepN* and Δ*pepA* and the double knockout Δ*pepN/pepA*. G-AisoK-dependent full-length sfGFP expression is significantly reduced in Δ*pepN/pepA*, indicating that pepA and pepN are the main peptidases responsible for cleavage of the N-terminal glycine. Results for other knockouts are presented in Supplementary Fig. 3. Consistent results were obtained over three independent replicate experiments. Arrow indicates full-length sfGFP, asterisk indicates truncated sfGFP. **e**, Proposed mechanism of Opp-mediated uptake. G-AisoK binds to OppA in the periplasm and is shuttled to membrane-bound OppB and OppC. The tripeptide is actively transported into the cytosol in an ATP-dependent manner, where it is cleaved by pepN and pepA to AisoK. OppA, in its apo-form, is released from the TMDs to allow binding of new G-AisoK.

high concentrations and serves as a substrate for *Mb*PylRS, leading to efficient AisoK encoding (Fig. 1f).

## An ABC transporter enables G-XisoK uptake

In Gram-negative bacteria such as *E. coli*, small peptides enter the periplasm by diffusion through outer membrane porins[37]. Within the inner membrane, two major peptide-transporter classes facilitate peptide uptake into the cytosol: proton-dependent oligopeptide transporters (POTs) and ABC transporters[5] (Supplementary Fig. 2a). To identify a potential uptake system for G-AisoK, we screened *E. coli* single-gene knockouts[38] with deletions of individual transporters or transporter domains for amber suppression of sfGFP-N150TAG in the presence of the wt-*Mb*PylRS/PylT pair and G-AisoK. We hypothesized that loss of a required transporter would reduce or abolish sfGFP expression. Whereas deletion of POT family members and dipeptide-specific ABC transporters had no effect, individual knockouts of genes constituting the *opp* operon completely abolished sfGFP expression with G-AisoK (Fig. 2a, Extended Data Fig. 1a and Supplementary Fig. 2b,c).

The Opp ABC transporter comprises the periplasmic binding protein (OppA), two transmembrane domains (TMDs) that span the inner membrane (OppB and OppC) and two cytosolic nucleotide-binding domains (NBDs) that drive ATP hydrolysis (OppD and OppF) (Fig. 2b). Peptide-bound OppA docks to the TMDs, triggering ATP-binding and substrate translocation into the cytosol[5]. Individual deletions of OppA, or any of the two TMDs or NBDs led to complete loss of amber suppression and sfGFP fluorescence with G-AisoK, but not with BocK, indicating Opp-dependent G-AisoK uptake (Fig. 2a and Extended Data Fig. 1a). Uptake assays confirmed that intracellular BocK levels were unchanged in Δ*oppA*-K12 compared with wild-type K12, whereas AisoK, which accumulated in millimolar concentrations in K12 treated with G-AisoK, was undetectable when *oppA* was deleted (Fig. 2c and Extended Data Fig. 1b).

To identify the enzyme responsible for processing of G-AisoK to AisoK, we performed amber suppression experiments with G-AisoK using single-gene knockouts that lack specific aminopeptidases[38]. However, none of the ten tested knockouts exhibited an effect on the amber suppression yield (Fig. 2d and Supplementary Fig. 3a). We therefore

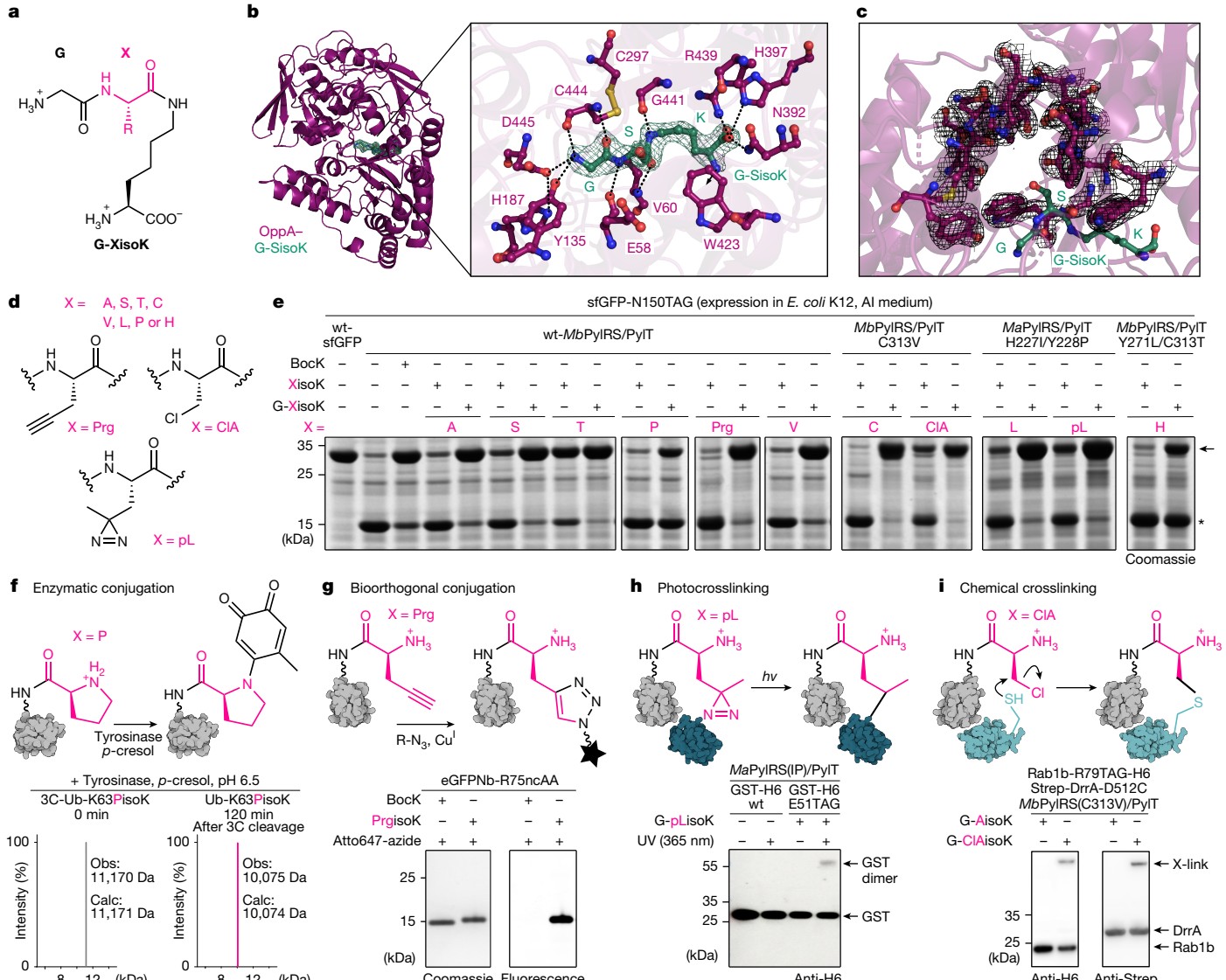

**Fig. 3 | A versatile G-XisoK toolbox. a**, Structure of a generalized G-XisoK tripeptide. **b**, X-ray structure of OppA bound to G-SisoK (PDB ID: 9RD1). G-SisoK forms extensive interactions with OppA residues via its N and C termini and its backbone amide groups. For a detailed description of the interactions, see Supplementary Fig. 4c. **c**, The OppA–G-SisoK complex around the serine side chain reveals a large cavity that is capable of accommodating bulky side chains. **d**, All functional groups incorporated via the G-XisoK scaffold. **e**, SDS–PAGE analysis of sfGFP-N150TAG expression in the presence of either 2 mM XisoK or G-XisoK. All G-XisoK derivatives show higher levels of full-length sfGFP expression using the corresponding PylRS/PylT pairs compared with cells grown with the corresponding XisoK. Arrow indicates full-length sfGFP; asterisk indicates truncated sfGFP. LC–MS analyses of purified sfGFP and Ub variants confirming the incorporation of XisoK derivatives are shown in Supplementary Figs. 5, 6 and 8. *Ma*PylRS, *Methanomethylophilus alvus* PylRS. **f**, LC–MS analysis of tyrosinase-mediated labelling of 3C-Ub bearing PisoK at K63 with *p*-cresol demonstrates quantitative conversion. For details and full data see Extended Data Fig. 2. **g**, SDS–PAGE analysis of CuAAC labelling of purified eGFPNb-R75PrgisoK with an Atto647-Azide fluorophore. No labelling is observed for eGFPNb-R75BocK. For details and full data see Extended Data Fig. 3a. **h**, Western blot analysis of GST-dimer crosslinking for GST-E51pLisoK after 365 nm UV illumination. No crosslink is observed for wild-type GST. For details and full data see Extended Data Fig. 3b. **i**, Western blot analysis of proximity-induced chemical crosslinking between Rab1b-R79ClAisoK and its interactor DrrA-D512C_{339–522}. Cells expressing both binding partners in the presence of G-ClAisoK display a higher molecular weight band corresponding to the crosslinked complex in both anti-H6 and anti-streptavidin (Strep) blots. Full data and further experiments can be found in Extended Data Fig. 4. **e**–**i**, Consistent results were obtained over three independent replicate experiments.

generated multi-peptidase knockouts using a CRISPR–Cas12a-based genome editing platform for *E. coli*[39]. Notably, only cells with both *pepN* and *pepA* deleted (Δ*pepN/pepA*), showed a marked reduction in sfGFP expression with G-AisoK, whereas amber suppression yields with BocK remained unaffected (Fig. 2d and Supplementary Fig. 3b). Complementation with either pepA or pepN restored sfGFP expression with G-AisoK, indicating that either peptidase is sufficient for G-AisoK processing (Supplementary Fig. 3c).

Together, these findings support a model in which G-AisoK is actively imported via the Opp transporter into the cytosol of *E. coli*, where it is processed by endogenous peptidases, releasing AisoK for efficient amber suppression (Fig. 2e).

## A versatile G-XisoK toolbox

Next, we tested whether amino acids in a general G-XisoK (Fig. 3a) scaffold behaved similarly. Indeed, SisoK, bearing serine instead of alanine, was similarly incorporated in a tripeptide (G-SisoK)-dependent manner (Supplementary Fig. 4a,b). OppA is known to promiscuously bind 2- to 5-amino-acid-long peptides, favouring positively charged side chains.

To explore how OppA distinguishes G-XisoK from XisoK, we solved the crystal structure of OppA bound to G-SisoK (Protein Data Bank (PDB) ID: 9RD1; Supplementary Table 5). The structure shows a good overlap with previous ligand-bound OppA conformations[40] and adopts the closed state, with G-SisoK enclosed in the binding pocket (Fig. 3b). G-SisoK engages in extensive interactions with OppA through its backbone and termini. The N-terminal glycine forms key hydrogen bonds and electrostatic contacts: its protonated α-amine interacts with D445, whereas the C-terminal carboxylate is stabilized by hydrogen bonds involving the side chains of R439, H397 and N392 (Fig. 3b and Supplementary Fig. 4c). To validate these interactions, we expressed OppA variants in ΔoppA. Expression of wild-type OppA fully restored sfGFP expression with G-SisoK, whereas the D445A variant, which disrupts the interaction with the N-terminal α-amine of G-SisoK, did not rescue expression. Mutations targeting the hydrogen-bonding network at the C terminus of G-SisoK (for example, R439A), had less pronounced effects, suggesting that the OppA binding site possesses some structural flexibility (Supplementary Fig. 4d). These results highlight the essential role of the interaction between the α-amine of glycine and D445 for effective OppA binding and transport.

The OppA–G-SisoK crystal structure revealed no specific interactions with the serine side chain, which is accommodated in a spacious pocket (Fig. 3c). This suggests that OppA binding and uptake rely primarily on recognition of the tripeptide backbone and termini rather than side-chain identity. Accordingly, this mechanism may represent a more general concept that is applicable to a variety of ncAAs presented within a G-XisoK scaffold. We thus expanded our propeptide strategy to efficiently incorporate XisoK derivatives bearing functionalities commonly used in GCE, including moieties for site-specific protein conjugation and crosslinking (Fig. 3d). Supplementing E. coli K12 with G-XisoK derivatives—where X represents various side chains—enabled efficient suppression of sfGFP-N150TAG and Ub-K63TAG using either wt-MbPylRS/PylT or suitable synthetase variants identified from an in cellulo screen (Fig. 3e). Mass spectrometry analysis confirmed site-specific incorporation of the respective XisoK dipeptides (Supplementary Figs. 5 and 6), and supplementation with free XisoK derivatives led to minimal protein expression (Fig. 3e).

Efficient genetic encoding of XisoK derivatives is notable, as lysine aminoacylation (at the ε-amino group) with any of the 20 canonical amino acids is a recently identified reversible PTM[41,42]. Previous attempts at directly encoding SisoK, TisoK, PisoK or CisoK via GCE have proved highly inefficient[23,41,43,44] (Fig. 3e). Our strategy offers a high-efficiency alternative, providing a foundation for functional studies on these PTMs. CisoK-modified proteins are also ideally suited for native chemical ligation approaches[45]. Comparing obtained CisoK-incorporation efficiencies with previous yields using specifically evolved PylRS variants[23] highlights the benefit of actively importing G-CisoK, (Supplementary Fig. 7a), indicating that intracellular ncAA concentration may be more crucial for efficient ncAA incorporation than extensive PylRS engineering.

Site-specific incorporation of XisoK derivatives enables installation of an amino acid with an α-amine moiety, effectively creating a second, artificial N terminus for internal labelling[46]. For example, G-PisoK uptake allows installation of an internal proline bearing a free α-amine and its labelling with phenol derivatives using a chemoenzymatic approach[47]. Tyrosinase oxidizes p-cresol to the corresponding o-quinone, which oxidatively couples to the α-amine of proline in PisoK, enabling specific and quantitative labelling of PisoK-modified POIs (Fig. 3f and Extended Data Fig. 2).

When we screened for PylRS variants for bulky or aromatic X side chains, such as HisoK, no hits emerged using G-HisoK. To probe whether this was due to poor Opp transport or lack of appropriate HisoK-specific PylRS variants, we performed directed evolution using a custom-designed MbPylRS library. A novel MbPylRS variant supported HisoK incorporation with G-HisoK, but not with HisoK (Fig. 3e), indicating that OppA also delivers G-XisoK derivatives with bulky or aromatic X side chains. Efficient encoding of synthetically easily accessible histidine-containing ncAAs may expand the range of metal coordination sites in artificial metalloenzymes[48].

We further broadened the G-XisoK toolbox with non-canonical X side chains for bioorthogonal labelling[6] and crosslinking[7,49]. As a considerable advantage of our propeptide strategy, G-XisoK derivatives can be easily synthesized at large scales via solid-phase peptide synthesis from commercially available building blocks, overcoming synthetic limitations of previous methods. A propargyl-containing derivative (G-PrgisoK) enabled efficient installation of PrgisoK (Fig. 3e and Supplementary Fig. 8) and subsequent fluorophore labelling via Cu(I)-catalysed azide alkyne cycloaddition (CuAAC) on an eGFP-specific nanobody (eGFPNb) (Fig. 3g and Extended Data Fig. 3a). PrgisoK incorporation using G-PrgisoK compares favourably with recently reported efficiencies using a dedicated PrgisoK-PylRS variant[41] (Supplementary Fig. 7b).

To map and trap protein–protein interactions (PPIs), we used our propeptide strategy to efficiently incorporate crosslinkers. Incorporating photoleucine (pL), a commercially available ncAA, as X in the G-XisoK scaffold enabled diazirine encoding into POIs using a Methanomethylophilus alvus PylRS variant (Fig. 3e and Supplementary Fig. 8). UV-induced crosslinking confirmed functionality by capturing PPIs, exemplified by successful crosslinking of glutathione-S-transferase (GST) and sfGFP dimers (Fig. 3h and Extended Data Fig. 3b).

For proximity-based crosslinking, we designed G-ClAisoK to endow POIs with chloroalanine (ClA). Supplementation of K12 with G-ClAisoK led to efficient ClAisoK incorporation (Fig. 3e and Supplementary Fig. 8), enabling $S_N2$-mediated crosslinking with nearby nucleophiles (such as cysteines) in interacting proteins. By pairwise incorporation of ClAisoK and cysteine residues at protein–protein interfaces, we covalently stabilized various low-affinity PPIs (dissociation constant ($K_d$) in the micromolar to low millimolar range), including sfGFP homodimers, affibody–protein Z[50], and Rab1b–DrrA[51] complexes (Fig. 3i and Extended Data Fig. 4). Distances of 8–12 Å between the corresponding Cα atoms could be efficiently crosslinked.

## Scalable XisoK encoding via OppA evolution

All tested G-XisoK tripeptides enabled efficient protein production in chemically defined autoinduction (AI) media[52], but not in nutrient-rich conditions, such as 2-YT medium (Fig. 4a and Supplementary Fig. 9a,b). We hypothesized that short peptides in tryptone and peptone-rich medium may compete with G-XisoK tripeptides for OppA binding. Supporting this, intracellular SisoK levels were sixfold lower in 2-YT medium compared with AI medium after G-SisoK supplementation, indicating impaired OppA-mediated uptake under nutrient-rich conditions (Supplementary Fig. 10). This poses a challenge for scalable, cost-effective use of the G-XisoK toolbox, as AI media are expensive, cumbersome to prepare and lead to lower biomass, diminishing expression yields of ncAA-modified proteins.

To overcome this, we engineered an OppA variant with increased selectivity for G-SisoK over linear tripeptides. We developed a fluorescence-activated cell sorting (FACS)-based platform to screen an error-prone OppA library in ΔoppA cells containing the wt-MbPylRS/PylT pair and sfGFP-N150TAG through successive enrichment in increasing tryptone concentrations. This system couples uptake of the G-SisoK tripeptide to sfGFP fluorescence (Fig. 4b). Four converging OppA variants with four or five mutations distributed all over the OppA-fold were identified. Mutational hotspots were targeted for saturation mutagenesis and the obtained library was subjected to multiple FACS-based enrichment steps in peptide-rich medium, yielding the final OppA variant (OppA-iso). OppA-iso contains seven mutations, with only R439Q occurring near the binding site (Extended Data Fig. 5a). Genomic integration of OppA-iso into the K12 genome via lambda red-mediated homologous recombination created the IsoK12 strain.

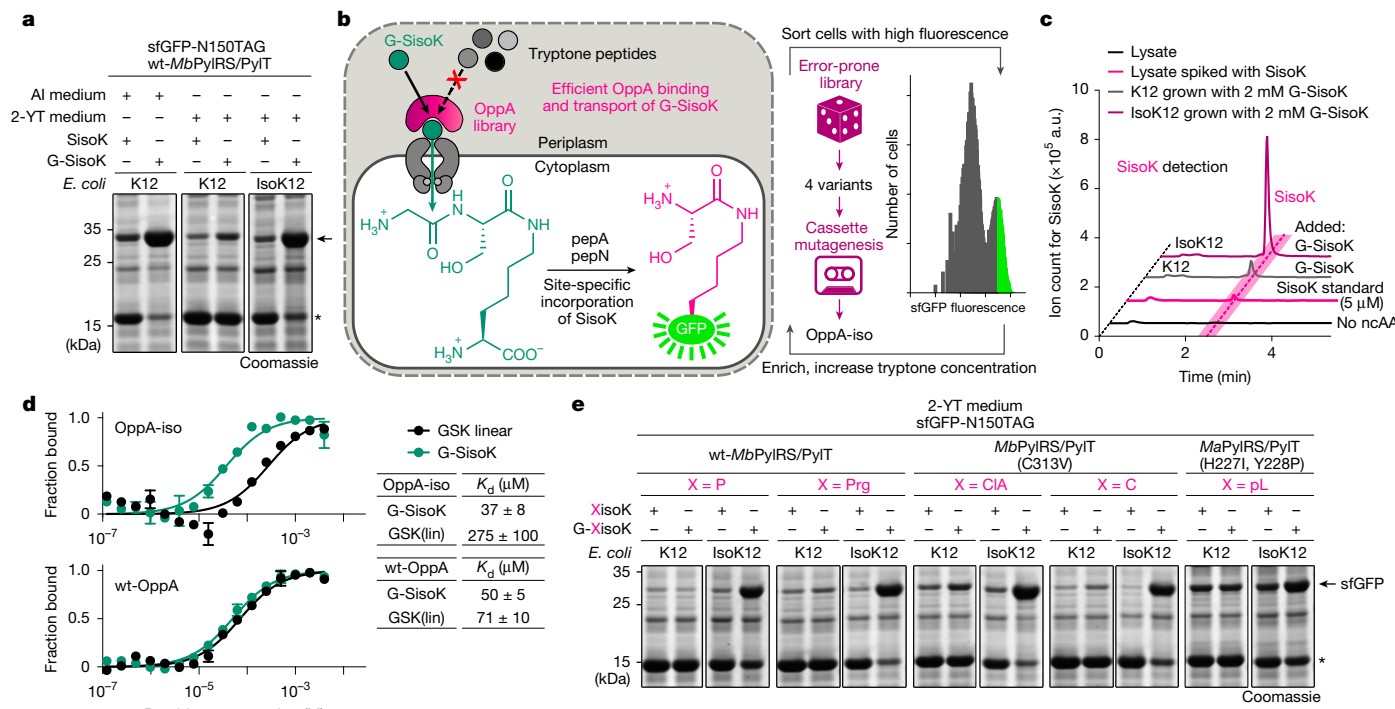

**Fig. 4 | Scalable XisoK incorporation through OppA evolution. a**, SDS–PAGE analysis of sfGFP-N150TAG expression in the presence of SisoK or G-SisoK in AI or 2-YT medium. In wild-type K12, full-length sfGFP expression yields are significantly reduced when using tryptone-containing 2-YT medium owing to competition between tryptic peptides and G-SisoK for OppA binding. sfGFP expression in 2-YT medium is recovered when using the engineered IsoK12 strain. Consistent results were obtained over three independent replicate experiments. Arrow indicates sfGFP, asterisk indicates truncated sfGFP. **b**, Scheme for OppA evolution to improve G-SisoK uptake in tryptone-containing medium. OppA libraries were screened for increased G-SisoK uptake under increasing tryptone concentrations by monitoring amber suppression of sfGFP-N150TAG and sorting fluorescent cells. Initial screening of an error-prone library yielded four variants, which were used as basis for creating a cassette mutagenesis library. Screening of this library identified OppA-iso. **c**, Extracted ion chromatograms to determine intracellular SisoK concentrations.

IsoK12 cells grown in 2-YT medium show 7- to 10-fold higher intracellular SisoK concentrations compared with K12 cells when adding 2 mM G-SisoK to 2-YT medium. Consistent results were obtained over three independent replicate experiments. **d**, Affinity measurements of G-SisoK and a linear GSK peptide (GSK(lin)) towards wild-type OppA (wt-OppA) and OppA-iso using microscale thermophoresis. Data are mean $K_d$ values ± s.e.m. calculated from three biologically independent experiments ($n = 3$). All data processing was performed using GraphPad Prism 10 (GraphPad software) and MO.affinity Analysis (v.3.0.5, NanoTemper Technologies). **e**, SDS–PAGE analysis of sfGFP-N150TAG expression in wild-type K12 versus IsoK12 grown in 2-YT medium in the presence of G-XisoK derivatives with X = P, Prg, ClA, C or pL. Full-length sfGFP expression is greatly increased in IsoK12 for all G-XisoK derivatives. Consistent results were obtained over three independent experiments. Asterisk indicates truncated sfGFP. Full gels and gels for other G-XisoK derivatives are presented in Supplementary Fig. 9.

IsoK12 and parental K12 showed similar doubling times in AI and 2-YT media (Supplementary Fig. 11a). In 2-YT medium, IsoK12 exhibited sevenfold to tenfold higher intracellular SisoK concentrations compared with K12 when supplemented with G-SisoK, whereas BocK levels remained unchanged, confirming enhanced G-SisoK transport with OppA-iso (Fig. 4c and Supplementary Fig. 10). Notably, IsoK12 restored amber suppression efficiency in 2-YT medium to levels observed for K12 in AI medium (Fig. 4a and Supplementary Fig. 9).

Analysis using microscale thermophoresis showed that wild-type OppA and OppA-iso exhibited similarly low binding affinities towards SisoK (around 300 μM), whereas OppA-iso exhibited a slightly improved affinity (37 μM) towards G-SisoK, compared with wild-type OppA (50 μM). Notably, binding of a linear GSK tripeptide (mimicking linear tryptone peptides) was fourfold lower for OppA-iso (275 μM) compared with wild-type OppA (71 μM), validating its altered selectivity (Fig. 4d and Supplementary Fig. 11b).

Structural analysis showed that most mutated residues in OppA-iso do not directly contact G-SisoK. One notable exception is R439, which lies within hydrogen-bonding distance of the C terminus of the ligand and is replaced by glutamine in OppA-iso. Since the R439A mutant still supports efficient G-SisoK uptake (Supplementary Fig. 4d), binding is likely to be maintained through compensatory interactions with H397 and N392, which are well positioned to interact with the lysine

carboxylate of G-SisoK. By contrast, linear tripeptides such as GSK are dependent on R439 for binding, so its mutation reduces affinity, consistent with structural and binding data (Extended Data Fig. 5b).

Notably, the uptake benefit of OppA-iso extended to other G-XisoK derivatives. IsoK12 grown in 2-YT medium achieved efficient amber suppression for all tested X residues (A, S, T, C, V, L, P, H, Prg, pL and ClA) with minimal truncation and yields matching those obtained in AI medium (Fig. 4e and Supplementary Fig. 9). In fact, tripeptide uptake was so efficient in IsoK12 that G-SisoK concentrations as low as 50–100 μM matched incorporation levels seen in K12 with 1 mM G-SisoK, reducing required ncAA concentrations by a factor of around 10 (Extended Data Fig. 6a).

The G-XisoK/IsoK12 system enabled high-yield XisoK incorporation at various positions in a wide range of target proteins (PCNA, β-lactamase, SUMO2, calmodulin, eGFPNb, interleukin-2, human growth hormone, RanGAP and Hsp82) ranging in size from 7 to 85 kDa, including therapeutically relevant examples. Suppression efficiencies surpassed those with BocK and matched wild-type levels (Extended Data Fig. 6b and Supplementary Figs. 12–14). Preparative large-scale production of eGFPNb bearing PrgisoK resulted in similar purified protein yields (44 mg l⁻¹) as obtained for wild-type expression (41 mg l⁻¹), exceeding yields from a previously optimized alkyne-ncAA/PylRS combination (Extended Data Fig. 6c).

Increasing intracellular ncAA concentrations via efficient tripeptide uptake also facilitated multi-site amber suppression. We introduced up to three TAG codons into histone H3 (K27, K79 and K122) and expressed the corresponding variants with G-SisoK. IsoK12 outperformed K12 in single, double and triple suppression, achieving wild-type-like expression in the first two cases and significant H3 yields even for triple suppression. By contrast, BocK produced only trace amounts of doubly and triply suppressed variants (Extended Data Fig. 6d and Supplementary Fig. 15).

## Generalized ncAA uptake using Z-AisoKs

Efficient uptake and processing of G-XisoK results in high intracellular concentrations of both the XisoK dipeptide and cleaved N-terminal glycine. Crystallographic analysis of the OppA–G-SisoK complex revealed a spacious cavity extending from the serine side chain to the N-terminal glycine (Extended Data Fig. 7a). Given the promiscuity of OppA, we hypothesized that other side chains, including those of ncAAs, could also be accommodated at this position, enabling efficient transport of diverse non-canonical tripeptides into the *E. coli* cytosol. If different N-terminal residues (Z) in a Z-AisoK scaffold (Fig. 5a) are also efficiently cleaved, the strategy could broadly enable intracellular delivery of various amino acids.

We synthesized a panel of 14 Z-AisoK tripeptides bearing either natural amino acids or ncAAs with diverse side chains, including bulky or negatively charged groups with poor or negligible cell permeability as N-terminal Z residues (Fig. 5a). To monitor uptake and cleavage of Z, we assessed AisoK incorporation into sfGFP-N150TAG using the wt-*Mb*PylRS/tRNA pair. For 8 out of 14 Z residues, supplementation of K12 with corresponding Z-AisoK tripeptides, yielded amber suppression efficiencies resembling those obtained with G-AisoK (Fig. 5b and Extended Data Fig. 7b), with LC–MS confirming AisoK incorporation in all cases (Supplementary Fig. 16a). No sfGFP expression was observed in Δ*opp*A cells, confirming dependence on OppA-mediated uptake (Extended Data Fig. 7b).

By contrast, tripeptides bearing bulkier or negatively charged Z side chains (compounds **5**, **6** and **12**–**15**; Fig. 5a), resulted in reduced (**5** and **6**) or completely abolished (**12**–**15**) amber suppression (Fig. 5b and Extended Data Fig. 7b), suggesting inefficient uptake and/or cleavage. To address this, we leveraged our OppA engineering platform to evolve new OppA variants capable of accommodating tripeptides with larger or negatively charged Z residues (Fig. 5c). Guided by our OppA–G-SisoK structure, we selected four residues around glycine of G-SisoK for site-saturation mutagenesis (Fig. 5d and Extended Data Fig. 7a) and subjected a corresponding library to multiple FACS-based enrichments in the presence of tripeptides **13** and **15**. From these screens, we isolated two OppA variants with enhanced uptake: OppA-Z1 (evolved with **13**) and OppA-Z2 (evolved with **15**). Both variants feature small side chains at the targeted positions, which are likely to increase the size of the binding pocket to accommodate larger substrates, and in the case of OppA-Z2, a non-programmed R439H mutation. These findings provide further evidence that the R439 variant can enhance binding of isopeptide-linked scaffolds.

We genomically introduced these OppA variants, generating K12-Z1 and K12-Z2 strains, respectively. Supplementing K12-Z1 with bulky Z tripeptides (**5**, **6**, **12** and **13**) enabled efficient AisoK incorporation, but uptake of tripeptides with negatively charged Z residues (**14** and **15**) was not supported by this engineered strain (Extended Data Fig. 7c). By contrast, K12-Z2 enabled efficient uptake of all 14 Z-AisoK tripeptides, including those bearing negatively charged ncAAs such as SucK and GluK as Z residues (Fig. 5b, Extended Data Fig. 7c and Supplementary Fig. 16b).

To demonstrate that tripeptide-based uptake is superior to direct ncAA supplementation, we focused on selected Z residues that are known to have low suppression efficiencies, for which limited uptake was suspected to be the main bottleneck. We compared amber suppression yields after supplementing cells either with Z directly or with the Z-AisoK tripeptide. Compound **3** (with acetyl-lysine (AcK) as the Z

residue) yielded higher AcK incorporation than AcK alone using the *Mb*PylRS variant AcKRS3[53] (Extended Data Fig. 8a), highlighting the benefit of active delivery. Notably, compound **13** (bearing LipK as Z residue) or LipK alone led to negligible expression of full-length sfGFP in the presence of the LipKRS/tRNA pair[54]. By contrast, K12-Z1 supplemented with **13** enabled efficient sfGFP production, with LipK incorporation confirmed by LC–MS (Fig. 5e). Similar improvements were observed for tripeptides **5** and **6** in K12-Z2, confirming that evolved OppA variants enable effective delivery of tripeptides that are impermeable in wild-type strains (Extended Data Fig. 8b).

## Efficient encoding of two distinct ncAAs

Given the adaptability of OppA in substrate recognition, we envisioned a broadly applicable strategy to co-deliver two ncAAs using a single, easily synthesized Z-XisoK tripeptide. To leverage such a mechanism for site-specific dual ncAA incorporation into a single protein, we designed tripeptide **16**, bearing AcK (a PTM) as Z and pLisoK (a photocrosslinker) as XisoK (Fig. 5f). Such a setup would represent an ideal tool to investigate PTM-specific protein interactors or to chemically stabilize transient POI–reader complexes[55].

Notably, the PylRS/PylT pairs for AcK and pLisoK are mutually orthogonal[56] (Extended Data Fig. 8c). After OppA-mediated uptake and cleavage of **16**, both AcK and pLisoK accumulate intracellularly, allowing dual suppression of TAA and TAG codons within the same target protein (sfGFP-N40AcK-N150pLisoK; Fig. 5f), using the respective PylRS/PylT pairs. Notably, protein yields were significantly higher when cells were supplemented with tripeptide **16** compared with addition of free AcK and G-pLisoK, underscoring the enhanced efficiency of transporter-mediated delivery. LC–MS of full-length sfGFP confirmed dual incorporation of AcK and pLisoK (Fig. 5f). Mutual orthogonality and accurate decoding of both ncAAs was furthermore shown using a Ub–SUMO2 fusion construct containing a Tobacco Etch Virus (TEV) protease site (Ub-K48TAG-TEV-SUMO-K11TAA). LC–MS analysis after TEV cleavage revealed specific pLisoK incorporation at TAG48 of Ub and AcK encoding in response to TAA11 of SUMO2 (Extended Data Fig. 8d).

## Discussion

Low protein yields and the limited chemical accessibility of many ncAAs remain major obstacles to the routine application of GCE for the generation of proteins with therapeutic or biotechnological potential. Here we overcome these limitations by combining a modular propeptide strategy with the programmed 'hijacking' of the Opp transporter, enabling active and tailored import of a broad range of ncAAs, including those that have historically been refractory to efficient uptake and encoding. Isopeptide-linked Z-XisoK tripeptides act as Trojan horses that can be readily synthesized from commercially available building blocks via solid-phase peptide synthesis, and function as privileged ligands for the periplasmic binding protein OppA, enabling their ATP-driven transport into the cytosol. Once inside the cell, Z-XisoK tripeptides are enzymatically processed, leading to intracellular accumulation of isopeptide-linked XisoK ncAAs and Z residues. Using a FACS-based directed evolution approach, we reprogrammed OppA to selectively discriminate against linear peptides that are present in complex media. The resulting engineered *E. coli* strain, IsoK12, enables cost-effective, high-yield production of modified proteins in nutrient-rich conditions using minimal tripeptide concentrations. This platform allows robust incorporation of 11 previously inaccessible XisoK ncAAs, expanding the chemical space available through GCE. Among these functionalities are ncAAs bearing bioorthogonal handles, novel PTMs, chemical and photocrosslinkers, and functional groups for chemoenzymatic ligations, all of which we demonstrated in proof-of-principle applications. The positively charged side chains of XisoK ncAAs make them ideal moieties for applications such as protein labelling, for which site-directed incorporation

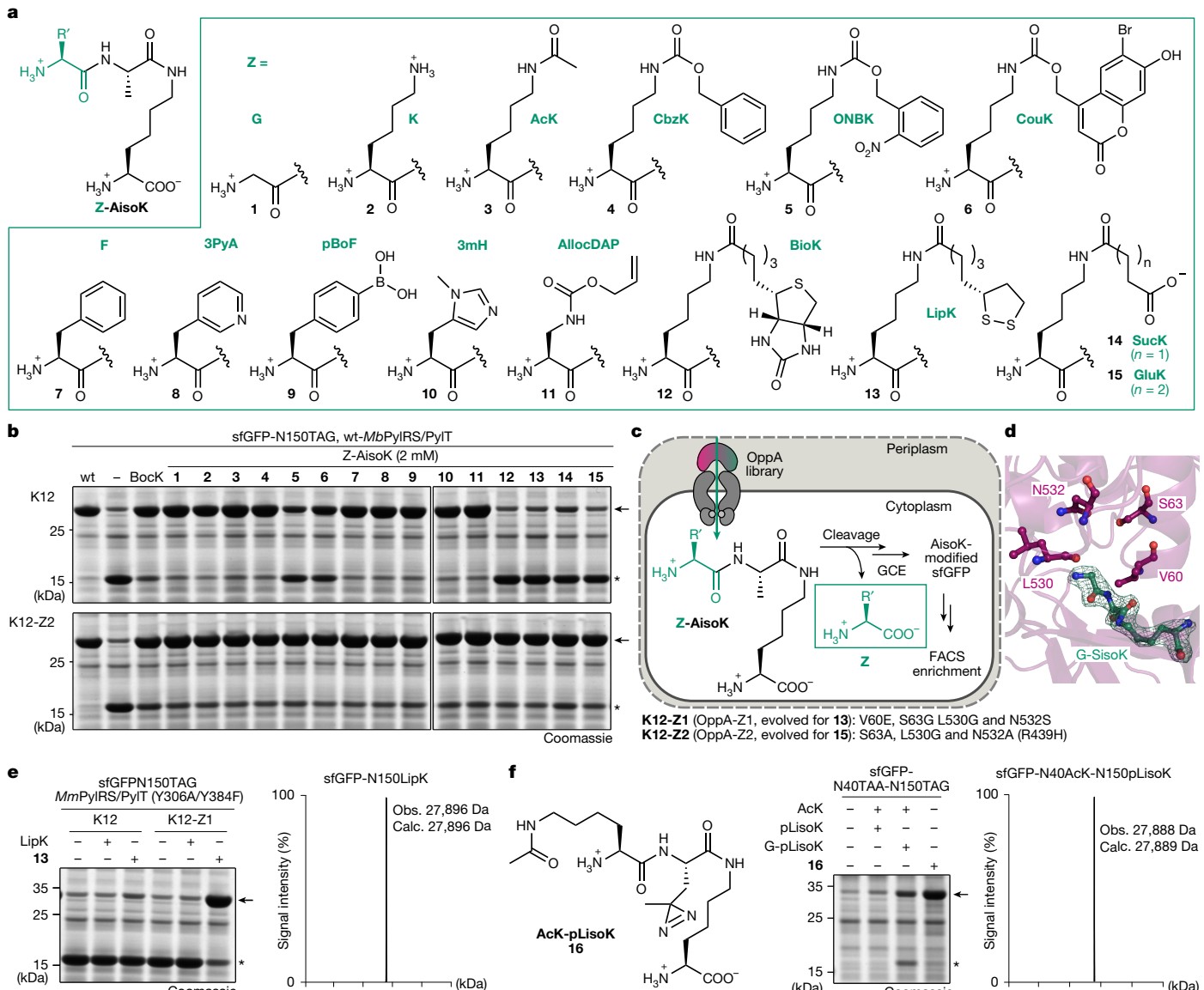

**Fig. 5 | Generalized ncAA uptake using Z-AisoK tripeptides. a**, Structure of tested Z residues within the Z-AisoK scaffold. **b**, SDS–PAGE analysis of sfGFP-N150TAG expression in the presence of 2 mM of BocK or Z-AisoK tripeptides with wt-*Mb*PylRS/PylT. Expression levels of full-length sfGFP indicate successful transport of the Z-AisoK tripeptides, subsequent cleavage to Z and AisoK and AisoK incorporation. Top, expression in wild-type K12. Bottom, expression in the evolved strain K12-Z2. Consistent results were obtained over three independent replicate experiments. Arrow indicates full-length sfGFP, asterisk indicates truncated sfGFP. **c**, Scheme for OppA evolution to accommodate novel Z-AisoK substrates. Successful transport by OppA library variants was evaluated by incorporation of AisoK into sfGFP-N150TAG. Variants with high sfGFP fluorescence were enriched via three rounds of FACS. **d**, X-ray structure of the OppA–G-SisoK complex, highlighting four residues surrounding the N-terminal glycine of G-SisoK that were targeted for site-saturation mutagenesis to enable recognition of novel Z-AisoK substrates. **e**, Left, SDS–PAGE analysis of sfGFPN150TAG expression with 0.25 mM LipK or the corresponding Z-AisoK tripeptide **13** with a LipK-specific *Methanosarcina mazei* PylRS/PylT (*Mm*PylRS/PylT) variant (Y306A/Y384F). Expression levels are highest with peptide **13** in the K12-Z1 strain, which expresses OppA-Z1, evolved specifically for the transport of **13**. Right, LC–MS analysis of sfGFP purified from K12-Z1 cultures grown with **13**. Observed mass confirms incorporation of LipK. Consistent results were obtained over three independent replicate experiments. **f**, Dual stop codon suppression using a single isopeptide-linked tripeptide. Left, chemical structure of the tripeptide AcK-pLisoK (**16**). Middle, SDS–PAGE analysis of sfGFP-N40TAA-N150TAG expression in the presence of either AcK and G-pLisoK (or pLisoK) separately added to medium or in the presence of tripeptide **16** in IsoK12. Right, LC–MS analysis of purified sfGFP confirms dual ncAA incorporation (AcK and pLisoK) after the addition of tripeptide **16**. Consistent results were obtained over three independent replicate experiments.

at surface-exposed positions is essential. This strategy avoids aggregation and misfolding that are often associated with bulky, hydrophobic ncAAs commonly used for labelling purposes[6]. With these advantages, we anticipate that the XisoK toolbox can be further expanded through aaRS engineering to include functionalities for inverse-electron-demand Diels–Alder cycloadditions or spectroscopic probes.

Notably, uptake and processing of Z-XisoK tripeptides can also be leveraged for customized delivery and intracellular accumulation of challenging Z residues that typically lack cell permeability. By synthesizing a panel of 14 different Z-AisoK tripeptides, we demonstrate that AisoK incorporation into sfGFP serves as a straightforward readout for efficient tripeptide uptake and processing, thereby eliminating the need for intensive mass spectrometry-based uptake assays[21]. Our platform provides a foundation for developing OppA variants with specific binding sites for tripeptides carrying otherwise cell-impermeable Z residues, such as bulky and negatively charged groups. Thus, our

innovation enables the intracellular delivery of these challenging ncAAs. Site-specific incorporation of Z residues from Z-XisoK significantly outperforms direct Z supplementation. As OppA evolution for a specific Z residue is decoupled from availability of a Z-specific orthogonal aaRS, our system presents a practical and modular strategy for dissecting ncAA uptake from its co-translational incorporation, complementing recent efforts in decoupling and optimizing individual GCE steps towards synthesis of non-canonical biopolymers in *E. coli*[57,58]. The current system relies on orthogonality of Z incorporation to AisoK-selective PylRS variants, but future efforts will focus on relaxing this constraint by diversifying the XisoK scaffold to further expand the versatility and applicability of the system. Notably, the ability to co-import two distinct ncAAs via a single Z-XisoK scaffold enables efficient dual ncAA incorporation. We envision extending our concept towards multi-ncAA encoding in synergy with advances in non-canonical polymer biosynthesis[57,58] and *E. coli* strains with compressed genomes[18–20].

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

## Methods

### Expression of ncAA bearing proteins

Chemically competent *E. coli* K12 cells were co-transformed with pBAD_POI (encoding the POI with a C-terminal H6 tag) and pEVOL_PylRS (encoding two copies of the PylRS variant and tRNA$_{CUA}$). After recovery in 1 ml SOC medium for 1 h at 37 °C, cells were cultured in 5 ml of 2-YT medium supplemented with ampicillin (100 µg ml$^{-1}$) and chloramphenicol (50 µg ml$^{-1}$) and grown overnight at 37 °C. The overnight culture was then diluted to an OD$_{600}$ of 0.05 in either 2-YT or AI media[52] supplemented with ampicillin (100 µg ml$^{-1}$), chloramphenicol (50 µg ml$^{-1}$) and ncAA. Cells grown in 2-YT medium were grown to an OD$_{600}$ of 0.6 and induced with 0.05% L-arabinose and grown overnight at 37 °C. Cells grown in AI media[52] were directly grown overnight at 37 °C. Cells of the overnight culture were collected by centrifugation at 4,000*g* for 10 min at 4 °C and the pellets were analysed by SDS–PAGE or stored at −20 °C for further use.

For the incorporation of XisoK ncAAs into proteins for purification and downstream characterization, the corresponding G-XisoK peptides were used during expression. The dipeptide XisoK was only used for comparing expression levels in whole lysate. For the incorporation of Z ncAAs, both the free amino acid Z and the corresponding Z-AisoK tripeptide were used. Proteins expressed in the presence of Z-AisoK tripeptides were purified and analysed via LC–MS to confirm Z or AisoK incorporation. For dual incorporation, peptide **16** was used for purification experiments.

### Purification of His$_6$-tagged proteins

Cell pellets were resuspended in lysis buffer (20 mM Tris-HCl pH 8.0, 300 mM NaCl, 30 mM imidazole, 1 mM PMSF) and the cells were lysed via sonication in an ice water bath. Lysed cells were centrifuged at 14,000*g* for 20 min at 4 °C and cleared lysate was incubated with Ni Sepharose fast flow beads (1 ml slurry per 1 l culture, Cytiva) pre-equilibrated with Ni-NTA wash buffer (20 mM Tris-HCl pH 8.0, 300 mM NaCl, 30 mM imidazole) and incubated on a roller for 1 h at 4 °C. Beads were then washed with 10 column volumes of wash buffer 3 times and protein was eluted with elution buffer (20 mM Tris-HCl pH 8.0, 300 mM NaCl, 300 mM imidazole). Resulting protein was either directly used for mass determination or further purification was performed via gel filtration using a Superdex 75 increase 10/300 GL column (Cytiva) equilibrated with SE buffer (2× PBS pH 7.0 for eGFPNb or 20 mM potassium phosphate buffer pH 6.5, 100 mM NaCl for PisoK-bearing proteins or 1× PBS pH 7.4 for Affibody and ProteinZ). Fractions containing protein were pooled and concentrated using Amicon centrifugal filter units with the appropriate molecular weight cut-off (MWCO). Protein was then flash frozen in liquid nitrogen and stored at −80 °C till further use. Proteins bearing CisoK were treated with methoxyamine (10 µM protein, 100 mM methoxyamine at pH 4.0 ammonium acetate) and buffer exchanged in 1× PBS before storage.

### Generation of an OppA error-prone library

An error-prone library of *opp*A was generated using an error-prone PCR kit (Jena Bioscience PP-102). Primers OppA_EP_fwd and OppA_EP_rev (Supplementary Table 2) were used to amplify *oppA* from pEVOL_MbPylRS_oppA according to the manufacturer's instructions, running for 25 cycles and adding 2.5 µl error-prone solution. Resultant amplicon was purified on a 1% agarose gel and digested with NdeI and PstI-HF. Digested insert was then ligated with T4 ligase into a backbone generated by digesting pEVOL_MbPylRS_oppA with NdeI and PstI-HF, followed by dephosphorylation with Antarctic Phosphatase. Electrocompetent DH10β cells were then transformed with the purified ligation. After recovery in SOC medium for 1 h at 37 °C, cells were added to 50 ml of 2-YT medium supplemented with chloramphenicol (50 µg ml$^{-1}$) and grown overnight at 37 °C. Library plasmid from the overnight culture was purified and used for subsequent steps. The size

of the library was determined by plating the freshly transformed cells in a dilution series on LB agar plates supplemented with chloramphenicol (50 µg ml$^{-1}$). The size was calculated to be $4 \times 10^6$. Multiple single clones were sequenced to determine the average error rate and was optimized to be ~0.8% or an average of 5 amino acid mutations per gene.

### Generation of OppA site-saturation libraries

For the tryptone resistance site-saturation library, four hotspot positions in OppA were selected for site-saturation, on the basis of sequenced variants from selection of the error-prone library. Positions were chosen if mutations occurred more than once (R439 and S460) or if mutations occurred close to each other in different variants (D221 and W222).

For the site-saturation library targeting the pocket of the Z position, four positions were selected on the basis of the X-ray crystal structure of wt-OppA–G-SisoK around the N-terminal glycine of G-SisoK (V60, S63, L530 and N532) to accommodate new N-terminal amino acids.

Sites were randomized using degenerate trimer primers (Ella Biotech) or NNK primers. For the tryptone resistance library, a mix of templates based on pEVOL_MbPylRS_oppA containing wild-type OppA and four variants from the error-prone library were amplified with library primers (Supplementary Table 2). For the Z position library, pEVOL_MbPylRS_oppA_wt was used as a template. Both tryptone resistance template mix and Z library wild-type OppA template were amplified with their respective library primers (Supplementary Table 2). The linear amplicon was then purified on a 1% agarose gel and digested with BbsI-HF and DpnI. Digested fragments were circularized via ligation with T4 ligase and electrocompetent DH10β cells were transformed with the circularized library plasmid. After recovery in SOC medium for 1 h at 37 °C, cells were added to 50 ml of 2-YT medium supplemented with chloramphenicol (50 µg ml$^{-1}$) and grown overnight at 37 °C. Library plasmid was purified from the overnight culture and used as the template for the next round of amplification to randomize the next position(s).

For the tryptone resistance library, this process was repeated 3 times, each time randomizing a different position, to give a library with positions D221, W222, R439 and S460 mutated to all 20 amino acids on wild-type OppA and 4 error-prone variants. The library size was $8 \times 10^5$. For the Z library the process was repeated 2 times to cover all 4 positions (V60, S63, L530 and N532). The library size was $1 \times 10^6$.

### FACS-based screening protocol

Electrocompetent *E. coli* K12 Δ*opp*A cells containing sfGFP reporter plasmid pBAD_sfGFP_N150TAG_H6 were transformed with the OppA library (pEVOL_MbPylRS_oppA_EP_lib, pEVOL_MbPylRS_oppA_trimer_lib, pEVOL_MbPylRS_oppA_Z_lib or pEVOL_MbPylRS_oppA_wt). After recovery in 1 ml SOC for 1 h at 37 °C, transformed cells were diluted in 50 ml of non-inducing medium (AI medium[19] without arabinose) supplemented with ampicillin (100 µg ml$^{-1}$) and chloramphenicol (50 µg ml$^{-1}$) and grown overnight. The overnight culture was then diluted in non-inducing medium and grown to an OD$_{600}$ of 0.6.

At this point, 0.05% arabinose was added to induce sfGFP expression and the culture was split into smaller cultures. For the tryptone resistance screening campaign, cultures were supplemented with or without 0.5 mM G-SisoK and varying amounts of tryptone to apply a selection pressure towards OppA variants which preferably bound to G-SisoK.

For the screening of the Z library, cultures were supplemented with 2 mM of each Z-AisoK tripeptide of interest (**13** or **15**). Cultures were then grown for 4 h at 37 °C to allow for sfGFP expression. To halt growth, the cultures were cooled on ice for 10 min, centrifuged (4,000*g*, 5 min, 4 °C) and resuspended in ice cold PBS pH 7.0. The PBS cell suspension was sorted on a Sony cell sorter (SH800) using a 70-µm chip sorting for cells with highest sfGFP fluorescence intensity. Gating was decided on the basis of the positive control (tryptone resistance screening: *wt-oppA*, 0.5 mM G-SisoK, 0 g l$^{-1}$ tryptone, Z library screening: *wt-oppA*,

2 mM G-AisoK, 0 g l$^{-1}$ tryptone) and the negative controls (*wt-oppA*, 0.5 mM G-SisoK, *X* g l$^{-1}$ tryptone, with X being the tryptone concentration used in that round of enrichment, and *wt-oppA*, 0 mM G-SisoK, 0 g l$^{-1}$ tryptone). For each round, cells with the top 0.5–2% sfGFP fluorescence were sorted. Sorted cells were recovered in SOC medium supplemented with ampicillin (50 µg ml$^{-1}$) and chloramphenicol (25 µg ml$^{-1}$) overnight at 37 °C and the process was repeated for further enrichment. After multiple rounds of enrichment, cells were sorted into a 96-well plate containing SOC supplemented with ampicillin (50 µg ml$^{-1}$) and chloramphenicol (25 µg ml$^{-1}$) and grown overnight at 37 °C. Cultures grown from single cells were further evaluated via the fluorescence plate reader assay and variants which showed the desired phenotype were sent for Sanger sequencing. The error-prone library was subjected to 5 rounds of enrichment, each time doubling the tryptone concentration from 1 g l$^{-1}$ (round 1) to 16 g l$^{-1}$ (round 5). The site-saturation library was directly grown in LB medium (10 g l$^{-1}$ tryptone) for 2 rounds of enrichment and 2-YT medium (16 g l$^{-1}$) for 3 rounds of enrichment. The Z library was subjected to 3 rounds of enrichment at 2 mM of Z-AisoK tripeptide **13** or **15**.

### Preparation of *E. coli* lysates for LC–MS based uptake assays

The uptake assay protocol was adapted from previously published protocols[21,36]. Relevant *E. coli* strains were transformed with a pBAD plasmid to prevent contamination of cultures. After recovery in 1 ml SOC for 1 h at 37 °C, cells were cultured in 5 ml of 2-YT or AI medium supplemented with ampicillin (100 µg ml$^{-1}$) and grown overnight at 37 °C. Overnight cultures were diluted to an OD$_{600}$ of 0.05 in 5 ml of 2-YT or AI medium supplemented with ampicillin (100 µg ml$^{-1}$) and ncAA or peptide and grown overnight at 37 °C. The OD$_{600}$ of the overnight cultures was determined and 12 OD ml were collected by centrifugation at 4,000*g* for 10 min. Cell pellets were then washed 3 times with 1 ml of cold medium and resuspended in 400 µl of a methanol:water solution (60:40). Cells were lysed via 5 freeze thaw cycles in liquid nitrogen and a 42 °C water bath. Lysate was cleared by centrifugation at 17,900*g* for 20 min. Five-hundred microlitres of cleared lysate was passed through Amicon centrifugal filter units (Millipore, 3 kDa MWCO) and the flow through was injected onto the LC–MS for analysis. For cultures incubated with BocK, samples were injected onto a Zorbax SB-C18 (Agilent, 4.6 × 150 mm) column and a gradient of 5–95% was used. For cultures incubated with XisoK and G-XisoK peptides, samples were injected on a Poroshell 120 HILIC-Z (Agilent, 2.1 × 100 mm) column using a gradient of 95–10%. The mass spectrometer was set to single ion mode to detect the relevant *m/z* for each ncAA or peptide.

To determine intracellular concentrations, calibration points of lysate spiked with known ncAA or peptide concentrations were measured. Ion peaks were integrated, and integral values plotted against concentration to determine a linear calibration line. Integral values of unknown samples were interpolated on calibration line to determine lysate concentrations. Intracellular concentrations were estimated assuming 1 OD$_{600}$ = 8 × 10$^8$ cells per ml and the volume of an *E. coli* cell (0.6 fl).

### Determination of $K_d$ using microscale thermophoresis

Microscale thermophoresis was performed on the NanoTemper Monolith NT.115 (NanoTemper Technologies). Wild-type OppA and evolved variants were fluorescently labelled using the Monolith Protein Labeling Kit RED-NHS 2nd Generation (NanoTemper Technologies) and diluted to 100 nM in assay buffer (2× PBS pH 7.0, 0.02% Tween-20). Peptides were diluted to double the highest measured concentration in assay buffer and diluted twofold in a dilution series to give 16 peptide concentrations. Equal volumes of protein and peptide solutions were mixed (final protein concentration of 50 nM) and incubated at room temperature for 30 min. Samples were loaded into capillaries (Monolith NT.115 Capillaries, NanoTemper Technologies) and measured according to manufacturer's instructions. Three independent

replicates were measured for each peptide–protein combination and data analysis was performed with MO.affinity Analysis (v.3.0.5, NanoTemper Technologies).

### Generating isoK12 strain via homologous recombination

Primers with 50 bp overhangs homologous to regions upstream and downstream of the OppA locus in *E. coli* genome were used to amplify *oppA-iso* from pEVOL_MbPylRS_oppA-iso. (Supplementary Table 2).

A single clone of *E. coli* K12 Δ*oppA* cells transformed with a pSIJ8 plasmid (Supplementary Table 2) was cultured in 2-YT medium supplemented with ampicillin (50 µg ml$^{-1}$) and grown at 30 °C, 200 rpm until an OD$_{600}$ of 0.3 followed by induction with 15 mM arabinose for the expression of lambda red recombineering genes. After incubation for 45 min at 37 °C the culture was cooled on ice to halt growth and made electrocompetent. The resulting electrocompetent *E. coli* K12 Δ*oppA* cells with expressed recombineering genes were then transformed with the linear DNA fragment encoding oppA-iso. After recovery in 1 ml SOC for 2 h at 37 °C, cells were diluted in 2-YT medium and grown overnight at 37 °C for the curing of thermosensitive plasmid pSIJ8. The overnight culture (containing a mix of Δ*opp*A and knock-in cells) was diluted and grown to an OD$_{600}$ of 0.6 and made electrocompetent. These cells were then co-transformed with aaRS plasmid pEVOL_MbPylRS and sfGFP reporter pBAD_sfGFP_N150TAG_H6. Transformed cells were recovered in 1 ml SOC for 1 h at 37 °C and diluted in 2-YT medium supplemented with ampicillin (100 µg ml$^{-1}$) and chloramphenicol (50 µg ml$^{-1}$) and grown to an OD$_{600}$ of 0.6 and sfGFP expression was induced with 0.05% arabinose and 0.5 mM G-SisoK was added. The culture was grown for 4 h, cooled on ice, centrifuged (4,000*g*, 5 min, 4 °C), and resuspended in ice cold PBS pH 7.0. Cells with the highest sfGFP fluorescence were sorted as single cells into a 96-well plate containing 2-YT medium and grown overnight. Clones with the correct genomic insert were confirmed with sequencing of the genomic locus and whole-genome sequencing. Once confirmed, plasmids were cured from the strain via electroporation.

### Generating K12-Z1 and K12-Z2 strains via CRISPR-mediated genome editing

Knock-in generation was adapted from a previously published protocol[39]. In brief, a single clone of *E. coli* K12 Δ*oppA* cells transformed with pSIMcpf1 was cultured in 2-YT medium supplemented with hygromycin (150 µg ml$^{-1}$) and grown at 30 °C and 200 rpm until an OD$_{600}$ of 0.2 was reached. At this point, expression of lambda red recombineering genes was induced by incubation at 42 °C for 15 min. Cultures were then cooled on ice for 20 min to halt the growth and cells were made electrocompetent. The resulting electrocompetent cells were transformed with pTF_oppA-Z1 or pTF_oppA-Z2, which carried a donor DNA with genes for the respective OppA variant along with 50 bp upstream and downstream homologous regions of the *oppA* locus, as well as a CRISPR array encoding a guide RNA (gRNA) targeting the FRT site present in the knocked out *oppA* locus (Supplementary Table 7). After electroporation, cells were rescued in SOC medium for 1 h, plated on LB agar with hygromycin (150 µg ml$^{-1}$) and spectinomycin (120 µg ml$^{-1}$) and incubated overnight at 30 °C. Successfully knocked-in clones were confirmed via colony PCR and grown in 2-YT medium with 0.05% arabinose for 5 h at 30 °C and then grown overnight at 37 °C to cure the cells of plasmids. To further confirm successful integration, cells were sent for whole-genome sequencing.

### Generating peptidase double knockouts via CRISPR-mediated genome editing

Peptidase knockouts were generated analogously to the K12-Z1 and K12-Z2 strains. *E. coli* K12 Δ*pepN* cells from the Keio collection[38] were transformed with pSIMcpf1 (Supplementary Table 2) and were prepared as previously described. pTF plasmids (Supplementary Table 1), which carry a donor DNA of 50 bp upstream and downstream of the

peptidase genomic locus, as well as a CRISPR array encoding two gRNAs that target the corresponding peptidase, were used (Supplementary Table 7). Colonies were confirmed via colony PCR and whole-genome sequencing. Plasmids were cured as previously described.

## Dual stop codon suppression for incorporation of AcK and pLisoK into proteins

Chemically competent *E. coli* K12 cells were co-transformed with pBAD_POI (either pBAD_sfGFP_N40TAA_N150TAG_H6 or pBAD_Ub_K48TAA_TEV_SUMO2_K11TAG_H6) with a C-terminal His$_6$ tag) and pEVOL_AcKRS3(TAA)_RBS_MaPylRS_IP(TAG) (encoding AcKRS3 and MaPylRS_IP polycistronically and their respective tRNAs). After recovery in 1 ml SOC medium for 1 h at 37 °C, cells were cultured in 5 ml of 2-YT medium supplemented with ampicillin (100 μg ml$^{-1}$) and chloramphenicol (50 μg ml$^{-1}$) and incubated overnight at 37 °C, 200 rpm. The overnight culture was then diluted to an OD$_{600}$ of 0.05 in AI medium[19] supplemented with ampicillin (100 μg ml$^{-1}$), chloramphenicol (50 μg ml$^{-1}$) and respective ncAA and/or peptide. Cells were grown overnight at 37 °C and the overnight culture was collected by centrifugation at 4,000*g* for 10 min at 4 °C and the pellets were stored at −20 °C till further use. Ub-K48pLisoK-TEV-SUMO2-K11AcK-H6 was purified as described in 'Purification of His$_6$-tagged proteins'. For cleavage, Ub-K48pLisoK-TEV-SUMO2-K11AcK-H was diluted into buffer (PBS pH 7.0 with 3 mM DTT) and incubated with TEV protease (0.1 mg ml$^{-1}$) for 30 min at room temperature.

## Reporting summary

Further information on research design is available in the Nature Portfolio Reporting Summary linked to this article.

## Data availability

Uncropped and unprocessed gels are presented in Supplementary Fig. 21. A list of plasmids (Supplementary Table 1), oligonucleotides (Supplementary Tables 2 and 3) and protein sequences used in this study is available in the Supplementary Information. All other data are presented in the main text, Supplementary Information and Methods. Any additional information is available upon request from the corresponding author. Crystallographic data for the OppA–G-SisoK structure was deposited in the RCSB Protein Data Bank with the PDB identifier 9RD1. Other X-ray crystal structures mentioned in the paper are available from the RCSB Protein Data bank under PDB identifiers 3TCF, 1GFL, 1LP1 and 3JZA. Source data are provided with this paper.

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

**Acknowledgements** This work was supported by funding from ETH Zurich and the European Research Council (ERC under the European Union's Horizon 2020 research and innovation programme, grant agreement no. 101003289—Ubl-tool to K.L.) and the Deutsche Forschungsgemeinschaft (DFG)—SFB 1309-325871075 (to M.G.). The authors thank D. Kvasha for synthesis of pBOF amino acid; Lang group members for useful discussions and input; and the staff of beamline P13 at PETRA III (DESY, Hamburg, Germany) for their support during data collection. Beamtime was allocated under proposal MX-1019.

**Author contributions** T.I., M.F. and K.L. conceived and designed the project. M.F. and T.I. led the experiments and data analysis presented in this work. C.P. performed initial experiments on G-XisoK ncAAs. P.B. performed knockout experiments to identify the Opp transporter. Y.M. created and screened OppA libraries. M.G. performed X-ray analysis. All authors analysed data and K.L. wrote the paper with input from all authors.

**Funding** Open access funding provided by Swiss Federal Institute of Technology Zurich.

**Competing interests** K.L., M.F. and T.I are named as inventors on a pending patent application EP24205590.3. The other authors declare no competing interests.

**Additional information**
**Correspondence and requests for materials** should be addressed to Kathrin Lang.

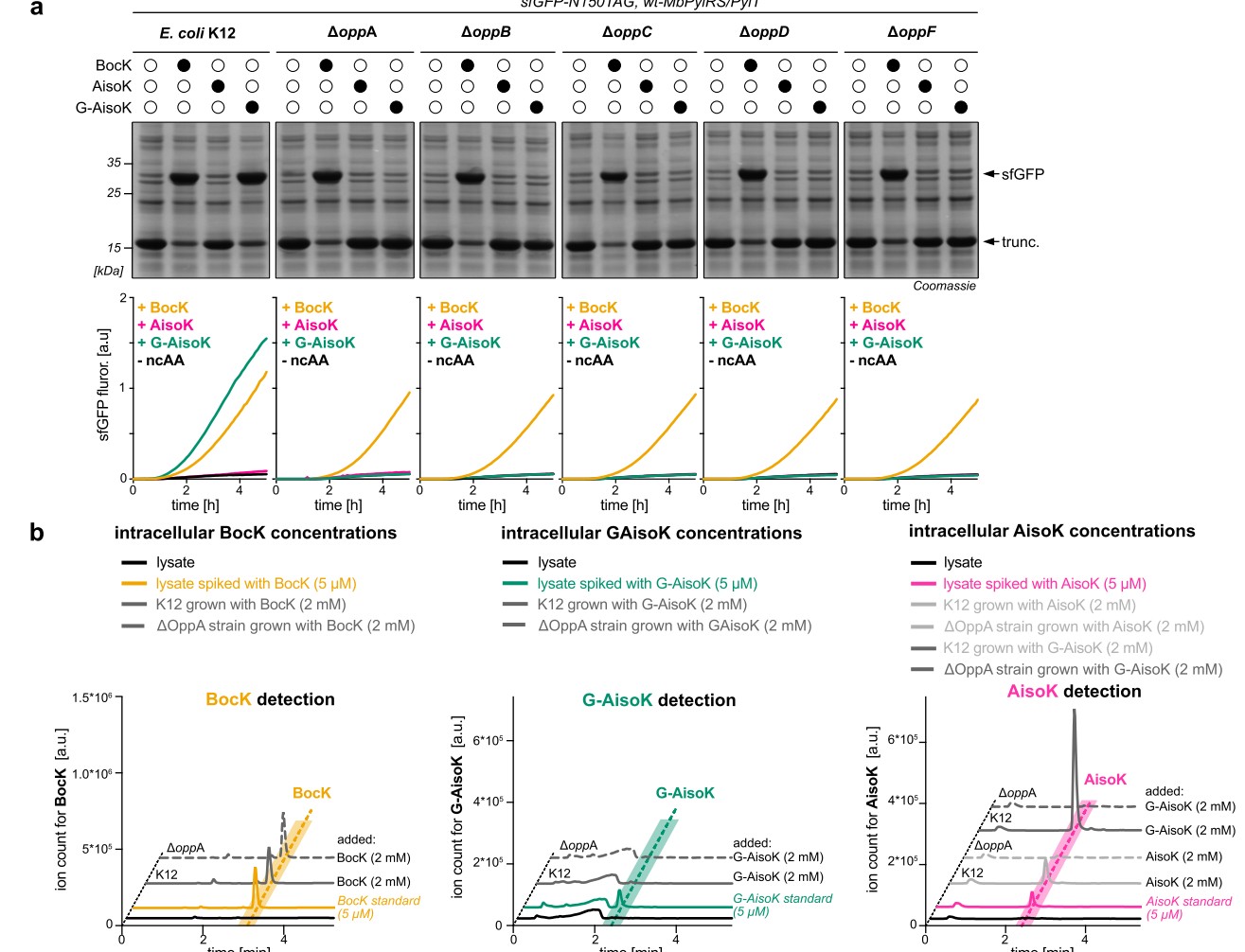

**Extended Data Fig. 1 | G-AisoK uptake is dependent on the Opp transporter.**
**a**. SDS-PAGE analysis (top) and time course fluorescence measurements (bottom) of K12 cells and knockout (KO) cell lines that have individual components of the Opp transporter genomically deleted, expressing sfGFP-N150TAG in the presence of 2 mM BocK, AisoK or G-AisoK using wt-*Mb*PylRS/PylT. Full-length sfGFP expression in presence of G-AisoK is abolished in all KOs (Δ*oppA-F*), while sfGFP expression in presence of BocK remains comparable between wt-K12 cells and KO cell lines, indicating that the Opp transporter is responsible for G-AisoK uptake and associated amber codon suppression. **b**. Extracted ion chromatograms of *E. coli* lysates to determine intracellular concentrations of BocK (left), G-AisoK (middle) and AisoK (right) in wt-K12 and Δ*oppA* cells. Intracellular BocK concentrations for cells grown with 2 mM BocK are similar between wt-K12 and Δ*oppA*-K12 (left: dashed, dark grey). Intracellular G-AisoK concentrations were negligible for both cell types (middle). In Δ*oppA* cells intracellular AisoK concentrations were negligible for cells grown in 2 mM AisoK (dashed, light grey) as well as 2 mM G-AisoK (right, dashed, dark grey). Consistent results were obtained over three distinct replicate experiments.

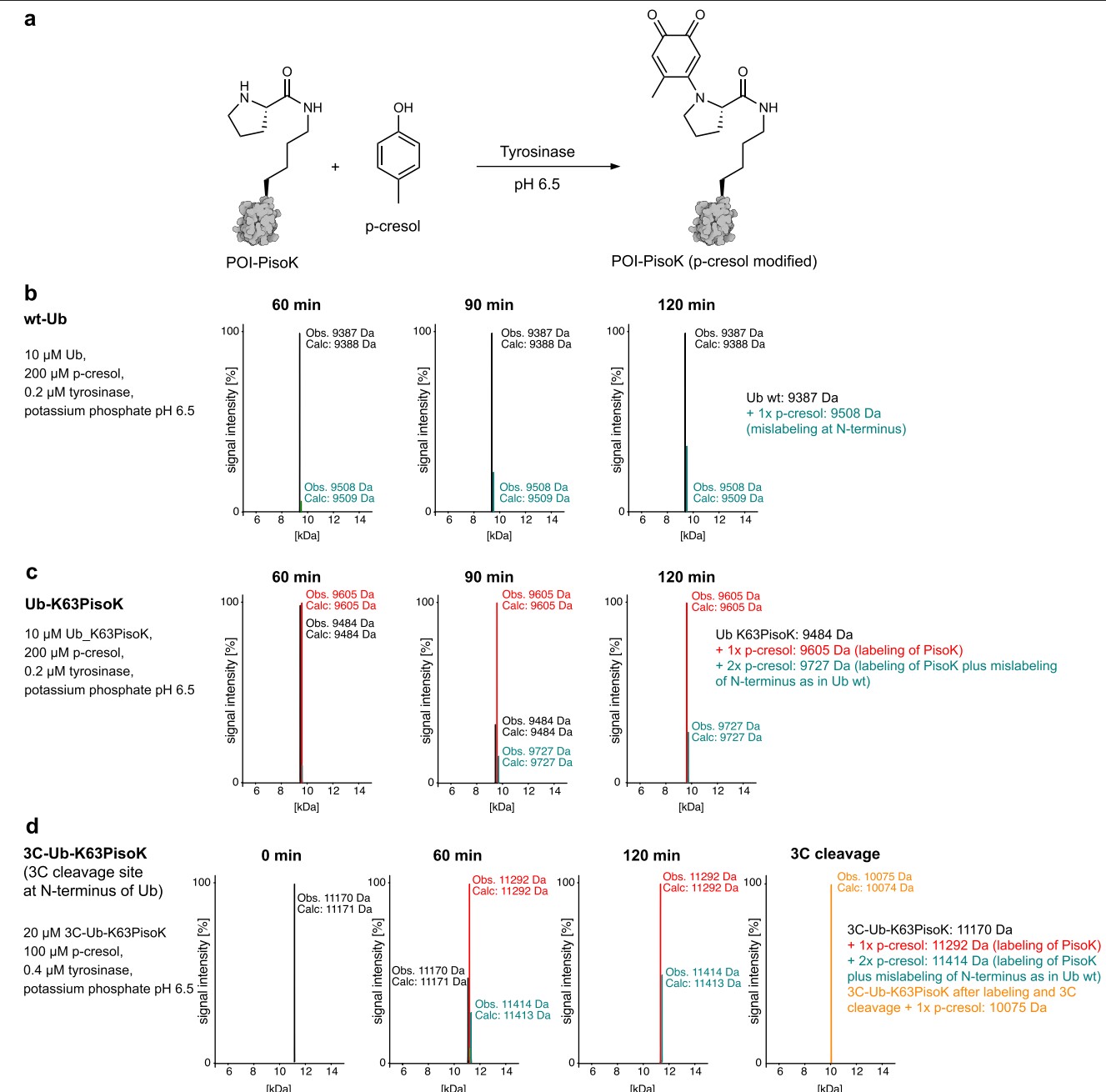

**Extended Data Fig. 2 | Tyrosinase-mediated labeling of PisoK-bearing proteins. a**. Scheme of labeling a protein of interest (POI) bearing PisoK with model substrate p-cresol mediated by tyrosinase. **b**. Incubation of wt-Ub (10 µM) with p-cresol (200 µM) in the presence of 0.2 µM tyrosinase at pH 6.5, r.t. shows mislabeling at Ub N terminus (green peak) after 120 min, as observed by LC-MS analysis. **c**. Incubation of 10 µM Ub-K63PisoK with 200 µM p-cresol with 0.2 µM tyrosinase after 120 min gives complete labeling of PisoK and some mislabeling of the N terminus as observed for wt-Ub. **d**. To confirm quantitative Ub-K63PisoK labeling, Ub bearing an N-terminal 3C protease cleavage site was incubated under the same conditions with 0.4 µM tyrosinase for 120 min, after which the labeled protein was subjected to 3C cleavage to reveal quantitative internal PisoK labeling. Consistent results were obtained over three distinct replicate experiments.

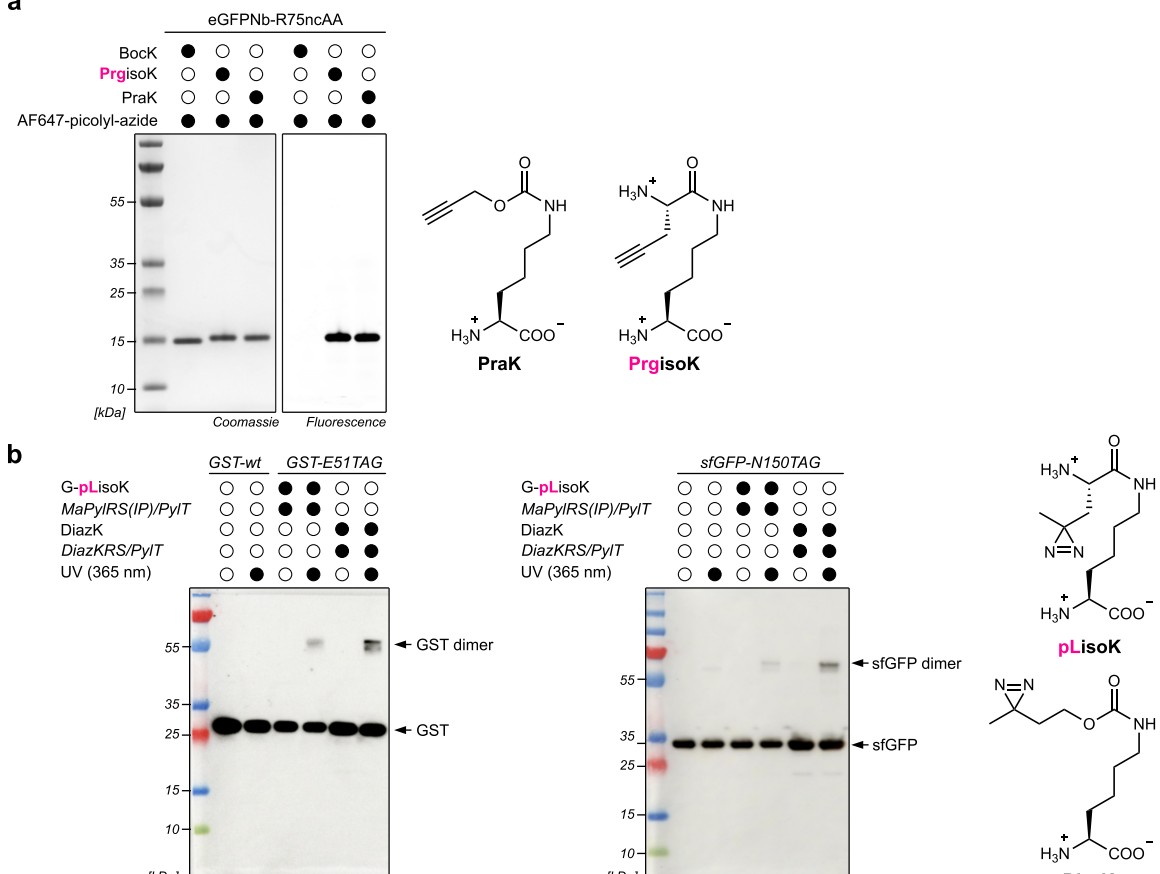

**Extended Data Fig. 3 | XisoK ncAAs for bioorthogonal labeling and photocrosslinking. a**. CuAAC-based labeling of proteins bearing a terminal alkyne moiety. Purified eGFPNb bearing either PrgisoK or the previously reported ncAA PraK[59] at position 75 are labeled with comparable efficiencies with an azide-bearing fluorophore (conditions: 100 μM AF647-Picolyl-Azide, 50 μM CuSO$_4$, 250 μM THPTA and 2.5 mM ascorbic acid in PBS for 30 min at room temperature). BocK-bearing eGFPNb is not labeled under the same conditions, as expected. Consistent results were obtained over three distinct replicate experiments. **b**. Photo-crosslinking of diazirine-bearing glutathione-transferase

(GST) and sfGFP. left: Western blot analysis of cells expressing GST-E51TAG in the presence of G-pLisoK and the previously reported DiazK[49]. Upon UV$_{365nm}$ illumination both G-pLisoK and DiazK samples show crosslinked GST dimers. Right: Western blot analysis of cells expressing sfGFP-N150TAG in the presence of G-pLisoK and DiazK. Illumination with UV$_{365nm}$ for G-pLisoK and DiazK samples show crosslinked sfGFP dimers. (conditions: cells expressing POIs were diluted to OD$_{600}$ = 1 and irradiated for 15 min at 365 nm, 15 W). Consistent results were obtained over three distinct replicate experiments.

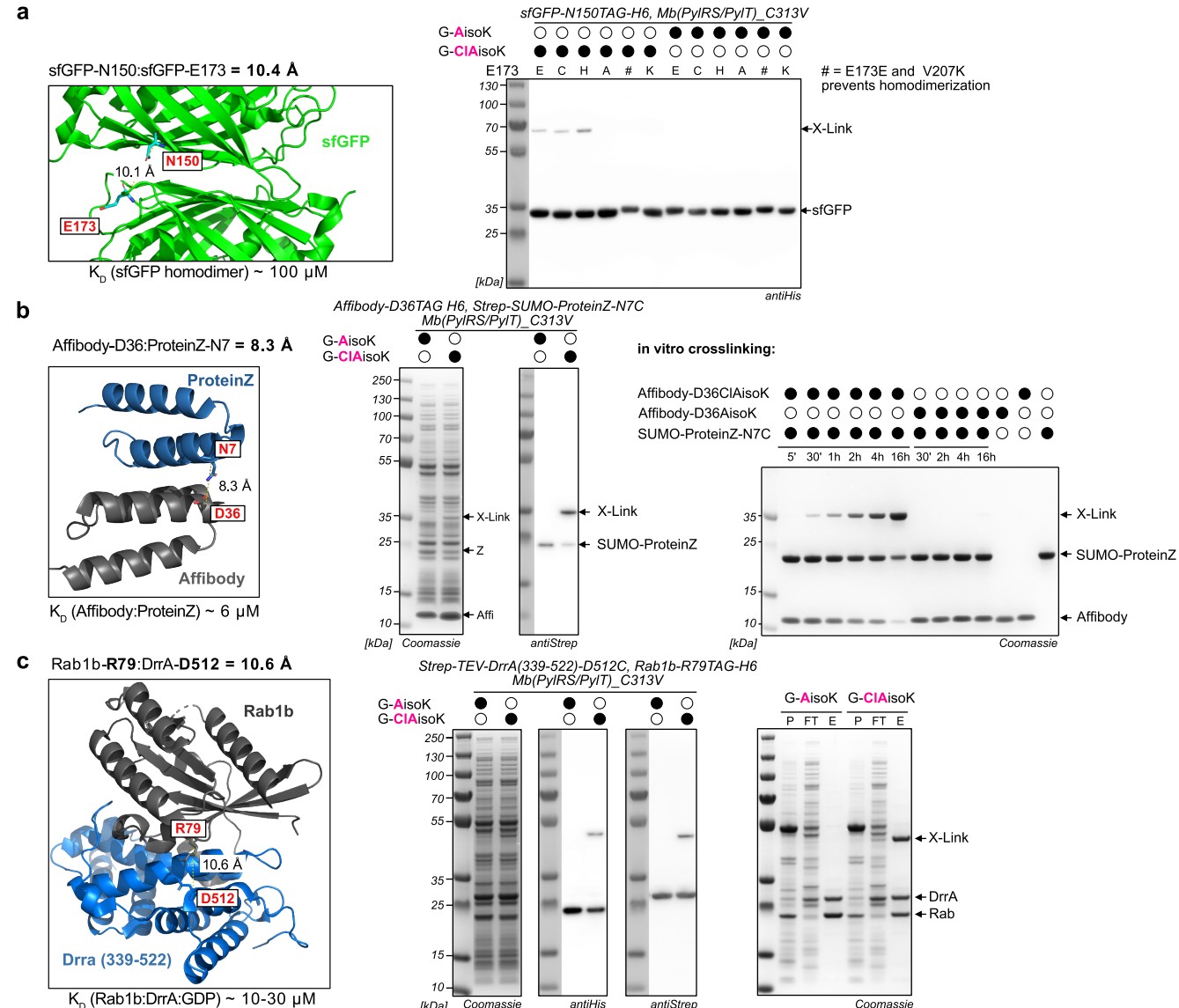

**Extended Data Fig. 4 | Proximity-induced chemical crosslinking using ClAisoK-bearing proteins. a**. Chemical crosslinking of ClAisoK-bearing sfGFP homodimer. Left: X-ray crystal structure of the sfGFP dimeric interface (PDB ID: 1GFL) denoting position N150 (where ncAAs are incorporated) and E173. The distance between the two Cα atoms of these two amino acids is 10.4 Å; the $K_D$ of sfGFP dimerization has been determined to be 100 µM[60]. Right: Western blot analysis of cells expressing sfGFP-N150TAG in the presence of G-AisoK or G-ClAisoK. Crosslinked sfGFP homodimer is observed for samples grown in the presence of G-ClAisoK and with residue 173 mutated to E, C or H. # denotes a sfGFP variant, in which a mutation (V207K) that disrupts the dimerization interface has been introduced. **b**. Chemical crosslinking of protein Z with selected affibody[50]. Left: X-ray crystal structure of affibody-protein Z complex (PDB ID: 1LP1) highlighting residues D36 in affibody (position for ncAA incorporation) and N7 in protein Z, which is mutated to cysteine to allow for crosslinking. The distance between the two Cα atoms of these two residues is 8.3 Å; the $K_D$ has been determined to be 6 µM[50]. Middle: In cellulo crosslinking: SDS-PAGE and Western blot analysis of cells expressing Strep-tagged SUMO2-proteinZ-N7C and affibody-D36TAG-H6 in the presence of either G-AisoK or

G-ClAisoK. Crosslinked complex is only observed for the G-ClAisoK sample. Right: SDS-PAGE analysis of in vitro crosslinking with purified binding partners. Crosslinked complex is observed for ClAisoK-bearing affibody after 30 min and near quantitative covalent crosslinking between affibody and protein Z is observed after 16 h. No crosslinking is observed for AisoK-bearing affibody. **c**. Complete data from Fig. 3i. Chemical crosslinking between ClAisoK-bearing Rab1b GTPase and Legionella effector protein DrrA$_{339-522}$. Left: X-ray crystal structure of Rab1b-DrrA$_{339-522}$ complex (PDB ID: 3JZA), highlighting R79 in Rab1b (changed to ncAA) and D512 in DrrA (mutated to cysteine). The distance between the two Cα atoms of these two residues is 11.4 Å; the $K_D$ of ternary complex formation in presence of GDP has been determined to be ca. 10–30 µM[51]. Western blot analysis of cells expressing both binding partners in the presence of either G-AisoK or G-ClAisoK shows crosslinked complex formation only in the presence of G-ClAisoK (middle). Purification via Ni-NTA beads enriches both binding partners for both G-AisoK and G-ClAisoK, but covalently crosslinked complex is only present in G-ClAisoK sample (right). **a-c**. Consistent results were obtained over three replicate experiments. P = pellet, FT = flow through and E = elution.

# a

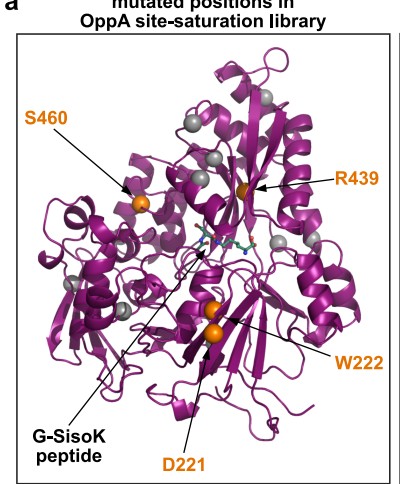

**mutated positions in OppA site-saturation library**

S460
R439
W222
G-SisoK peptide
D221

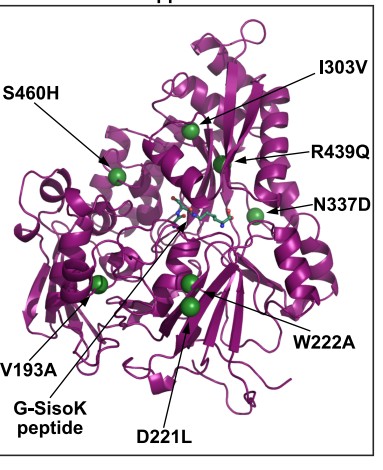

**OppA-iso**

S460H
I303V
R439Q
N337D
W222A
V193A
G-SisoK peptide
D221L

| OppA variant | mutations **bold** - targeted residues |
|---|---|
| error-prone variant 1 | T173N, **D221**G, K371E, **R439**H |
| error-prone variant 2 | L78I, **W222**R, K307R, K333N |
| error-prone variant 3 | T429A, **R439**L, **S460**C, V482A |
| error-prone variant 4 | V193A, I303V, N337D, **S460**N |
| site-saturation variant OppA-iso | V193A, **D221**L, **W222**A, I303V, N337D, **R439**Q, **S460**H |

# b

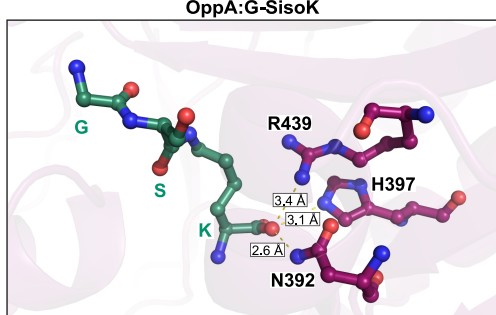

**OppA:G-SisoK**

G
S
K
R439
H397
N392
3.4 Å
3.1 Å
2.6 Å

**OppA:XXX (PDB ID: 3TCF)**

X
X
X
R439
H397
N392
3.7 Å
4.7 Å
4.9 Å

**Extended Data Fig. 5 | Evolved OppA-variant OppA-iso. a.** X-ray crystal structure of OppA:G-SisoK (PDB ID: 9RD1) with mutated positions highlighted as spheres. Left: Positions mutated in error prone OppA variants 1–4. Orange spheres indicate residues that were targeted for site-saturation mutagenesis; grey spheres denote additional mutations in the error prone OppA variants. Middle: Green spheres indicate residues mutated in the final evolved variant OppA-iso. Right: Table summarizing all mutations identified in error prone OppA variants 1–4 and in OppA-iso. Residues in bold were targeted for site saturation mutagenesis. **b.** Structural comparison of the OppA:G-SisoK complex (PDB ID: 9RD1) with the previously published structure showing OppA bound to endogenous linear peptide ligands (PDB ID: 3TCF) with a focus on binding motifs at the C-terminal end of the bound ligands. In OppA:G-SisoK, the carboxylate of lysine is in hydrogen-bonding distance to the side chains of R439, N392 and H397. In the linear peptide bound complex (OppA:XXX), the C-terminal carboxylate of the tripeptide XXX interacts primarily with the side chain of R439 and residues N392 and H397 are not ideally positioned to contribute to strong hydrogen-bonding, suggesting that the interaction with R439 is most important for linear tripeptides. In contrast, for isopeptide-linked tripeptides such as G-SisoK the hydrogen-bond donors, H397 and N392, are well positioned to interact with the lysine carboxylate, likely stabilizing the complex also in the absence of a positively charged side chain at position 439.

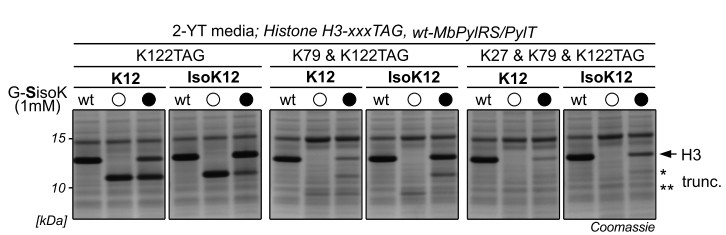

**Extended Data Fig. 6 | Incorporation of XisoK derivatives using the IsoK12 strain. a.** SDS-PAGE analysis of sfGFP-N150TAG expression in 2-YT, comparing K12 with IsoK12 cells at different G-SisoK concentrations. Full-length sfGFP levels at 0.1 mM G-SisoK in IsoK12 are comparable to those obtained with 1 mM G-SisoK in wt-K12. **b.** Top: SDS-PAGE analysis of amber suppression of proliferating cell nuclear antigen (PCNA-K164TAG), heat shock protein 82 (Hsp82-D452TAG), calmodulin (calmodulin-G40TAG) and interleukin-2 (IL-2-R38TAG) with 1 mM G-SisoK in wt-K12 or IsoK12 cells grown in 2-YT. The expression of full-length proteins is higher in IsoK12 strain compared to wt-K12. Bottom: SDS-PAGE analysis of amber suppression of RanGTPase activating protein (RanGAP-K524TAG) and human growth hormone (hGH-Y35TAG and hGH-K38TAG) expressed with 1 mM BocK or G-SisoK in IsoK12 cells grown in 2-YT. Full-length protein expression levels are higher with G-SisoK in comparison to BocK. **c.** Left: SDS-PAGE analysis of eGFPNb-R17TAG expression in the presence of 1 mM G-PrgisoK or PraK[59] in AI or 2-YT media comparing wt-K12 with IsoK12. Full-length eGFPNb expression is highest in IsoK12 cells grown in 2-YT. Right: Preparative protein yields of wt-eGFPNb, eGFPNb-R17PrgisoK or eGFPNb-R17PraK. wt-MbPylRS/PylT pair was used for PraK and G-PrgisoK incorporation. PrgisoK-bearing eGFPNb yields exceed yields for PraK-bearing eGFPNb, matching wt-eGFPNb yields. For structure of PraK see Extended Data Fig. 3a. **d.** SDS-PAGE analysis of Histone H3 expression with single (K122TAG), double (K79TAG, K122TAG) and triple (K27TAG, K79TAG, K122TAG) amber suppression in presence of 1 mM G-SisoK comparing K12 with IsoK12 cells grown in 2-YT media. **a-d.** Consistent results were obtained over three distinct replicate experiments. Arrows indicated full-length POIs, asterisks indicated truncated POIs.

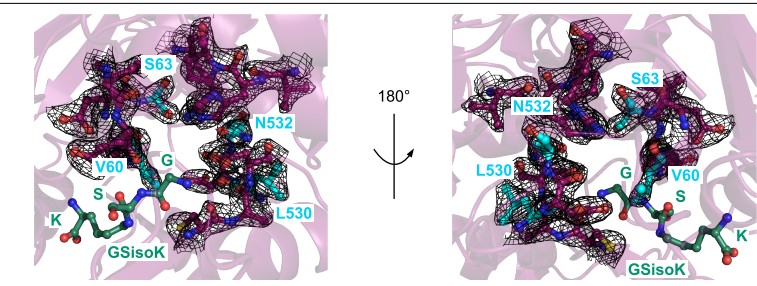

**a**

**b**

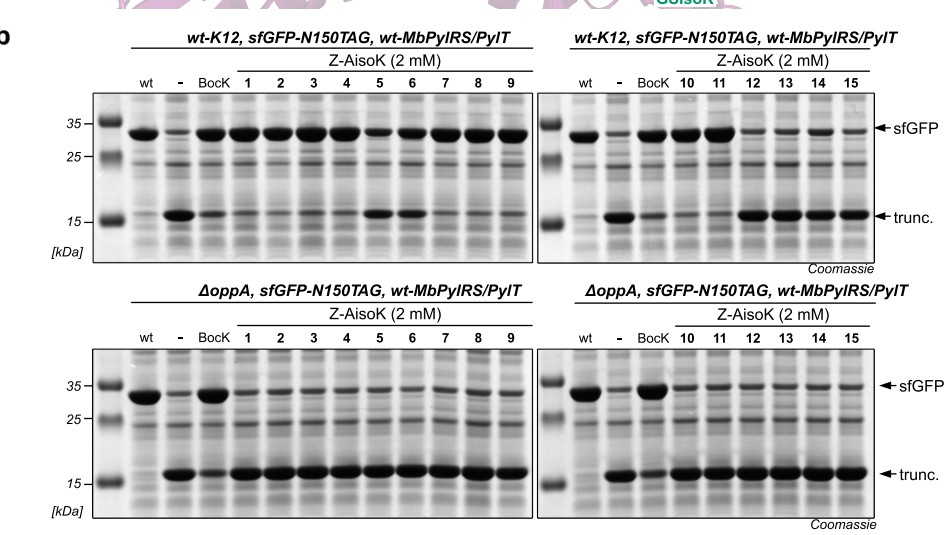

**c**

OppA-Z1 (evolved in presence of 13): V60E, S63G L530G and N532S -> **K12-Z1 strain**
OppA-Z2 (evolved in presence of 15): S63A, L530G and N532A and non-programmed R439H -> **K12-Z2 strain**

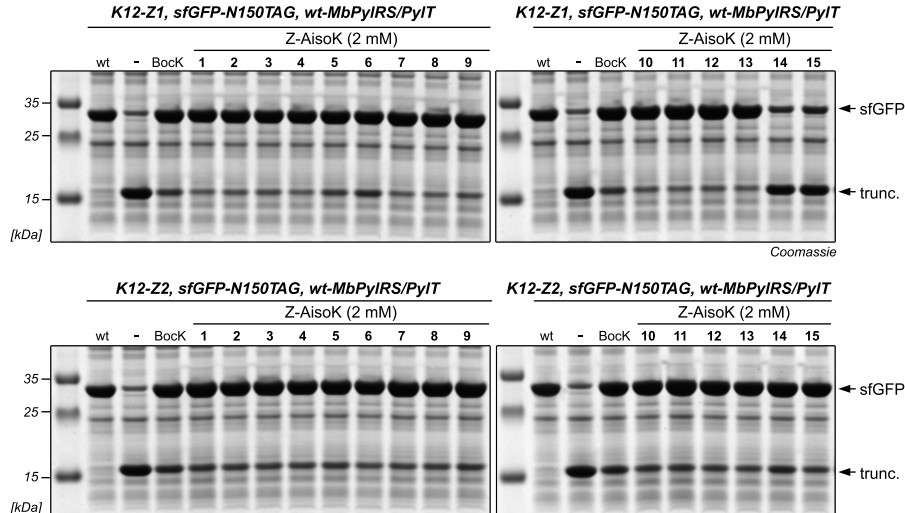

**Extended Data Fig. 7 | Transport of ncAAs in the 'Z' position of a Z-AisoK tripeptide. a.** X-ray crystal structure of OppA:G-SisoK reveals a cavity around the N-terminal glycine, suggesting that larger sidechains can be accommodated at the 'Z' position of a generalized Z-XisoK peptide. **b.** SDS-PAGE analysis of sfGFP-N150TAG expression in the presence of different Z-AisoK peptides with wt-*Mb*PylRS/PylT. Full-length sfGFP expression indicates efficient Z-AisoK transport, intracellular processing into Z and AisoK and AisoK incorporation by the wt-*Mb*PylRS/PylT. Top: sfGFP expression in wt-K12 cells. Z-AisoK tripeptides 1, 2, 3, 4, 6, 7, 8, 9, 10 and 11 lead to efficient full-length sfGFP expression comparable to wt-sfGFP expression. Bottom: sfGFP expression in

*ΔoppA* cells. Full-length sfGFP expression in the presence of Z-AisoK peptides is completely abolished, indicating that Z-AisoK transport is dependent on OppA. **c.** SDS-PAGE analysis of sfGFP-N150TAG expression in evolved Z-strains (as in (b)). Top: Expression in K12-Z1. All Z-AisoK tripeptides, apart from 14 and 15, bearing negatively charged Z-residues lead to efficient AisoK incorporation, indicating that they are successfully imported and cleaved. Bottom: Expression in K12-Z2. All Z-AisoK tripeptides, including those bearing negatively charged Z-residues (14,15) lead to efficient full-length sfGFP expression and AisoK incorporation. **b,c.** Consistent results were obtained over three distinct replicate experiments.

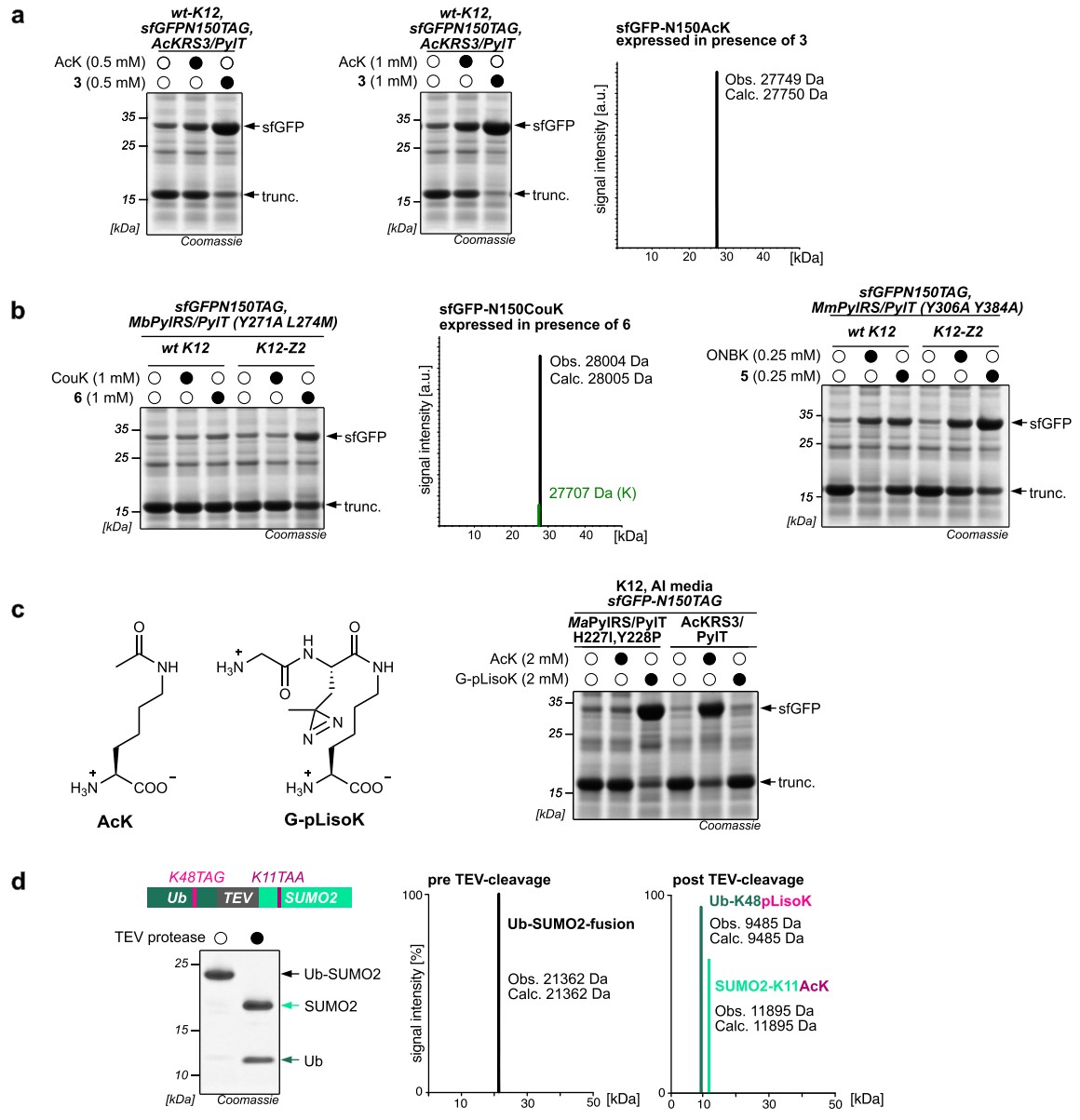

**Extended Data Fig. 8 | Z-AisoK tripeptides allow efficient incorporation of Z and dual encoding of two ncAAs using a single tripeptide. a**. Left: SDS-PAGE analysis of sfGFP-N150TAG expression in the presence of AcK or the corresponding Z-AisoK peptide AcK-AisoK (3) with AcKRS3/PylT[53]. Full-length sfGFP expression is significantly higher when supplementing 3 instead of AcK. Right: LC-MS analysis of sfGFP purified from cultures grown in the presence of 3. Observed mass indicates AcK incorporation. Consistent results were obtained over three distinct replicates. **b**. Left: SDS-PAGE analysis of sfGFP-N150TAG expression in the presence of CouK or the corresponding Z-AisoK tripeptide CouK-AisoK (6). Full-length sfGFP expression is higher in K12-Z2 cells in presence of 6 compared to wt-K12 or compared to supplementation with CouK. Middle: LC-MS analysis of sfGFP purified from K12-Z2 cells grown in the presence of tripeptide 6. Observed mass indicates incorporation of CouK with a second peak indicating incorporation of lysine likely due to photo-decaging of CouK. Right: SDS-PAGE analysis of sfGFP-N150TAG expression in the presence of

ONBK or the corresponding Z-AisoK tripeptide (5). Full-length sfGFP expression is higher in K12-Z2 cells with tripeptide 5 in comparison to the same in wt-K12 cells or with ONBK. Consistent results were obtained over three distinct replicates. **c**. Left: Chemical structures of ncAAs for dual stop codon suppression to allow for orthogonal incorporation of AcK and pLisoK. Right: SDS-PAGE analysis of sfGFP-N150TAG expression in the presence of 2 mM AcK or 2 mM G-pLisoK and their corresponding PylRS/PylT pairs to confirm substrate orthogonality between the pairs. Full-length sfGFP expression in presence of AcK is only observed with AcKRS3/PylT and in presence of G-pLisoK only with *Ma*PylRS(H227I/Y228P)/PylT, hence confirming orthogonality. **d**. Left: Scheme of Ub-SUMO fusion construct used to confirm dual and orthogonal incorporation of pLisoK and AcK. SDS-PAGE analysis shows purified full-length construct and products of TEV protease cleavage. Right: LC-MS analysis of full-length construct and cleaved products confirming pLisoK and AcK incorporation at the intended positions.

# Reporting Summary

## Statistics

For all statistical analyses, confirm that the following items are present in the figure legend, table legend, main text, or Methods section.

| n/a | Confirmed | |
|---|---|---|
| ☐ | ☒ | The exact sample size (*n*) for each experimental group/condition, given as a discrete number and unit of measurement |
| ☐ | ☒ | A statement on whether measurements were taken from distinct samples or whether the same sample was measured repeatedly |
| ☒ | ☐ | The statistical test(s) used AND whether they are one- or two-sided *Only common tests should be described solely by name; describe more complex techniques in the Methods section.* |
| ☒ | ☐ | A description of all covariates tested |
| ☒ | ☐ | A description of any assumptions or corrections, such as tests of normality and adjustment for multiple comparisons |
| ☐ | ☒ | A full description of the statistical parameters including central tendency (e.g. means) or other basic estimates (e.g. regression coefficient) AND variation (e.g. standard deviation) or associated estimates of uncertainty (e.g. confidence intervals) |
| ☒ | ☐ | For null hypothesis testing, the test statistic (e.g. *F*, *t*, *r*) with confidence intervals, effect sizes, degrees of freedom and *P* value noted *Give P values as exact values whenever suitable.* |
| ☒ | ☐ | For Bayesian analysis, information on the choice of priors and Markov chain Monte Carlo settings |
| ☒ | ☐ | For hierarchical and complex designs, identification of the appropriate level for tests and full reporting of outcomes |
| ☒ | ☐ | Estimates of effect sizes (e.g. Cohen's *d*, Pearson's *r*), indicating how they were calculated |

*Our web collection on statistics for biologists contains articles on many of the points above.*

## Software and code

Policy information about availability of computer code

| | |
|---|---|
| Data collection | All gels and western blots were imaged using: iBright Imager TM Smart Digital Imaging (Thermo Fischer Scientific, 1.8.1.), Amersham ImageQuant 800 Control Software (Cytiva, 2.0.0) LC-MS data was collected using OpenLab ChemStation (Agilent, LTS01.11 (251). |
| Data analysis | Graphpad Prism 10 was used to generate all graphs in this study, data analysis for microscale thermophoresis were performed on MO.affinity Analysis (v3.0.5, NanoTemper Technologies). LC-MS data was analyzed on OpenLab ChemStation (Agilent, LTS01.11 (251) XDS , REFMAC 5, ARP/wARP 8.0, PRODRG, MOLPROBITY 4.0.4, PHASER 2.7.0, COOT v0.9, PyMol 2.3.5 and CCP4 suite 7.0 for X-ray structure analysis |

For manuscripts utilizing custom algorithms or software that are central to the research but not yet described in published literature, software must be made available to editors and reviewers. We strongly encourage code deposition in a community repository (e.g. GitHub). See the Nature Portfolio guidelines for submitting code & software for further information.

## Data

Policy information about availability of data

All manuscripts must include a data availability statement. This statement should provide the following information, where applicable:

- Accession codes, unique identifiers, or web links for publicly available datasets
- A description of any restrictions on data availability
- For clinical datasets or third party data, please ensure that the statement adheres to our policy

Source data for graphs in this study can be found in supplementary data. Uncropped and unprocessed gels can be found in Supplementary information Figure S20. A list of plasmids (Table 1) oligonucleotides (Table 2, 3) and protein sequences used in this study are available in Supplementary Information. All other relevant data is present in the main text, supplementary information and methods. Any additional data is available upon request from the corresponding author. Crystallographic data for the OppA:GSisoK structure was deposited in the RCSB Protein Data Bank with the PDB identification numbers 9RD1. Other X-ry crystal structures mentioned in the paper are available in the RCSB Protein Data bank under identifications numbers: 3TCF, 1GFL, 1LP1, 3JZA.

## Research involving human participants, their data, or biological material

Policy information about studies with human participants or human data. See also policy information about sex, gender (identity/presentation), and sexual orientation and race, ethnicity and racism.

| | |
|---|---|
| Reporting on sex and gender | not applicabe |
| Reporting on race, ethnicity, or other socially relevant groupings | not applicable |
| Population characteristics | not applicable |
| Recruitment | not applicable |
| Ethics oversight | not applicable |

Note that full information on the approval of the study protocol must also be provided in the manuscript.

# Field-specific reporting

Please select the one below that is the best fit for your research. If you are not sure, read the appropriate sections before making your selection.

☒ Life sciences ☐ Behavioural & social sciences ☐ Ecological, evolutionary & environmental sciences

For a reference copy of the document with all sections, see nature.com/documents/nr-reporting-summary-flat.pdf

# Life sciences study design

All studies must disclose on these points even when the disclosure is negative.

| | |
|---|---|
| Sample size | No statistical methods were used to determine sample size. For all experiments (SDS-PAGE, western blot, fluorescnce traces, LC-MS assays as well as determination of binding constants) three distinct replicates were anaylzed which is common practice in biological sciences. |
| Data exclusions | no data was excluded |
| Replication | For all experiments three distinct replicates were performed and all attempts at replication were successful. |
| Randomization | Randomization was not relevant for this study. Potential covariates were controlled by ensuring uniform handling across all samples. All samples were processed using standardized protocols, and exposure to procedural variation was minimized. |
| Blinding | Blinding was not performed as experimental conditions were evident. All samples were processed using standardized protocols, and exposure to procedural variation was minimized. |

# Reporting for specific materials, systems and methods

We require information from authors about some types of materials, experimental systems and methods used in many studies. Here, indicate whether each material, system or method listed is relevant to your study. If you are not sure if a list item applies to your research, read the appropriate section before selecting a response.

## Materials & experimental systems

| n/a | Involved in the study |
|-----|----------------------|
| ☐ | ☒ Antibodies |
| ☒ | ☐ Eukaryotic cell lines |
| ☒ | ☐ Palaeontology and archaeology |
| ☒ | ☐ Animals and other organisms |
| ☒ | ☐ Clinical data |
| ☒ | ☐ Dual use research of concern |
| ☒ | ☐ Plants |

## Methods

| n/a | Involved in the study |
|-----|----------------------|
| ☒ | ☐ ChIP-seq |
| ☒ | ☐ Flow cytometry |
| ☒ | ☐ MRI-based neuroimaging |

## Antibodies

| | |
|---|---|
| Antibodies used | anti-His-HRP dilution 1:10000 (Anti-His-Preoxidase, Roche Cat. no. 11965085001), Anti-Strep-HRP 1:10000 (StrepMAB-Classic HRP, IBA, Cat.No. 2-1509-001). |
| Validation | The validation statements of the commercial antibodies are available on the website of the suppliers:<br><br>anti-His-HRP (Roche Cat. no. 11965085001): https://www.sigmaaldrich.com/CH/en/product/roche/11965085001<br>anti-Strep-HRP (Cat.No. 2-1509-001): https://www.iba-lifesciences.com/strepmab-classic-hrp/2-1509-001 |

## Plants

| | |
|---|---|
| Seed stocks | not applicable |
| Novel plant genotypes | not applicable |
| Authentication | not applicable |

