## [Peer Review File · Nature]

Hijacking a bacterial membrane transporter for efficient genetic code expansion

Corresponding Author: Professor Kathrin Lang

Version 0:

Reviewer comments:

Referee #1

(Remarks to the Author)

The manuscript from lype et al. presents an elegant body of work elucidating how the import of synthetic noncanonical amino acids (ncAAs) can be engineered to significantly improve their co-translational incorporation efficiency in *E. coli*. The issue of poor uptake of various synthetic ncAAs is well-established, particularly for those with polar side chains that hinder diffusion across the plasma membrane. Although there have been sporadic attempts in the past to address this issue using different approaches, including short peptide transporters, an in-depth understanding of the underlying process is lacking, making it a major unresolved challenge.

Through careful scholarly work, this manuscript provides deep insights into a specific transport system in *E. coli* that enables the efficient uptake of isoK tri-peptides and their subsequent hydrolysis to isoK dipeptides. Using clever engineering, the authors further expand the scope of this transport machinery to facilitate the efficient uptake and incorporation of several ncAAs. The paper is well-written and technically rigorous.

The transporter described here appears to be specific to the unique isoK chemotype. While useful for this particular subclass of ncAAs, this specificity would have limited the approach by excluding numerous other challenging ncAAs with different architectures. However, the exciting experiments described in Figure 5 suggest otherwise. In these experiments, an Ack is included in the isoK tripeptide scaffold to generate Ack-pLisoK, which is cleaved after cellular uptake to produce free Ack. If this mechanism is generalizable, it could provide a great strategy to efficiently deliver many other challenging ncAAs into *E. coli*.

Currently, only one such example is shown (Ack-pLisoK), and it is unclear how general this strategy is. I would encourage the authors to explore the generality of this approach by demonstrating the delivery of a few other challenging ncAAs using this strategy, which would significantly strengthen the impact of their work. Even if some of the other ncAAs don't work immediately, this would not be a negative outcome. The Opp transporter can be engineered to accept them in the future using the elegant directed evolution platform already established by the authors. However, this exploration would provide the necessary insight into the current scope of this promising strategy.

A few minor things:

- I'd recommend providing a summary figure in the SI showing structures of all ncAAs in one place,
- line 69: When discussing biosynthetic generation of ncAAs, authors may consider including the following references: [10.1016/j.chempr.2020.07.013](https://doi.org/10.1016/j.chempr.2020.07.013), [10.1002/anie.202014540](https://doi.org/10.1002/anie.202014540)
- line 84: When discussing lysine dipeptides, the authors may consider including the following reference: [10.1021/acs.biochem.4c00530](https://doi.org/10.1021/acs.biochem.4c00530)
- line 166: Shuttle?

Referee #2

(Remarks to the Author)

lype et al. describe the engineering of a bacterial membrane transport system that facilitates import of tripeptides bearing non-canonical amino acids (ncAAs) into bacterial cells. This system if broadly applicable overcomes important bottlenecks associated with poor ncAA encoding into target proteins due to poor cellular import of ncAAs. The authors show clearly that the encoding efficiency of many ncAAs is significantly improved using this transporter-based strategy, including 11 novel ncAAs. They also engineer a variant of the OppA with increased affinity for the tripeptides resulting in improved encoding at lower peptide concentrations with a variety of media. The manuscript is well written and organized, and the data presented clearly and convincingly. Overall, while the study is clear and thorough, I believe the results presented here are of moderate to high interest to researchers in this specific discipline, and of limited interest to a broader audience based on the major comments below. My support enthusiasm would be higher if truly important or novel functional groups that have not been encoded before were enabled by this technology.

Major comments:

- 1) As noted by the authors in the introduction, functionalizing ncAAs into pro-peptides for improved transport and encoding has previously been proposed and demonstrated, both by themselves as well as others (refs. 28, 30-32). Presented here is largely an improved adaptation of these same concepts which reduces the significance of the work.
- 2) The ncAAs encoded here with this system are also restricted to the XisoK structural scaffold and so the scope of ncAAs whose encoding would benefit from this transporter-assisted system appears quite narrow and restrictive. This particular point is not discussed or acknowledged by the authors.
- 3) While the authors are correct that many of the ncAAs encoded with this new system are structurally novel, these ncAAs have chemical functionalities (e.g. alkynes, diazurines, proximity crosslinkers, etc) that are commonly and efficiently encoded using alternative ncAA structures and so the practical advance isn't as obvious.
- 4) On this note, while the paper is largely focused on demonstrating installation of isoK based ncAAs, I find most interesting their finding that the amino acid in the first position of peptide could be altered from Gly to an ncAA, and this ncAA can be genetically encoded. This implies that ncAAs installed at the first position of the pro-peptide are not restricted to the XisoK structural scaffold (acetyl-lysine in this case) and they can still be encoded. Evaluating the scope of ncAA structures that can be tolerated at this position without the XisoK restriction would make this strategy much more versatile.

Minor issue

- 1) The MS data is provided for all appropriate new encoding technology. The mass axis is broad showing that there are no significant changes in mass but it is so broad that small changes in mass that would result from ncAA degradation, reaction and or GCE miss encoding evens from near cognate suppression are not detectable. The claims on efficient encoding of ncAAs would be supported if supplemental information contained MS data with an axis scale enabling detection of salt ions and small MW changes from the predicted mass.

Referee #3

(Remarks to the Author)

This study addresses the low efficiency in incorporating non-canonical amino acids (ncAAs) by genetic code expansion. Low yields and limited access to these specialized amino acids have hindered a wider adoption of such methods for producing protein with potential medical and industrial uses. In this study, a new approach is introduced that combines a well-known "pro-peptide strategy" with a well-characterized bacterial oligopeptide ABC transport system OppA to boost intracellular levels of ncAAs. The researchers identified inadequate cellular uptake of ncAAs on *E. coli* as a primary barrier and tackled it by repurposing a well-characterized oligopeptide ABC transport system to piggyback ncAAs within tripeptide scaffolds with isopeptide-linked ncAAs.

Strengths: This method allowed the efficient incorporation of eleven previously inaccessible ncAAs with diverse functionalities. Additionally, the authors engineered an *E. coli* strain with a transporter optimized for tripeptide-linked ncAAs, achieving incorporation efficiencies comparable to wildtype protein levels. The authors adapted the system for co-transporting two distinct ncAAs, enabling their simultaneous incorporation. This work highlights the importance of optimizing ncAA uptake in *E. coli*, paving the way for scalable production of modified proteins and expanding protein chemistry without relying on passive membrane permeability.

Weaknesses: The important limitation of the study is that the system is currently restricted to *E. coli*. There are doubts about its potential to adapt to mammalian cells. In addition, all test proteins (e.g., eGFP, histone, ubiquitin, etc.) have been used previously and have been shown to be effective in amber suppression. No new targets were tested or validated in this study. Furthermore, no new biological question has been addressed. Alternatively, isopeptide-linked tripeptides could be shuttled into cells via other hydrophobic, cleavable modifications. Finally, these isopeptide-linked peptide conjugates have recently been studied by the authors (ref. 30-32). The present study is based on the observation that the N-terminal glycine is cleaved off in the cytosol and that tripeptides G-AisoK in contrast to the dipeptide AisoK display a high level of amber-suppressed sfGFPN150TAG production.

In addition to the peptide transport system OppA, it would be a decisive step forward if the authors could identify the peptidase that is essential for the removal of the N-terminal glycine. Instead of single-gene knockouts the approach is biased on a potential target list, which is unfortunately not provided in the manuscript. Double/triple knockouts or transposon mutagenesis should be considered to identify unknown peptidases for the cleavage of the N-terminal glycine.

Specific points:

- 1) Line 74-76: The author hypothesized that their approach holds the potential of being widely applicable; however, how can

this be expanded to mammalian cells? The authors could easily test whether some of the isopeptide-linked tripeptides are taken up by mammalian cells.

2) Figure 1 and throughout the manuscript: The ratios between full-length and truncate proteins should always be quantified in biological triplicates.

3) Line 181-189: This part can be shortened as this transport system is well described. It should be explained in the SI. In several case the manuscript contains very long introductions into side topics, which are well-characterized in literature.

4) Line 201-213: Similarly, this part can be shortened.

5) Figure 2b and 2c should be removed as the mechanism of peptide translocation is not supported by the experimental data and well-described in literature. In addition, the structure of the import system OppABCDF does not provide additional support for the present study.

6) Line 255: It is not clear how the tests were done, in vitro with purified protein or in cellulo or in lysate.

7) Figure 3b: Why is XisoK equally well incorporated as G-XisoK?

8) Figure 3: The cross-linked bands should be quantified.

9) Line 407-419: As the crystallization of OppA is established and well-described, the binding properties of the evolved OppA-ev2 and the structural consequence of the seven mutations should be examined via an experimental x-ray structure in a G-SisoK/OppA-ev2 complex. Just mapping the seven mutations found in OppA-ev2 onto a published OppA structure liganded to a linear tripeptide can hardly explain the significant substrate preference for isopeptide-linked G-SisoK versus GSK.

10) Line 443-445: To strengthen the conclusion, the authors should quantify the intensities.

11) Line 497-499: Frequently, the authors state the significantly higher protein yields were achieved by their approach by using isopeptide-linked ncAAs. However, these statements must be corroborated by quantitative data on at least three biological replica and different model systems.

12) Line 570-573: The system is currently limited to *E. coli*. There are doubts about its potential to adapt to mammalian or yeast cells.

13) The authors speculate the availability of ncAA is limited by passive diffusion. This statement needs to be carefully discussed and reconsidered. The novelty of the approach is hampered as similar strategies were applied using the dipeptide transporter Dpp.

14) The conclusion is more a summary of the results and slightly redundant. Instead, the conclusions should address the major limitation of the study and important future developments.

15) Supporting Information: Synthesis via solid phase peptide synthesis: Which orthogonal protection group chemistry was used to introduce the iso-peptide bonds on solid-phase synthesis. This is not stated.

16) SI Table 4: The purity of the peptide should be given.

Minor points:

1) Line 85: Abbreviation for Dpp is missing.

2) Figure 1: Coomassie needs to be capitalized.

3) Line 162: ABC transporter should be written without hyphen throughout the entire manuscript.

4) Line 163 should read: In gram-negative bacteria.

5) Figure 2: It is not clear what the dots stand for. I assume the presence of Bock, AisoK and G-AisoK. Please explain in the figure legend.

6) Line 259: no hyphen between MS-analysis.

7) Line 496: sfGFP-N40Ack should be sfGFP-N40AcK.

8) Line 533: This information should be mentioned before.

Version 1:

Reviewer comments:

Referee #1

(Remarks to the Author)

The revised manuscript includes a substantial body of new data that convincingly highlights the broad applicability of the authors' ncAA import platform. The manuscript is significantly improved and addresses all of my previous suggestions. Congratulations to the authors on this great story.

Referee #3

(Remarks to the Author)

The authors have thoroughly addressed the reviewers' critical concerns in a detailed point-to-point response. Importantly, they have provided a substantial set of additional experiments, which significantly strengthen their overall conclusions.

Given that genetic code expansion in the bacterial model *Escherichia coli* remains an active area of research, the authors' innovative approach, hijacking a type I ABC import system of *E. coli*, represents a notable advance. This strategy enables efficient incorporation of a broad range of previously inaccessible non-canonical amino acids. Additionally, the development of a directed evolution platform to tailor transporters for specific ncAAs further enhances the impact and versatility of the method.

The initial claims of translating the approach to eukaryotic system, particularly human cells, has been appropriately moderated, reflecting current technical challenges.

We are grateful for the thoughtful and thorough comments provided by the reviewers, which have helped us to significantly expand and strengthen our study. Please find our detailed point-by-point responses to all reviewer comments below. All novel data, new figures, and manuscript changes have been clearly marked in grey in the revised version of the manuscript for ease of reference.

Referees' comments:

Referee #1 (Remarks to the Author):

The manuscript from Iype et al. presents an elegant body of work elucidating how the import of synthetic noncanonical amino acids (ncAAs) can be engineered to significantly improve their co-translational incorporation efficiency in *E. coli*. The issue of poor uptake of various synthetic ncAAs is well-established, particularly for those with polar side chains that hinder diffusion across the plasma membrane. Although there have been sporadic attempts in the past to address this issue using different approaches, including short peptide transporters, an in-depth understanding of the underlying process is lacking, making it a major unresolved challenge.

Through careful scholarly work, this manuscript provides deep insights into a specific transport system in *E. coli* that enables the efficient uptake of isoK tri-peptides and their subsequent hydrolysis to isoK dipeptides. Using clever engineering, the authors further expand the scope of this transport machinery to facilitate the efficient uptake and incorporation of several ncAAs. The paper is well-written and technically rigorous.

We thank this reviewer for the enthusiastic feedback.

The transporter described here appears to be specific to the unique isoK chemotype. While useful for this particular subclass of ncAAs, this specificity would have limited the approach by excluding numerous other challenging ncAAs with different architectures. However, the exciting experiments described in Figure 5 suggest otherwise. In these experiments, an AcK is included in the isoK tripeptide scaffold to generate AcK-pLisoK, which is cleaved after cellular uptake to produce free AcK. If this mechanism is generalizable, it could provide a great strategy to efficiently deliver many other challenging ncAAs into *E. coli*.

Currently, only one such example is shown (AcK-pLisoK), and it is unclear how general this strategy is. I would encourage the authors to explore the generality of this approach by demonstrating the delivery of a few other challenging ncAAs using this strategy, which would significantly strengthen the impact of their work. Even if some of the other ncAAs don't work immediately, this would not be a negative outcome. The Opp transporter can be engineered to accept them in the future using the elegant directed evolution platform already established by the authors. However, this exploration would provide the necessary insight into the current scope of this promising strategy.

We thank the reviewer for these suggestions. We agree that generalization of our approach to any kind of ncAAs, including challenging ones that are known to show low-cell permeability, would be a game changer for genetic code expansion.

Towards this goal we have adapted our uptake strategy for tripeptides of a general Z-XisoK scaffold. We have synthesized a panel of 14 different tripeptides based on a Z-AisoK scaffold. As N-terminal residue Z, we selected either natural amino acids or ncAAs with diverse side

chains, including bulky or negatively charged groups known to exhibit poor or negligible cell permeability. We used AisoK incorporation into sfGFP-N150TAG to monitor efficient uptake and cleavage of Z. Eight out of these 14 tripeptides showed efficient uptake and processing in wt K12 cells, as seen by efficient AisoK incorporation and we confirmed that Z-AisoK uptake and subsequent cleavage and AisoK incorporation is indeed dependent on wt-OppA. In contrast, Z-AisoK tripeptides bearing bulkier or negatively charged Z residues, resulted in either reduced or completely abolished AisoK incorporation into sfGFP, suggesting inefficient import and/or cleavage of these substrates by wt-OppA and endogenous peptidases. Given that we had already established a robust platform for OppA evolution, we sought to determine whether AisoK incorporation into sfGFP-N150TAG could again be leveraged as a selection tool to evolve new OppA variants capable of accommodating tripeptides exhibiting larger or negatively charged Z-residues. Guided by the crystal structure of a wt-OppA:G-SisoK complex – that we have solved during revision - we selected residues surrounding the glycine-binding pocket to be randomized for accommodating Z-AisoK tripeptides bearing bulky or negatively charged Zs. We subjected a corresponding site-saturation library to multiple FACS-based enrichments in presence of two of these Z-AisoK tripeptides (with Z being either lipoyl-lysine (bulky) or glutaryl-lysine (negatively charged)). Indeed, from these screens we isolated two OppA variants (Z1 and Z2) with enlarged pockets. Genomic integration of these variants into *E. coli* strains led to efficient uptake of Z-AisoK tripeptides, displaying various bulky or negatively charged Z residues.

Importantly, using these strains, site-specific incorporation of Z residues from Z-AisoK supplementation significantly outperforms direct Z supplementation, as shown for a variety of Z-AisoK tripeptides bearing bulky Z residues. This allows for efficient genetic encoding of ncAAs such as Lipoyl-lysine and coumarin-lysine, ncAAs that previously have shown negligible incorporation efficiencies.

The new data can be found in the paragraph: *Generalized ncAA uptake using Z-AisoK tripeptides* and in new Fig. 5, as well as new Figs. S21-23

We see our OppA evolution approach as a generalizable platform that allows for the engineering of customized ncAA transport, independent of the availability of a corresponding aaRS/tRNA pair. Accumulating otherwise impermeable ncAAs inside cells will now allow to start PyIRS evolution campaigns for these ncAAs. Our approach thereby provides a practical and modular strategy to dissect ncAA uptake from its co-translational incorporation, complementing recent efforts in decoupling and optimizing individual GCE-steps towards synthesis of non-canonical biopolymers in *E. coli* (Soni. C. et al., 2024, *ACS Central Science* 10 (6), 1211-1220, Dunkelmann, D.L. et al., 2024, *Nature* 625 603–610).

A few minor things:

- I'd recommend providing a summary figure in the SI showing structures of all ncAAs in one place,

Thanks for this suggestion. We agree and we have now added an SI figure (S24) showing structures of all ncAAs (all used di- and tripeptides).

- line 69: When discussing biosynthetic generation of ncAAs, authors may consider including the following references: 10.1016/j.chempr.2020.07.013, 10.1002/anie.202014540

Thanks, we have included these references in the manuscript.

- line 84: When discussing lysine dipeptides, the authors may consider including the following reference: [10.1021/acs.biochem.4c00530](https://doi.org/10.1021/acs.biochem.4c00530)

Many thanks we have added this new reference to the manuscript.

- line 166: Shuttle?

Thanks, we have changed shuffle to shuttle.

Referee #2 (Remarks to the Author):

lype et al. describe the engineering of a bacterial membrane transport system that facilitates import of tripeptides bearing non-canonical amino acids (ncAAs) into bacterial cells. This system if broadly applicable overcomes important bottlenecks associated with poor ncAA encoding into target proteins due to poor cellular import of ncAAs. The authors show clearly that the encoding efficiency of many ncAAs is significantly improved using this transporter-based strategy, including 11 novel ncAAs. They also engineer a variant of the OppA with increased affinity for the tripeptides resulting in improved encoding at lower peptide concentrations with a variety of media. The manuscript is well written and organized, and the data presented clearly and convincingly. Overall, while the study is clear and thorough, I believe the results presented here are of moderate to high interest to researchers in this specific discipline, and of limited interest to a broader audience based on the major comments below. My support enthusiasm would be higher if truly important or novel functional groups that have not been encoded before were enabled by this technology.

We thank the reviewer for carefully reading our manuscript and for their valuable comments and suggestions. In response, we have substantially expanded our approach by further engineering Opp-based transport systems to support generalizable and customizable ncAA uptake for various 'Z' ncAAs within a Z-XisoK scaffold. This enhanced system enables efficient import and incorporation of various challenging ncAAs – particularly those that previously could not be incorporated due to poor cell permeability. Below, we address each of the reviewer's specific comments in detail.

Major comments:

- 1) As noted by the authors in the introduction, functionalizing ncAAs into pro-peptides for improved transport and encoding has previously been proposed and demonstrated, both by themselves as well as others (refs. 28, 30-32). Presented here is largely an improved adaptation of these same concepts which reduces the significance of the work.

We thank the reviewer for this observation and the opportunity to clarify the novelty and scope of our approach:

Clarification on reference 28 (now reference 35 in the revised manuscript):

Luo, X. et al (2017 *Nat Chem Biol* 13, 845–849) report the use of a dipeptide (a phosphotyrosine mimic is coupled to lysine) to increase suppression efficiency of a phosphotyrosine mimic. This was done without a clear understanding of whether the observed effect was due solely to improved solubility of the phosphotyrosine mimic or whether it involved

active import of the dipeptide. The dipeptide permease Dpp is mentioned in the manuscript, but no data supporting this claim are shown and it is unclear whether such a method is generalizable. So, while the concept of a propeptide-based transport strategy has sporadically been proposed in the past, our work significantly advances the field by providing a rational and generalizable framework for engineering ncAA import. Specifically, with the added data on OppA engineering for challenging ncAAs Z (within a Z-XisoK tripeptide, see response to comment 4 below) our approach allows the design of customized import systems for different ncAAs that have historically been refractory to cell permeability.

We have revised the relevant sentence in the introduction to better reflect these distinctions and address the reviewer's concern.

Clarification on References 30-32 (now references 36-38 in the revised manuscript):

These refer to our previously published work involving the ncAA AzGGisoK (in previous papers also named AzGGK). We respectfully clarify that AzGGisoK does not function as a propeptide in the same sense as the G-XisoK or Z-XisoK scaffolds discussed in the current manuscript. Instead, the azide group in AzGGisoK is installed to block the GG nucleophile preventing sortase-mediated transpeptidation before decaging. Protection via azide makes AzGGisoK also refractory towards peptidase processing, thereby not leveraging the propeptide strategy described here. AzGGisoK is incorporated into POIs via a specifically evolved aaRS/tRNA pair and subsequently the azide is reduced on-protein via Staudinger reduction upon addition of a phosphine, rendering the POI competent for sortylation.

Our OppA:G-SisoK structure shows that the charged alpha-amine of the Z residue in Z-XisoK tripeptides is necessary to bind to OppA (via an electrostatic interaction with D445 that is retained in all evolved OppA variants and that excludes binding of the smaller X-isoK dipeptides) and we have no evidence that AzGGisoK ncAAs would engage with wt-OppA.

Thus, we respectfully note that AzGGisoK differs both mechanistically and functionally from the tripeptide-based scaffolds developed in the current study.

We have accordingly revised the relevant passage at the beginning of the Results and Discussion part to clarify this distinction.

- 2) The ncAAs encoded here with this system are also restricted to the XisoK structural scaffold and so the scope of ncAAs whose encoding would benefit from this transporter-assisted system appears quite narrow and restrictive. This particular point is not discussed or acknowledged by the authors.

In our original manuscript, we demonstrated robust and high-yielding incorporation of eleven – previously inaccessible – XisoK-type ncAAs bearing functionalities such as bioorthogonal handles, novel PTMs, chemical crosslinkers and photocrosslinkers, as well as functionalities for chemoenzymatic ligations. These functional groups represent novel PTMs and cover most of the chemically useful groups that have been incorporated via genetic code expansion to date (for specific advantages of XisoK ncAAs, please see our response to Comment 3).

As commented in the beginning and noted in our response to Comment 4 below, we have in response to insightful comments from Reviewers #1 and #2 now considerably broadened the scope of our approach by engineering efficient uptake for diverse Z-XisoK tripeptides, with Z representing diverse ncAAs, including those with bulky or negatively charged groups known to exhibit poor or negligible cell permeability. By demonstrating the wide scope of ncAAs that can be accommodated at the Z position of the generalized Z-XisoK scaffold, we believe we have addressed the original limitation of our system.

- 3) While the authors are correct that many of the ncAAs encoded with this new system are structurally novel, these ncAAs have chemical functionalities (e.g. alkynes, diazirines, proximity crosslinkers, etc) that are commonly and efficiently encoded using alternative ncAA structures and so the practical advance isn't as obvious.

We thank the reviewer for this comment. While it is true that certain chemical functionalities - such as alkynes and diazirines - can be encoded via alternative ncAAs, XisoK-based ncAAs, offer several distinct and practical advantages, which we outline below:

- (1) Ease of synthesis and efficient uptake: The used tripeptides (G-XisoK and Z-XisoK) can be easily synthesized via solid-phase-peptide synthesis (SPPS) using commercially available building blocks. As they are actively pumped into the cell, one requires only μM concentrations of tripeptide to get higher protein yields than what is typically achieved with mM concentrations of previously used ncAAs, offering an economical and scalable strategy for producing ncAA-bearing proteins.
- (2) Superior incorporation yields: There have been previous attempts at directly incorporating four out of the eleven XisoK-type ncAAs described in the literature (CisoK, PisoK, PrgisoK and TisoK) without using the here described G-XisoK-based uptake strategy. Due to lack of ncAA uptake, incorporation of these ncAAs was extremely inefficient, even when using mM concentration of the ncAAs and despite multiple efforts to evolve efficient PylRS variants for these XisoK ncAAs. In contrast, our strategy achieves incorporation yields of these XisoK ncAAs that are close to wt expression levels when the corresponding G-XisoK tripeptides are supplied (see comparison in Figure S8 and Fig 3e).
- (3) Scalable high-yield expression of modified POIs: We demonstrate the efficiency of our system by preparative large-scale production and purification of a nanobody with an ncAA bearing a propargyl moiety. Expression in the presence of tripeptide G-PrgisoK (using the IsoK12 strain and 2-YT media) resulted in similar protein yields (44 mg/L) as obtained for wt expression (41 mg/L) and exceeded yields from a previously reported optimized alkyne-bearing ncAA/PylRS combination (27 mg/L; see Fig. S19).
- (4) Studying novel PTMs: Several XisoK ncAAs, where X corresponds to a natural amino acid, mimic recently discovered PTMs (Zang, J. et al. 2023 *Nat Struct Mol Biol* **30**, 62–71, He, Xia-Di et al. 2018 *Cell Metabolism*, Volume 27, Issue 1, 151 - 166) that remain poorly understood. To date, the effects of these modifications – and their corresponding reader, writer, and eraser enzymes – have not been studied, primarily due to the lack of methods for producing homogeneously and site-specifically modified POIs. Our approach now enables such investigation by allowing efficient production of such modified POIs.
- (5) Favourable physicochemical properties of XisoK ncAAs for incorporation at protein surfaces: Unlike most previously used ncAAs, XisoK-type ncAAs have positively charged side chains under physiological conditions. This makes them ideal moieties for display on protein surfaces, which one requires for many GCE applications, such as bioconjugation and crosslinking. Installing an ncAA with a charge minimizes risks of unwanted protein aggregation or degradation – common issues with displaying hydrophobic ncAAs, as typically used for bioorthogonal labelling approaches.

- (6) Enhanced folding and suppression efficiencies: Genetically encoding hydrophobic ncAAs may also lead to misfolding of the targeted POI. We show the advantage of incorporating ncAAs bearing a positively charged amino group in their side chain by efficiently suppressing a panel of more challenging target proteins (interleukin IL-2, Ran GTPase-activating protein RanGAP, human growth hormone hGH etc) and compare incorporation efficiencies of XisoK ncAAs to the gold-standard ncAA Bock as well as wt-protein expression levels. In all cases we get considerably higher yields with XisoK compared to Bock (See Fig. 4i and Fig. S18).

In response to this helpful comment, we have revised the relevant text in the manuscript to better emphasize these practical advantages and clarify the value of our XisoK toolbox in comparison to existing systems.

- 4) On this note, while the paper is largely focused on demonstrating installation of isoK based ncAAs, I find most interesting their finding that the amino acid in the first position of peptide could be altered from Gly to an ncAA, and this ncAA can be genetically encoded. This implies that ncAAs installed at the first position of the pro-peptide are not restricted to the XisoK structural scaffold (acetyl-lysine in this case) and they can still be encoded. Evaluating the scope of ncAA structures that can be tolerated at this position without the XisoK restriction would make this strategy much more versatile.

Many thanks for this suggestion. In the revised manuscript we have addressed this limitation and we have explored the scope of ncAAs that can be accommodated as Z residues in a generalized Z-XisoK scaffold.

For this we have synthesized a panel of 14 tripeptides based on a generalized Z-AisoK scaffold, where the N-terminal residue Z was varied to include either natural amino acids or ncAAs with diverse side chains, specifically, those that are bulky or negatively charged and thus typically exhibit poor cell permeability. To assess uptake and Z-AisoK processing efficiency, we monitored AisoK incorporation into sfGFP-N150TAG using wt *MbPylRS/PylT*. Of the 14 tested tripeptides, eight showed efficient uptake and processing in wt *E. coli* K12 cells, as evidenced by robust AisoK incorporation and we confirmed that Z-AisoK uptake was in all cases dependent on wt OppA.

In contrast, tripeptides with bulkier or negatively charged Z residues resulted in significantly reduced or abolished AisoK incorporation, indicating inefficient import and/or cleavage by native OppA and endogenous peptidases. Given our previously established platform for OppA evolution, we explored whether AisoK incorporation could be leveraged as a selection tool to evolve new OppA variants capable of transporting these more challenging substrates. Guided by the crystal structure of the wt OppA:G-SisoK complex, which we solved during revision, we targeted residues surrounding the glycine-binding pocket for site-saturation mutagenesis to accommodate bulkier or negatively charged Z residues. We subjected a corresponding library to multiple rounds of FACS-based enrichment using two Z-AisoK tripeptides: one with lipoyl-lysine (bulky) and the other with glutaryl-lysine (negatively charged) as the Z residue. From these screens, we isolated two OppA variants, Z1 and Z2, which possess enlarged binding pockets.

Upon genomic integration of these OppA variants into *E. coli*, we observed efficient uptake of a range of Z-AisoK tripeptides bearing both bulky and negatively charged Z residues. Importantly, in these engineered strains, site-specific incorporation of Z residues via Z-AisoK

supplementation significantly outperformed incorporation from direct Z supplementation, as demonstrated across diverse examples of bulky Z groups.

These results and supporting data are presented in the section *Generalized ncAA uptake using Z-AisoK tripeptides* and in Fig. 5, as well as Supplementary Figures S21–S23.

We consider our OppA evolution strategy a broadly applicable platform for engineering customized ncAA transport, independent of the availability of a matching aaRS/tRNA pair for Z-incorporation. This modular approach decouples cellular uptake from translational incorporation, enabling practical dissection and optimization of individual steps in GCE (Soni, C. et al., 2024, *ACS Central Science* 10 (6), 1211-1220, Dunkelmann, D.L. et al., 2024, *Nature* 625 603–610).

Notably, our strategy presents the first successful approach for intracellular delivery of the negatively charged ncAAs succinyl-lysine and glutaryl-lysine. By enabling their accumulation in the *E. coli* cytosol, our method opens the door for targeted PylRS engineering campaigns to incorporate these challenging residues.

Minor issue

1) The MS data is provided for all appropriate new encoding technology. The mass axis is broad showing that there are no significant changes in mass but it is so broad that small changes in mass that would result from ncAA degradation, reaction and or GCE miss encoding evens from near cognate suppression are not detectable. The claims on efficient encoding of ncAAs would be supported if supplemental information contained MS data with an axis scale enabling detection of salt ions and small MW changes from the predicted mass.

We have added m/z data for all our MS analyses in the Supplementary Information (Fig. S25). These data confirm that incorporation of XisoK ncAAs is clean and specific. The only exception is CisoK, for which we observe minor side reactions with cellular metabolites such as pyruvate or alpha-ketoglutarate, consistent with previous reports and dependent on the exact methoxy-amine protection conditions.

In each case, we also selected a representative X axis that allows us to monitor potential mis-incorporation, degradation or ncAA instability. For example, in Fig. S23, which represents some of our new data for Coumarin-Lysine (CouK) incorporation using tripeptide 6 (with CouK as residue Z) we observe partial decaging of CouK to K, visible as a second peak with < 10% intensity. Also, for incorporation of CIAisoK (Fig. S10) we see a second visible MS peak likely corresponding to chlorine elimination.

Referee #3 (Remarks to the Author):

This study addresses the low efficiency in incorporating non-canonical amino acids (ncAAs) by genetic code expansion. Low yields and limited access to these specialized amino acids have hindered a wider adoption of such methods for producing protein with potential medical and industrial uses. In this study, a new approach is introduced that combines a well-known “pro-peptide strategy” with a well-characterized bacterial oligopeptide ABC transport system OppA to boost intracellular levels of ncAAs. The researchers identified inadequate cellular uptake of ncAAs on *E. coli* as a primary barrier and tackled it by repurposing a well-

characterized oligopeptide ABC transport system to piggyback ncAAs within tripeptide scaffolds with isopeptide-linked ncAAs.

Strengths: This method allowed the efficient incorporation of eleven previously inaccessible ncAAs with diverse functionalities. Additionally, the authors engineered an *E. coli* strain with a transporter optimized for tripeptide-linked ncAAs, achieving incorporation efficiencies comparable to wildtype protein levels. The authors adapted the system for co-transporting two distinct ncAAs, enabling their simultaneous incorporation. This work highlights the importance of optimizing ncAA uptake in *E. coli*, paving the way for scalable production of modified proteins and expanding protein chemistry without relying on passive membrane permeability.

We thank the reviewer for their thoughtful assessment, positive feedback, and constructive suggestions. We address all the comments in detail below.

Weaknesses: The important limitation of the study is that the system is currently restricted to *E. coli*. There are doubts about its potential to adapt to mammalian cells.

We agree with the reviewer that our system is currently limited to *E. coli* and we have adjusted the corresponding sentence in the revised manuscript.

That said, we would like to emphasize that achieving high-efficiency and reliable ncAA incorporation (including multi-site incorporation) in *E. coli* remains a critical milestone. *E. coli* continues to serve as the central workhorse for much of protein biochemistry, synthetic biology, and industrial biotechnology. Many foundational discoveries and applications in genetic code expansion, protein engineering and synthetic genomics are developed and validated in *E. coli*, underscoring its enduring relevance.

Importantly, inefficient ncAA uptake has long represented a major bottleneck for genetic code expansion applications in bacteria and a more widespread use in both academic as well as industrial settings. Our study provides a practical and modular solution to this challenge. Especially with the new data on customized transporter engineering for virtually any kind of ncAA, including those that are known to be cell-impermeable, we believe that this advance will have immediate utility in accessing modified proteins, improving yields and reproducibility of ncAA-containing proteins in *E. coli*, which is essential for both fundamental studies and scalable production.

Moreover, we see strong potential for synergy with recent developments in genomically recoded *E. coli* strains (e.g. Fredens, J. et al. 2019, *Nature* 569, 514–518; Grome, M.W. 2025, *Nature* 639, 512–521), which enable multi-ncAA encoding and are ideally suited for the biosynthesis of non-canonical polymers.

We have revised the manuscript to clarify this point in the updated *Conclusion and Outlook* section.

In addition, all test proteins (e.g., eGFP, histone, ubiquitin, etc.) have been used previously and have been shown to be effective in amber suppression. No new targets were tested or validated in this study. Furthermore, no new biological question has been addressed.

We appreciate the reviewer's comment and agree that applying GCE to new biologically relevant proteins is important for demonstrating broader utility. In the revised manuscript, we have expanded the scope of our study to include therapeutically relevant and more

challenging protein targets. Specifically, we show successful incorporation of XisoK ncAAs into a panel of more challenging and therapeutically relevant target proteins, such as human interleukin-2 (IL-2), human growth hormone (hGH), and Ran GTPase-activating protein (RanGAP). Many of these are poorly suppressed in presence of the gold-standard ncAA BockK and other hydrophobic ncAAs, while with our approach for incorporating XisoK ncAAs we typically obtain close to wt expression efficiencies, highlighting the robustness and benefits of our system.

In total, we establish our method on 19 distinct proteins and incorporation sites, using 15 chemically diverse ncAAs (XisoK and Z). To our knowledge, this represents the broadest validation of ncAA incorporation efficiency and versatility in any genetic code expansion study to date. Furthermore, we also show efficient double and triple incorporation of ncAAs. While some of the standard test proteins (e.g. GFP, ubiquitin) have indeed been previously used in amber suppression experiments, our work is the first to demonstrate their efficient modification with expression levels par to wt expression levels using ncAA delivery via engineered transport.

The new results regarding suppression and expression of therapeutically relevant proteins can be found in Fig 4h and 4i, as well as in Fig. S18.

Regarding the concern that no new biological question/application is addressed, we respectfully note that the primary aim of our study is to introduce and validate a novel technology platform – a modular and generalizable strategy for active ncAA import and incorporation into protein in *E. coli*. This technological advance, which overcomes a key limitation in bacterial genetic code expansion, lays the groundwork for addressing future biological questions using this toolbox. Our approach thereby provides a practical and modular strategy to dissect ncAA uptake from its co-translational incorporation, complementing recent efforts in decoupling and optimizing individual genetic code expansion steps towards synthesis of non-canonical biopolymers in *E. coli* (Soni. C. et al., 2024, *ACS Central Science* 10 (6), 1211-1220, Dunkelmann, D.L. et al., 2024, *Nature* 625 603–610).

Alternatively, isopeptide-linked tripeptides could be shuttled into cells via other hydrophobic, cleavable modifications.

We agree that ncAAs could, in principle, be shuttled into cells via hydrophobic, cleavable modifications. This approach has however critical limitations. Such a strategy would still primarily depend on passive diffusion, which is inherently inefficient and non-specific. Furthermore, addition of hydrophobic moieties may limit solubility of corresponding ncAAs, as well as also synthetic accessibility.

In contrast, our approach relies on highly soluble tripeptides that can be easily accessed from commercially available building blocks through solid-phase-peptide synthesis. By leveraging the Opp system's active transport mechanism – supported by new data on Z-XisoK tripeptides added during revision, we demonstrate that customized uptake pathways can be engineered for a wide range of otherwise impermeable ncAAs.

Notably, some of the newly added Z-XisoK tripeptides (e.g. 5, 6, 12, 13 in Figure 5a bearing lipoyl, coumarin or Cbz moieties) contain bulky and hydrophobic Z-residues. However, the presence of these features alone does not guarantee uptake (see Fig. 5 and Figures S22 and S23). We demonstrate that the uptake of these tripeptides is clearly dependent on OppA.

Critically, efficient uptake is only achieved upon engineering tailored OppA variants specifically designed to accommodate such bulky tripeptides.

While it is obviously plausible that other active uptake systems could be engineered for peptide/ncAA transport, merely appending a hydrophobic group may prove insufficient. Effective intracellular release of such a hydrophobic group may necessitate precisely engineered, system-specific cleavage mechanisms.

Finally, these isopeptide-linked peptide conjugates have recently been studied by the authors (ref. 30-32). The present study is based on the observation that the N-terminal glycine is cleaved off in the cytosol and that tripeptides G-AisoK in contrast to the dipeptide AisoK display a high level of amber-suppressed sfGFPN150TAG production.

We would like to respectfully clarify a misunderstanding related to our previously published work involving the ncAA AzGGisoK (in previous papers (ref 30-32, in revised manuscript ref 36-38) also named AzGGK). Contrary to the classification in the current manuscript, AzGGisoK does not function as a propeptide in the same sense as the G-XisoK or Z-XisoK tripeptides developed in this manuscript. Instead, the azide group in AzGGisoK is installed to block the GG nucleophile preventing sortase-mediated transpeptidation before decaging. Importantly, protection via azide makes AzGGisoK also refractory towards peptidase processing thereby not leveraging the propeptide strategy described here. AzGGisoK is incorporated into POIs via a specifically evolved aaRS/tRNA pair and subsequently the azide is reduced on-protein via Staudinger reduction upon addition of a phosphine, rendering the POI competent for sortylation.

The OppA:G-SisoK structure, which we solved during revision, shows that the charged alpha-amine of the Z residue in Z-XisoK tripeptides is necessary to bind to OppA (via an electrostatic interaction with D445 that is retained in all evolved OppA variants and that excludes binding of the smaller XisoK dipeptides) and we have no evidence that AzGGisoK ncAAs would engage with wt-OppA.

Thus, we respectfully note that AzGGisoK differs both mechanistically and functionally from the tripeptide-based scaffolds featured in the current study.

We have revised accordingly the relevant section at the beginning of the Results and Discussion part to clarify this distinction.

In addition to the peptide transport system OppA, it would be a decisive step forward if the authors could identify the peptidase that is essential for the removal of the N-terminal glycine. Instead of single-gene knockouts the approach is biased on a potential target list, which is unfortunately not provided in the manuscript. Double/triple knockouts or transposon mutagenesis should be considered to identify unknown peptidases for the cleavage of the N-terminal glycine.

We thank the reviewer for this suggestion. In response to this concern, we expanded our investigation to include double and triple knockouts of known peptidases in *E. coli*. A literature survey identified pepN as the major aminopeptidase in *E. coli* with promiscuous substrate specificity, including activity toward different N-terminal residues. Based on this, we generated double and triple knockout strains in a pepN deletion background using a CRISPR-Cas12a-based genome editing platform for *E. coli*. We generated, starting from a pepN KO strain, three double KOs by individually deleting pepA, pepB or pepT, as well as a triple KO strain

that lacks *pepN*, *ypdE* as well as *ypdF*. Interestingly, only in *E. coli* cells with both *pepN* and *pepA* deleted ($\Delta pepN/pepA$), we observed a drastic reduction in GFP expression in the presence of G-AisoK. Importantly, amber suppression yields in presence of other ncAAs were not affected in this KO. GFP expression yields could be rescued to wt K12 level, when this double KO was rescued with constitutive plasmid-based *pepA* or *pepN* expression, indicating that either peptidase is sufficient for G-AisoK processing. These findings strengthen the link between peptide transport and intracellular processing and provide a foundation for future identification of additional, potentially redundant, peptidases.

These novel data are presented in Fig. 2d and Fig. S4 and the manuscript text has been adjusted accordingly.

Specific points:

- 1) Line 74-76: The author hypothesized that their approach holds the potential of being widely applicable; however, how can this be expanded to mammalian cells? The authors could easily test whether some of the isopeptide-linked tripeptides are taken up by mammalian cells.

We appreciate the reviewer's suggestion.

We would like to stress that GCE and site-specific incorporation of one or more ncAAs into proteins expressed in prokaryotic systems represent indispensable tools in biological and biotechnological research. Making such tools vastly more efficient and widely applicable (customized engineering for any kind of Z residue in Z-XisoK scaffolds, ease of implementation, straightforward, cheap tripeptide synthesis) is in our opinion an endeavour with great interest to the general biochemical audience as it has the potential to provide immediate impact to many academic and industrial labs. As mentioned above, we see our platform being particularly powerful when used in synergy with recent advances in allowing multi-ncAA encoding and non-canonical polymer synthesis in engineered *E. coli* strains (Robertson W.E. et al. 2021 *Science* 372,1057-1062. Grome, M.W. 2025, *Nature* 639, 512–521).

We have revised the manuscript to clarify this point in the updated *Conclusion and Outlook* section.

- 2) Figure 1 and throughout the manuscript: The ratios between full-length and truncate proteins should always be quantified in biological triplicates.

We confirm that all SDS-PAGE experiments shown were performed in biological triplicates and the representative gels shown reflect consistent trends observed across independent replicates. While quantification via densitometry is in principle possible, we find that

quantification of Coomassie staining lacks the reliability, sensitivity and linearity needed for meaningful quantitative ratio comparisons between full-length and truncated protein, especially for gels exhibiting the whole *E. coli* proteome background. Since our conclusions rely on clear qualitative differences, we chose not to include numerical values that could give a misleading sense of precision.

3) Line 181-189: This part can be shortened as this transport system is well described. It should be explained in the SI. In several case the manuscript contains very long introductions into side topics, which are well-characterized in literature.

Thanks for the suggestion. We agree and have shortened the corresponding part.

4) Line 201-213: Similarly, this part can be shortened.

Thanks for the suggestion. We have changed this part, as we have now solved a X-tal structure of G-SisoK bound to OppA (see response to comment 9 below).

5) Figure 2b and 2c should be removed as the mechanism of peptide translocation is not supported by the experimental data und well-described in literature. In addition, the structure of the import system OppABCDF does not provide additional support for the present study.

We appreciate the reviewer's comment and acknowledge that the general mechanism of uptake of endogenous peptides via the Opp transporter is well established in the literature. It is however novel regarding isopeptide-linked tripeptides bearing ncAAs. Importantly, during the revision, we have identified the key peptidases (PepN and PepA) responsible for processing of these tripeptides. This additional finding strengthens our mechanistic model and, in our view, justifies the inclusion of a scheme summarizing the proposed pathway for uptake and processing of G-XisoK tripeptides.

We have also opted to retain Fig. 2b, which illustrates the overall architecture of the OppABCDF transporter. This figure is intended to benefit readers who may not be familiar with bacterial peptide import systems—especially given the interdisciplinary nature of the journal and the broad interest in genetic code expansion strategies.

6) Line 255: It is not clear how the tests were done, in vitro with purified protein or in cellulose or in lysate.

We think this comment refers to the sentence describing the screening for PylRS variants for incorporating some of the XisoK ncAAs. This is done in living cells using different PylRS variant-encoding plasmids.

We have adjusted the sentence to make it clearer.

7) Figure 3b: Why is XisoK equally well incorporated as G-XisoK?

We think this comment relates to slightly better incorporation of dipeptides TisoK and pLisoK out of the eleven XisoK ncAAs shown in Fig. 3e. For these two XisoK derivatives, we observe a certain level of incorporation from the corresponding dipeptides, suggesting that these two XisoK ncAAs may be taken up more efficiently compared to other XisoK derivatives. Furthermore, their corresponding PylRS variants (wt-MbPylRS and MaPylRS(H227I/Y228P)) may also show extraordinary activity towards these substrates. However, it is important to note that for all XisoKs - including TisoK and pLisoK - incorporation is markedly improved when cells are supplemented with the corresponding G-XisoK tripeptides. When

supplementing G-XisoK tripeptides, we observe wt expression yields and minimal accumulation of truncation products in all cases, highlighting the benefit of active uptake and processing through the G-XisoK route.

8) Figure 3: The cross-linked bands should be quantified.

We agree that quantification of cross-linked bands is technically straightforward. However, in our view, such quantification would not provide additional insights in the context of our study. The efficiency of crosslinking with both diazirine and chloro-alanine moieties is highly position-dependent and can vary substantially based on the local environment of the incorporation site. As our current data serve to demonstrate the feasibility of crosslinking using XisoK derivatives rather than to systematically compare crosslinking efficiencies, we believe that quantifying these individual cases would not yield meaningful or generalizable conclusions.

Moreover, both diazirine and haloalkyl moieties are established in the literature as effective crosslinkers. Quantitative comparisons would only become informative in a broader study evaluating multiple incorporation sites, varying crosslinking conditions (e.g., UV exposure time for diazirines, or different nucleophilic partners for chloro-alanine). For these reasons, we have opted to present the current results as qualitative demonstrations of crosslinking compatibility.

9) Line 407-419: As the crystallization of OppA is established and well-described, the binding properties of the evolved OppA-ev2 and the structural consequence of the seven mutations should be examined via an experimental x-ray structure in a G-SisoK/OppA-ev2 complex. Just mapping the seven mutations found in OppA-ev2 onto a published OppA structure liganded to a linear tripeptide can hardly explain the significant substrate preference for isopeptide-linked G-SisoK versus GSK.

We thank the reviewer for the suggestion. To address this, we initiated extensive crystallization campaigns aimed at obtaining diffracting crystals of both wt-OppA and the evolved variant OppA-iso (previously called OppA-ev2 in the manuscript) in complex with G-XisoK substrates. These efforts were carried out in collaboration with Prof. Michael Groll (TUM), whose expertise in crystallography significantly supported the experimental design and screening strategies. While we successfully obtained high-resolution crystal structures of wt-OppA in complex with G-SisoK, all crystallization attempts involving the evolved OppA-iso variant failed, despite systematic variation of crystallization conditions and extensive optimization.

Nevertheless, the structure of wt-OppA:G-SisoK as well as mutational experiments provided critical insight:

As described in the text and in Fig 3b and Fig. S5c, G-SisoK forms an extensive network of direct interactions with OppA, involving both its backbone and termini. The N-terminal glycine is tightly anchored through a coordinated set of hydrogen bonds and electrostatic contacts. Specifically, its protonated α -amine forms a strong interaction with D445, whereas the C-terminal carboxylate of G-SisoK is stabilized by hydrogen bonds involving the side chains of R439, H397, and N392 (Fig. 3b and Fig. S5c).

To experimentally validate the functional relevance of this interaction network, we performed amber suppression experiments using $\Delta oppA$ K12 cells expressing either wt-OppA or point mutants. Notably, the D445A variant, which disrupts the interaction with the N-terminal α -amine of G-SisoK, completely abolished G-SisoK-dependent suppression, confirming the essential role of this interaction. In contrast, a R439A variant had only minimal effect on substrate uptake, indicating redundancy in C-terminal recognition, an observation supported by the positioning of H397 and N392 in the crystal structure.

Interestingly, among the seven mutations in OppA-iso, only R439Q lies in proximity to the G-SisoK binding site. Given that an R439A OppA mutant still efficiently supports G-SisoK uptake in 2-YT media, this substitution appears to maintain binding, likely due to compensatory interactions. Two nearby hydrogen-bond donors H397 and N392 are clearly resolved in the crystal structure and well positioned to interact with the lysine carboxylate in G-SisoK, likely stabilizing the complex even in the absence of a positively charged side chain at position 439. In contrast, for linear tripeptides such as GSK, the lysine carboxylate interacts primarily with R439 (see Fig. S15) Accordingly, its mutation diminishes binding affinity of linear tripeptides, as supported by our structural analysis and affinity measurements.

Interestingly, one of the OppA variants, evolved for recognition of the Z-XisoK tripeptide 15 (see Fig. 5a) contained a non-programmed R439H mutation, providing further evidence that an arginine residue at this position is not needed for binding of our isopeptide-linked tripeptides.

In summary, our combined structural, mutational, and functional analyses provide compelling mechanistic insights into how wt-OppA discriminates between G-XisoK tripeptides and XisoK dipeptides and how OppA-iso may discriminate between linear and isopeptide-linked substrates. We believe these findings meaningfully advance the understanding of substrate recognition in the OppA scaffold and lay the groundwork for future directed evolution campaigns guided by structural and biophysical data.

The new data can be found in Fig 3b and 3c, as well as Figs. S5b, S5c and S15 and text changes have been marked in grey.

10) Line 443-445: To strengthen the conclusion, the authors should quantify the intensities.

We thank the reviewer for this suggestion and would like to refer to our response to point 1 above. All experiments were performed in biological triplicates, and we consistently observed clear qualitative trends. We believe that assigning precise numerical values to Coomassie-stained bands within the complex background of the *E. coli* proteome exceeds the resolution and reliability of the techniques. For this reason, we feel it is more scientifically appropriate to report the results qualitatively.

The revised manuscript reflects this approach, and the updated text now reads as follows: *“IsoK12 outperformed K12 in single, double, and triple amber suppression. For single and double suppression, protein expression levels were comparable to wt-H3. Even for triple suppression, we observed significant amber suppression for the G-AisoK/IsoK12 combination. In contrast, only minute amounts of doubly- and triply-suppressed proteins were obtained with BockK.”*

11) Line 497-499: Frequently, the authors state the significantly higher protein yields were achieved by their approach by using isopeptide-linked ncAAs. However, these statements must be corroborated by quantitative data on at least three biological replica and different model systems.

We appreciate the reviewer's request for quantitative support and would like to reiterate our explanation provided above (see response to points 1 and 10). All key experiments were performed in biological triplicates, and we observed consistently strong and reproducible differences in protein yields when using isopeptide-linked ncAAs across multiple model proteins. In all cases, the improvements are not marginal but substantial, making the enhanced performance of our approach qualitatively evident.

The inherent limitations of protein gel analysis within the complex *E. coli* proteome background make exact numerical comparisons imprecise. Rather than overstate accuracy, we have chosen to report results using qualitative language that faithfully reflects the magnitude of improvement observed.

That said, we have ensured that the trends are supported by data from multiple proteins, multiple incorporation sites and multiple ncAAs (e.g., H3, RanGAP, hGH, IL-2, nanobodies), which strengthens the generality of our conclusions. Moreover, the qualitative differences observed in Coomassie-stained gel intensities are further supported by quantified protein yields from purified proteins—for example, in the case of the nanobody shown in *Fig. S18*.

12) Line 570-573: The system is currently limited to *E. coli*. There are doubts about its potential to adapt to mammalian or yeast cells.

Thanks for addressing this point. We have deleted the corresponding sentence. See our response to comments above.

13) The authors speculate the availability of ncAA is limited by passive diffusion. This statement needs to be carefully discussed and reconsidered. The novelty of the approach is hampered as similar strategies were applied using the dipeptide transporter Dpp.

We thank the reviewer for this insightful comment. In response, we have revised the relevant section of the introduction to more carefully frame the issue of ncAA uptake and clarify the rationale for our study. The revised text now reads:

“In typical GCE experiments, chemically synthesized ncAAs are added to expression media and enter cells either via passive diffusion or via promiscuous uptake through native amino acid transporters. Intracellular concentrations commonly remain, however, well below those in the surrounding medium.²¹ This limitation is particularly detrimental for aaRS/ncAA combinations that are operating below saturating conditions, where reduced intracellular availability directly lowers aminoacylation efficiency⁴.”

Regarding the novelty of our approach, we respectfully disagree that it is significantly diminished by earlier reports involving the Dpp transporter. Specifically, reference 28 in the original manuscript (reference 35 in the revised manuscript, Luo, X. et al, 2017 *Nat Chem Biol* **13**, 845–849) describes the use of a phosphotyrosine mimic conjugated to lysine to improve suppression efficiency. This is done without a clear understanding if it only helps increasing solubility of the phosphotyrosine mimic or if it leverages active import of the dipeptide. The dipeptide permease Dpp is mentioned in the manuscript, but no data supporting this claim are shown and it is unclear whether such a method is generalizable.

So, while we agree in general that a propeptide-based transport strategy has sporadically been proposed in the past, our study establishes a rationalizable and generalizable method for engineering ncAA uptake. Especially with the new data added during revision on OppA engineering for challenging ncAAs Z (within a Z-XisoK tripeptide, see response to comments raised by other reviewers and added data in manuscript (section *Generalized ncAA uptake using Z-AisoK tripeptides* and in *Fig. 5*, as well as Supplementary Figures S21–S23)), our approach enables the design of customized import systems for structurally diverse ncAAs that have historically been refractory to cell permeability.

In contrast to previous limited reports on the use of dipeptides for improved ncAA incorporation, our study represents the first systematic use and directed evolution strategy of an endogenous peptide transporter to achieve robust intracellular accumulation and efficient

site-specific incorporation of ncAAs at near wt expression levels, thus addressing a longstanding bottleneck in genetic code expansion.

We have revised the relevant sentence in the introduction to better reflect these distinctions and address the reviewer's concern.

14) The conclusion is more a summary of the results and slightly redundant. Instead, the conclusions should address the major limitation of the study and important future developments.

We thank the reviewer for this suggestion. Since we have considerably strengthened our approach for customized and generalizable ncAA uptake, we have also revised our conclusion and also address potential limitations and discuss future directions.

15) Supporting Information: Synthesis via solid phase peptide synthesis: Which orthogonal protection group chemistry was used to introduce the iso-peptide bonds on solid-phase synthesis. This is not stated.

Thanks for this comment. All di- and tripeptides were synthesized via solid-phase-peptide synthesis using Fmoc strategy. We now described the approach in more detail. Essentially, lysine protected with Boc at its α -amino group and Fmoc at its ϵ -amino group was loaded on the resin via its carboxy group. Fmoc deprotection with piperidine yielded the free ϵ -amino group, which was coupled to the second amino acid (X), protected at its α -amine either with Boc (for synthesis of XisoK dipeptides), or with Fmoc in case of tripeptide synthesis, followed by coupling of the final Z amino acid (protected at its α -amine as Boc). Final Boc-deprotection and resin cleavage yields the di/tripeptides. General side chain protection groups are used as is state of the art for Fmoc-synthesis.

We adjusted the method section accordingly and also added a representative scheme (Scheme S1 in Supplementary Information).

16) SI Table 4: The purity of the peptide should be given.

Adjusted. All di- and tripeptides were analysed by HPLC-MS. The expected mass values as well as observed M+H peaks are given in Table S4. According to HPLC-MS spectra, purities of these simple SPPS-synthesized peptides were routinely > 95%.

Minor points:

1) Line 85: Abbreviation for Dpp is missing.

Adjusted

2) Figure 1: Coomassie needs to be capitalized.

Thanks. Adjusted in all main and SI figures.

3) Line 162: ABC transporter should be written without hyphen throughout the entire manuscript.

Thanks. Adjusted throughout the manuscript.

4) Line 163 should read: In gram-negative bacteria.

Thanks! Adjusted.

5) Figure 2: It is not clear what the dots stand for. I assume the presence of Bock, AisoK and G-AisoK. Please explain in the figure legend.

This part of the figure has changed, and the corresponding gels were moved to the SI (SI Figure S5, proper labelling of conditions in different lanes is now given)

6) Line 259: no hyphen between MS-analysis.

adjusted

7) Line 496: sfGFP-N40Ack should be sfGFP-N40AcK.

Thanks! Adjusted.

8) Line 533: This information should be mentioned before.

We think this referred to the following sentence: We show very efficient encoding of eleven different XisoK ncAAs, seven of which have not been incorporated into proteins before, greatly enhancing the chemical space of recombinantly generated proteins via GCE.

Since with new Z-XisoK tripeptide uptake, we have expanded the manuscript, this part of the conclusion was updated and reads differently now. However, to explain:

The comment referred to the original eleven XisoK amino acids whose efficient incorporation is described in our manuscript. For four out of the eleven XisoK ncAAs (CisoK, PisoK, PrgisoK, TisoK) incorporation has been attempted in the literature. However, due to lack of ncAA uptake (see Fig. 3e and Fig. S8), incorporation efficiencies were extremely low even despite several attempts at PylRS engineering. In contrast, incorporation yields of these XisoK ncAAs were close to wt expression, when supplementing cells with G-XisoK (see comparisons in Figure S8 and Fig 3e for previous incorporation versus our incorporation). The other seven XisoK ncAAs have never been described in literature before.

1. Please reduce the overall length of the article as detailed below (see 'LENGTH'). You will probably need to move figures or panels to Extended Data or Supplement and shorten the manuscript text.

done

2. Please submit a revised title within 75 characters (including spaces) that is free of any punctuation marks like colons, exclamation marks, full stops or speech marks.

Title has been shortened

3. Please add references to the abstract.

done

4. The number of main text references should be 60 in total or less - currently there are 73. Also, please create a separate reference list for the methods with continuous numbering (i.e. do not start at 1).

We now have exactly 60 references. As there are no Methods references that are not part of the main references, we do not have a separate list.

5. Please remove the main figures from the article file and re-supply them individually in an acceptable format such as EPS, AI, PS, PDF, PPT, PSD or XLS (for graphs) with editable vector files.

Files are provided as high-resolution PDF files.

6. Please ensure all main figure legends are 300 words or less.

Done

7. Please reduce subheadings to 40 characters (with spaces) or less.

done

8. Please provide a data availability statement in the main text of the manuscript (see 'DATA AND CODE AVAILABILITY STATEMENTS' below). This should contain Accession Codes currently found in a different section.

done

9. Please provide an author contributions statement in the main text of the manuscript.

done

10. Please ensure that the text size in all figures is at least 5 pt Arial.

done

11. The methods are provided entirely in Supplementary Information file, these should go into the main manuscript file. For reference, the Word file should be in the order: title & front matter, summary, main text, references, methods, data (&code) availability statement, additional references (with continuous numbering), acknowledgements, author contributions, additional information, figure legends, extended data figure legends.

We have now put the most important methods into the main manuscript file and organised the word file in the suggested order.

12. Please make use of our Extended Data (ED) format, which is easier for readers to access. See 'EXTENDED DATA' below for details. Briefly, we allow up to 10 Extended Data items (tables and figures), which must be submitted in .jpg, .tif or .eps format. We recommend you convert the most important SI figures to ED.

We have now reorganised figures such that we have 8 Extended Data Figures.

13. Please provide the manuscript in .docx format. Currently it is only available in PDF format.
done

14. There are potential third-party rights issues in some figures (i.e. schematics, illustrations). It is your responsibility to obtain the right to use any items (figures, tables, images, videos or text boxes) that are reproduced (or adapted) from material for which you do not hold copyright and to give proper attribution to the creators of that work. This includes work that has previously been published elsewhere, but also templated from e.g. BioRender. Regardless if third-party material is included, please fill out our Third Party Rights Table and submit it with the revised manuscript. If you generated all illustrations yourself, this can remain empty (except for the author / manuscript number fields).

We confirm that there are no third-party rights issues, all figures were made from scratch

15. Please provide a supplementary information guide (see 'SUPPLEMENTARY INFORMATION' below).
done

16. As several figures contain chemical structures, please pay particular attention to the instructions under 'CHEMICAL STRUCTURE PRESENTATION' below.

All chemical structures in figures have been adapted to Nature ChemDraw Style

17. In the Competing Interests statement, for patents, relevant authors should be listed and the patent number stated, e.g. "X.X and Y.Y are named as inventors on a pending patent application for ZZZ (insert patent number)."

done

18. The data deposited to the PDB with the dataset identifier 9RD1 is currently not available, please ensure its timely public release.

We have now released the data.

19. There is a reference to source data files in data section of the reporting summary; however, no such file was found among the manuscript files.

We prepared and uploaded an excel file for all graphs (Source Data file)

20. Full scans of gel images have been bordered with a solid box-border. We instead ask that authors use a light-dotted line. Please revise.

This has been revised. Can be found in Supplementary Fig. 21.

21. For any Supplementary Figures, please check and confirm that:

- * If data is presented as bar charts, individual data points are shown using overlaid dot plots.
- * The n number (i.e. the sample size used to derive statistics) is provided and defined as a precise value (not a range), using the wording "n=X samples/cells/independent experiments" etc. where applicable.
- * Any chart axis, error bars, scale bars, symbols and colour scales are defined.
- * Any statistical tests used for data analysis are specified and exact p-values are provided either on the figures themselves, in the legend or in the Source Data file.
- * Wherever representative data such as micrographs are shown, the legend indicates how many times the experiment was repeated with the same results.

done

STATISTICS: When revising your manuscript, you should ensure that any statistical analysis used is sound and that it conforms to our guidelines. A collection of articles explaining the basics of statistical analysis and advice on how to best present it can be found here.
done

REPRODUCIBILITY: To ensure that the quality and transparency of methods and statistical reporting (as discussed here) are sound before the paper is published, we have reviewed your Reporting summary editorially. I have attached two documents: one listing specific issues related to your manuscript and one containing an annotated version of the Reporting summary. Please ensure that, as well as the more general points below, the points highlighted in the attached documents are addressed in full, both on these forms and within the manuscript. Both forms should be uploaded as a "Related Manuscript" file type. The Reporting summary will be published with your paper.

IMPORTANT. Check reporting summary and second pdf.

All points have been addressed in full and we uploaded the two documents

LENGTH: In print, biological sciences papers do not normally exceed 8 pages on average; the final print length, however, is at the editor's discretion. The typical length of an 8-page article with 5 modest (quarter-page) display items is 4300 words. If a composite figure (with multiple panels) must occupy at least half a page in order for all the elements to be visible, the text length may need to be reduced accordingly to accommodate such figures. Essential but technical details can be moved into the Methods or Supplementary Information (see below).

In this case, we ask that you shorten the paper so that it does not exceed 8 pages maximum in print. Currently, the main text has 6120 words, so substantial shortening will be necessary. We note that there is the possibility to include text in the Supplement, e.g. as 'Supplementary Discussion' or 'Supplementary Notes'. Please do not resubmit your paper until you feel the length has been appropriately reduced.

We have reduced the main text to 4270 words and also reduced the size of Figs. 3 and 4.

TITLE: Titles cannot exceed 75 characters (including spaces); they must not contain punctuation.
done

SUMMARY PARAGRAPH: Papers start with a fully referenced, bold paragraph, ideally of about 200 words, aimed at readers in other disciplines. Numbers, abbreviations, acronyms or measurements should be avoided unless essential. The summary paragraph consists of 2 to 3 sentences of basic-level introduction to the field; a brief account of the background and rationale of the work; a statement of the main conclusions (introduced by the phrase 'Here we show' or its equivalent); and a conclusion of 2 to 3 sentences putting the main findings into general context so it is clear how the results described in the paper have moved the field forward. A downloadable, annotated example is available here.

done

MAIN TEXT: If further introductory material is necessary, the main text can begin with up to 500 words of introduction expanding on the background to the work (some overlap with the summary is acceptable), before proceeding to a concise, focused account of the findings, and ending with 1 or 2 short paragraphs of discussion. Sections are separated with subheadings (up to 40 characters including spaces) to aid navigation.

Done

REFERENCES: As a guideline, most papers should include no more than 50 main text references; all additional references can be cited in (and listed after) the Methods section, as detailed below.

We reduced to 60 references as suggested above (point 4).

FIGURE LEGENDS: These should be listed sequentially after the main text references and not in the figure files. Each legend should begin with a brief title for the whole figure and continue with a short description of each panel and the symbols used. Legends should not exceed 300 words each. Each figure legend should contain, for each panel where relevant, the following information:

- * the exact sample size (n) for each experimental group/condition, given as a number, not a range;
- * a description of the sample collection allowing the reader to understand whether the samples represent technical or biological replicates (including how many animals, litters, cultures, etc);
- * a statement of how many times the experiment shown was replicated;
- * definitions of statistical methods and measures:
- * very common tests (e.g. t-test, simple Chi-square tests, Wilcoxon and Mann-Whitney tests) can be identified by name only, but more complex techniques should be described in the Methods;
- * whether tests are one-sided or two-sided;
- * whether there are adjustments for multiple comparisons;
- * the statistical test results (e.g., P values);
- * the definition of 'center values' as median or average;
- * the definition of error bars as s.d. or s.e.m.

Descriptions that are too long for the figure legend should be included in the Methods section.

done

METHODS: The Methods section, which provides the full, step-by-step instructions that would allow other researchers to replicate the results, is included after the main text figure legends. The Methods section will not appear in print but will appear online in the full-text HTML and PDF versions. The Methods section should be written as concisely as possible but should contain all elements necessary to allow interpretation and reproduction of the results. If there are additional references (in the Methods section, Supplementary Information, etc), their numbering should continue from the last entry in the main text reference list, and they should be listed following the Methods section. Specialized methods that require chemical structures, figures, or tables cannot be accommodated in the Methods section of the main text file. If such information is part of the Methods, the entire Methods section must instead be included within a Supplementary Information text file.

done

ETHICS STATEMENT: For research involving human research participants, the Methods section must include an ethics statement. This statement should provide the name of the committee that approved the study; confirm that the research was performed in accordance with all relevant guidelines and regulations; and confirm that informed consent was obtained from all participants. If the study was granted an exemption from requiring ethics approval, details of the committee granting the exemption must be included.

Research involving human embryos or gametes, or human stem cells in contexts requiring ethical oversight, also must include an ethics statement in the Methods section. This statement should provide the name of the committee that approved the study and confirm that the research was performed in accordance with all relevant guidelines and regulations. We encourage authors to follow the principles laid out in the 2021 ISSCR Guidelines for Stem Cell Research and Clinical Translation. Where necessary, the ethics statement should also describe the conditions of donation of materials, such as human embryos or gametes, and confirm that informed consent was obtained from all donors of cells or tissues.

Not applicable

MAIN TEXT STATEMENTS: Several statements (which will not appear in print but will appear online in the full-text HTML and PDF) are required after the Methods (and additional references, if present). First, there should be an Acknowledgements section, listing grant/financial support. Next, we require a detailed Author Contribution statement; the specific contributions of each

author, particularly in terms of which authors performed which specific experiments, must be listed. This is followed by a Competing Interest statement. Financial and non-financial interests should be noted here, as well as any patents; patent information should include at a minimum patent number, what is covered by the patent, and who submitted the patent application. Finally, an Additional Information statement should include information regarding reprints and permissions and name the author(s) to whom correspondence and requests for materials should be addressed. Formatting details and an example are available here.

We have provided all requested statements

DATA AND CODE AVAILABILITY STATEMENTS: Any manuscript reporting original research must include a Data Availability statement that makes transparent to the reader the conditions of access to the “minimum dataset” that is necessary to interpret, verify and extend the research in the article. This minimum dataset may be provided through deposition in public community/discipline-specific repositories, custom proprietary repositories (for certain types of datasets), or general repositories like Figshare, Zenodo and Dryad. We strongly discourage providing large datasets in Supplementary Information; the preferred approach is to make data available in repositories. More information on Nature Portfolio’s reporting standards and guidance on preparing your Data Availability statement can be found here.

Done; X-ray structure is deposited in pdb data bank, other data is available in manuscript and SI.

For all studies using custom code or mathematical algorithms that are deemed central to the conclusions, a Code Availability statement must be included, indicating whether and how the code or algorithm can be accessed, including any restrictions to access. The Code Availability statement is listed as a separate section after the Data Availability statement but before any additional references. Code should be deposited in a DOI-minting repository such as Zenodo, Gigantum or Code Ocean and cited in the reference list. Authors are encouraged to manage subsequent code versions and to use a license approved by the open source initiative. Additional details can be found here.

Not applicable

DISPLAY ITEMS: We suggest that you take stock of all data that have been generated throughout the review process and ensure that only the data most central to the conclusions are presented in the main text figures. Any figures included within the main text file during the review process must be removed from the final main text file and uploaded as separate, individual files; they will be integrated into the main paper in print and online. An overview of the key features of this presentation may be found here.

Done, figures have been uploaded separately as high resolution pdf files (main figures and extended data figures)

Figures should be comprehensible to readers in other disciplines and assist in understanding of the paper. Main text figures (but **not** Extended Data) must be provided in production-quality versions in an editable format (i.e., .ai, .cmx, .cdr, .doc, .eps, .pdf, .ppt, .ps, .psd, .svg and .xls); we cannot accept figures in .cvs, .gif, .jpg, .png and .tif formats. We highly encourage you to consult our artwork guidelines. They should be as small and simple as is compatible with clarity. All panels of a figure should be logically connected and assembled on a single page in a rectangular shape; any essential alignments (parts horizontal, vertical, spacings, etc) should be indicated. Each panel of a multipart figure should be sized so that the whole figure can be proportionally reduced and reproduced on the printed page at the smallest size at which essential details are visible. Nature’s standard figure sizes are either 9 or 18 cm wide; the maximum permitted height is 17 cm. Panels should be arranged to fit these widths while minimizing excess space around the panels. Tables should be prepared using the Table menu in Word. As we must be able to edit the figures so that they conform to our house style, the submission of files that are incorrectly formatted, flattened, or of insufficient resolution may delay final acceptance of your manuscript.

We confirm that our figures comply with these requests.

THIRD PARTY RIGHTS: You must provide proof that you have secured permission to use any third party materials that appear in any part of your manuscript, including Extended Data and Supplementary Information. Please fill out a Third Party Rights Table, and upload this with the final version of your manuscript. Third party materials include any figures, tables, images, videos or text boxes that are reproductions or adaptations of items that have previously been published elsewhere and/or are owned by a third party. This includes pictures taken by professional photographers, maps and images downloaded from the internet. You will need to obtain the right to use each of these items before your paper can be accepted for publication. You will also need to give proper attribution to the copyright holders in your paper. Please ensure you upload any necessary grants of rights alongside the final version of your manuscript. More information is available on our Rights and permissions page. Failure to obtain the appropriate rights and to supply a completed third party rights table will delay the publication of your article. The editorial assistant (cc'd) can help with any questions.

No third-party rights material.

COVER ARTWORK: We welcome submissions of artwork for consideration for our cover. More information can be found in our guide for cover artwork. The file name(s) should include the manuscript reference number and be labelled as a cover suggestion; a short description is also preferred. Illustrations should be selected more for their aesthetic appeal than for their scientific content. We cannot promise that your suggestions will be selected for the cover, as competition is intense.

No cover artwork submitted.

CHEMICAL STRUCTURE PRESENTATION: Any chemical structures in the main text or Extended Data figures must conform to our chemical structure style guide. This guide lists the ChemDraw preferences and stylesheet that must be used to draw all structures. The style and size of chemical structures should not be modified from the default settings in the template, unless absolutely necessary (see the guide for examples), in which case 80% size and 5 pt font is the smallest size possible. Please export any ChemDraw (.cdx) files as a PDF, retaining editing capabilities — we find that 'print to pdf' works well for this — and upload this with your manuscript.

done

IMAGE INTEGRITY: We strongly advise that you go carefully through all the data (including Extended Data and Supplementary Information) to ensure there are no accidental image/data duplications, other image manipulations or data errors. Such issues generally require correction after publication. Any image provided for publication, either in print or online (including Extended Data and Supplemental Information), may be subject to a quality control process to check for image integrity and manipulation. A discussion of our standards regarding how images should be prepared and presented can be found here.

done

EXTENDED DATA: Extended Data do not appear in print but are included online within the full-text HTML and integrated in the downloadable PDF. Extended Data are an integral part of the paper, and only data that directly contribute to the main message should be included. All Extended Data must be referred to in the main text, and their legends should be listed sequentially at the end of the main text file, not in the Extended Data files. Extended Data should be assembled into a maximum of 10 A4 size, multi-panelled display items. They must be supplied as individual files in .jpg, .tif or .eps format **only**. They should be of the same quality as the main figures, but there are important differences in their formatting. More specific instructions are provided here. If you need to describe a complex process, we encourage you to add a schematic of the main finding as part of the Extended Data to aid readers unfamiliar with the immediate discipline.

We uploaded Extended Data figures separately in high resolution, put the corresponding figure legends at the end of the manuscript and also uploaded a compiled pdf file with all extended data figures and figure legends.

SUPPLEMENTARY INFORMATION: Supplementary Information (SI) is online-only, peer-reviewed material that is essential background to the study (e.g., large data sets, more complex methods, and calculations), but which is too large or impractical, or of interest only to a few specialists, to justify inclusion in the print version of the paper (see here for further details). While SI should not typically contain data figures (any figures additional to those appearing in the main text should be formatted as Extended Data), we require that the raw, uncropped data for gels be presented as an SI figure (see below). Tables may be included in SI, but only if they are unsuitable for formatting as Extended Data (e.g., tables containing large data sets that cannot fit a single page or raw data tables that are best suited to Excel files). If a manuscript has SI, each discrete SI item (e.g., videos, tables) must be referred to at an appropriate point in the main text file. You must also provide a Word file entitled "SI Guide", containing a cover page with manuscript title and author information; a table of contents (preferably with page numbers); and then any SI text, notes, figures, and titles and legends for any separate SI files; for additional information see here.

done

We recommend that you pay careful attention to the formatting of the SI because it is not subedited. After the paper has been accepted, SI files can only be amended for critical changes to the scientific content, not for style.

CELL LINE IDENTIFICATION: To help curb the inadvertent use of cross-contaminated or misidentified cell lines, we ask that you check your reagents against the list of commonly misidentified cell lines maintained by the International Cell Line Authentication Committee, which is also accessible through the NCBI BioSample database. If you have used a cell line that is on this list, you must provide a scientific justification and state the identity issue in the Methods. The editors reserve the right to demand that the data be removed from the paper if the justification is deemed unsatisfactory. In addition, authors must identify the source of cell lines (with catalog number if obtained from a vendor or cell bank) and report whether the cell lines have been authenticated, including the method used, the results, and the date authentication testing was last performed for that cell line. You should be able to provide the test results upon request. Mycoplasma contamination testing status must also be reported. These requirements will be particularly scrutinized for cancer studies, where the issue of cell line misidentification has been well documented. Resources on cell line authentication are available here.

Not applicable.

SOURCE DATA (GRAPHS): To increase transparency, we strongly encourage you to provide, in spreadsheet form, the data underlying the graphical representations used in figures. For all experiments presenting data from animal models, this is a requirement and is not optional. This is in addition to our well-established data-deposition policy for specific types of experiments and large datasets. Online readers of the manuscript will be able to access the graphical source data directly from the figure legend. Spreadsheets must be submitted in .xls, .xlsx or .csv formats. One file per figure is permitted. If there is a multi-panelled figure, the source data for each panel should be clearly labeled in the file; alternatively the source data for a figure can be included in multiple, clearly labeled sheets within an Excel file. File sizes of up to 30 MB are permitted, but it is expected that the vast majority of graphical source data files will be considerably smaller than this. When submitting these files with your manuscript, you should select the "Source Data" file type and use the title field in the file description tab to indicate the figure(s) to which the source data pertain. Source data should not be provided as Extended Data.

We have compiled and uploaded a corresponding Source Data File.

RAW DATA (GELS): You must provide the original source images for all data obtained by

electrophoretic separation (e.g., EMSA, northern/Southern/western blots, etc). The raw images must be assembled into a single .pdf or .tif file (multiple gels on a single page is encouraged). The file should be uploaded as Supplementary Figure 1. The full scanned images must be in uncropped form and contain labeled size/molecular weight markers and loading controls. There should be an accurate indication of how the gels were cropped for the final figure. The figure legends and raw data files should indicate whether controls (such as beta-actin) were run on the same gel as loading controls, or on separate gels as sample processing controls (see here for guidance). While the data can be displayed in a relatively informal style, there must be a correspondence between each source data image and a specific main text or Extended Data figure. The main text or Extended Data figure legends should refer to the uncropped scans explicitly (e.g., “For gel source data, see Supplementary Figure 1.”). For examples, see here or here.

Provided in Supplementary Figure 21.

TUMOUR DATA: The Methods must detail the maximal tumour measurements/volume/other endpoints that were permitted by your IACUC and state that these limits were not exceeded in any of the experiments. You must also provide the source data for any figures presenting an analysis of tumour growth. Where appropriate, these data should be displayed as the longitudinal measurements per mouse. They should be uploaded as a “Source Data” file type.
not applicable.

DATA DEPOSITION: The following specific points may be relevant to your paper, so please ensure that you provide the following information:

* Sequences for any RNAi/small RNA constructs must be included.

* Accession numbers for gene expression data or RNA sequencing data must be listed.

* Papers reporting protein structures must conform to our standards listed in the Guide to Authors. The Data Availability statement must state that the X-ray crystallographic coordinates and structure factor files (or comparable NMR or cryoEM data) have been deposited in the appropriate, named, public database, along with all relevant accession number(s). You must use the standard Nature templates for structural data; there are separate links to tables for X-ray crystallographic, NMR and cryoEM structures. These tables must be presented as Extended Data; if the number of entries causes the table to exceed a page, it must be divided into two Extended Data items. The contour level of any electron density maps presented, as well as the type of map (i.e., Fo-Fc or 2Fo-Fc), should be explicitly stated in the figure legend.

Done.

* For every new chemical compound, a complete description of the synthesis and the physical characterization (i.e., NMR, MS, etc) must be included in the Supplementary Information (see here).

Ok.

* Papers containing new or revised formal taxonomic nomenclature for animals, whether living or extinct, are accepted conditional on the provision of LSIDs (Life Science Identifiers) by means of registration of such nomenclature with ZooBank, the online registration system for the International Code of Zoological Nomenclature (ICZN). ZooBank LSIDs can be resolved and the associated information viewed through any standard web browser by appending the LSID to the prefix "<http://zoobank.org/>".

Not applicable

* We strongly encourage deposition of 3D morphological data in a suitable repository such as MorphoBank, MorphoSource or similar; the relevant accession numbers should be listed in the Data Availability statement.

Not applicable

* For animal experiments, you must confirm that all experiments were performed in accordance with relevant guidelines and regulations. There should be a statement identifying the institutional and/or licensing committee approving the experiments, including any relevant details. Sex and other characteristics of animals that may influence the results must be described. Details of housing and husbandry must be included if they are likely to influence experimental results. Further details can be found here.

Not applicable

* Human genotype data (e.g., SNP array data) should be deposited into a public database (dbGAP or EGA) with a controlled access policy.

Not applicable

* A full clinical and pathological characterization of patients/human subjects and samples should be provided in tabular format, including the magnitude of response for each patient (partial, complete, stable disease), the site of the biopsy, whether or not that lesion was progressing and mutational status if appropriate.

Not applicable

We will not send your revised paper for further review. If the revised paper is in our format (as detailed above), in accessible style and of appropriate length, we shall begin the acceptance process.

In order to accept your paper, we require the following electronic files:

* A cover letter describing your response to any editorial comments and detailing any format changes during revision, particularly if the overall length is affected.

provided

* A point-by-point response (preferably in Word) to any remaining issues raised by our referees.

No remaining issues.

* The final version of your text as a Word document. Word Equation Editor/MathType should be used only for formulae that cannot be produced using normal text or symbol font. If this is not possible, the manuscript can be supplied as a single plain vanilla TeX or LaTeX file that includes all references and abbreviations, with no special formatting, as well as a PDF version that is uploaded as a 'related manuscript file'.

provided

* Production-quality versions of all figures (see above).

provided

* The final version of the Extended Data.

provided

* The final version of any Supplementary Information, presented as one file (ideally a PDF) if feasible, as well as a separate SI Guide.

provided

* Source Data, if appropriate.

provided

* For optimal quality videos we encourage H.264 encoding and a standard aspect ratio of 16:9 (4:3 is second best), without compression.

not applicable

* Completed and signed copies of the following **four (or five) forms**, uploaded as a "Related Manuscript File" file type:
provided

- 1) Biology editorial checklist;
- 2) Manuscript checklist;
- 3) Reporting summary;
- 4) Third-party rights table;
- 5) Code and software submission checklist (if applicable).

Nature has now transitioned to a unified Rights Collection system which will allow our author services team to quickly and easily collect the rights and permissions required to publish your work. Once your paper is accepted, you will receive an email in approximately 10 business days providing you with a link to complete the grant of rights. If you choose to publish Open Access, our author services team will also be in touch at that time regarding any additional information that may be required to arrange payment for your article. If you have any questions please contact asjournals@springernature.com.

You may need to take specific actions to achieve compliance with funder and institutional open access mandates. If your research is supported by a funder that requires immediate open access (e.g. according to Plan S principles) then you should select the gold OA route, and we will direct you to the compliant route where possible. If you select the subscription publication route our standard licensing terms will need to be accepted, including our self-archiving policies. Those standard licensing terms will supersede any other terms that you or any third party may assert apply to any version of the manuscript.

All of the files should be uploaded using the following link:

<https://mts-nature.nature.com/cgi-bin/main.plex?el=A2K3L7A1IObf7J3A9ftdjfHYyYvgyBufIA7Ov93VQwZ>

NOTE: This url links to your confidential home page, which contains information about other manuscripts you may have submitted or be reviewing for us. Therefore, if you wish to forward this email to co-authors, please delete the link to your home page first.